# Antarctic snow-covered sea ice topography derivation from TanDEM-X using polarimetric SAR interferometry

Lanqing Huang[1], Georg Fischer[2], and Irena Hajnsek[1,2]

[1]Institute of Environmental Engineering, Swiss Federal Institute of Technology in Zurich (ETH), 8093 Zürich, Switzerland.
[2]Microwaves and Radar Institute, German Aerospace Center (DLR), Wessling 82234, Germany.

**Correspondence:** Lanqing Huang (huang@ifu.baug.ethz.ch)

**Abstract.** Single-pass interferometric synthetic aperture radar (InSAR) enables the possibility for sea ice topographic retrieval despite the inherent dynamics of sea ice. InSAR digital elevation models (DEM) are measuring the radar scattering center height. The height bias induced by the penetration of electromagnetic waves into snow and ice leads to inaccuracies of the InSAR DEM, especially for thick and deformed sea ice with snow cover. In this study, an elevation difference between the satellite-measured InSAR DEM and the airborne-measured optical DEM is observed from a coordinated campaign over the Western Weddell Sea in Antarctica. The objective is to correct the penetration bias and generate a precise sea ice topographic map from the single-pass InSAR data. With the potential of retrieving sea ice geophysical information by the polarimetric-interferometry (Pol-InSAR) technique, a two-layer plus volume model is proposed to represent the sea ice vertical structure and its scattering mechanisms. Furthermore, a simplified version of the model is derived, to allow its inversion with limited a priori knowledge, which is then applied to a topographic retrieval scheme. The experiments are performed across four polarizations: HH, VV, Pauli-1 (HH+VV), and Pauli-2 (HH-VV). The model-retrieved performance is validated with the optical-derived DEM of the sea ice topography, showing an excellent performance with root-mean-square error as low as $0.26\,\mathrm{m}$ in Pauli-1 (HH+VV) polarization.

## 1   Introduction

Sea ice topography is defined as the elevation of the ice volume including the snow cover above the sea level. The sea ice topographic height on spatial scales of meters is dominated by ice ridges, shear zones, and hummocks, due to the forces from ocean winds and currents, together with the blocking effects from the coast and islands (Rampal et al., 2009). Timco and Burden (1997) estimated the ratio of the keel-depth (i.e., depth of ice below the seawater) to sail-height (i.e., height of ice above the seawater) for both first-year and multi-year sea ice ridges in the Beaufort Sea and highlighted their differences in ridge height and shape. Haas et al. (1999) presented the pressure ridge frequencies to be $3-30$ ridges per kilometre over Bellingshausen, Amundsen, and Weddell Seas in Antarctica. Tin and Jeffries (2003) indicated that first-year ridges in the Antarctic are flatter and less massive than those in the Arctic. Sea ice ridging height is a crucial parameter to evaluate total ice mass in both polar regions (Hibler et al., 1974; Melling and Riedel, 1995; Lytle et al., 1998; Tin et al., 2003). In the Antarctic, the mean height of

the ridges in the Weddell Sea was found to be $\sim 1.1\,\mathrm{m}$ which is similar to the ridging statistics from the Ross Sea (Lytle and Ackley, 1991), whereas it is considerably less than in the Arctic (Lytle and Ackley, 1991; Dierking, 1995).

The snow layer and ice properties in the Arctic and Antarctic are significantly different due to the diverse growing conditions in the two polar regions (Gloersen, 1992; Walsh, 2009; Sturm and Massom, 2009; Webster et al., 2018). In the Antarctic, the snow depth is reported to be thicker than in the Arctic (Jeffries et al., 1997; Massom et al., 2001; Willatt et al., 2009). When thick enough, the snow will overburden the ice floe and be flooded by seawater, resulting in higher salinity of the snow layer in the Antarctic. Besides, compared to the Arctic, snow on the Antarctic sea ice comprises more heterogeneous layers resulting from highly variable temperature (Massom et al., 2001). The layer heterogeneity in types, density, salinity, and wetness determines the electromagnetic characteristics of the snow. As for ice properties in general, Antarctic sea ice is reported to be thinner (Worby et al., 2008; Kurtz and Markus, 2012; Lindsay and Schweiger, 2015), younger (Webster et al., 2018), and more saline than in the Arctic at comparable age and thickness (Gow et al., 1982). Quantitatively, the mean salinity of the Antarctic first-year ice and multi-year ice profiles are $4.6‰$ and $3.5‰$, respectively, whereas the average values are $3‰$ for the first-year ice and $2-2.5‰$ for the multi-year ice in the Arctic (Cox and Weeks, 1974). These variable properties of sea ice, its ridges and snow cover, at both small and large spatial scales, highlight the challenge and necessity for accurate sea ice topographic information with large spatial coverage and high resolution.

Characterizing sea ice topography is valuable for various geophysical parameters over polar oceans. For instance, the atmospheric drag coefficient over sea ice is an important topography-dependent parameter to understand the interaction at the ice-atmosphere boundary (Garbrecht et al., 2002; Castellani et al., 2014). Interpretation of sea ice topography is also essential in estimating sea ice thickness. Focusing on first-year sea ice in the Alaska region, Tucker and Govoni (1981) observed a square-root relation between the ridge height and thickness, which is further validated by additional in situ observations in (Tucker et al., 1984). Petty et al. (2016) presented a detailed characterization of Arctic sea ice topography across both first-year and multi-year sea ice and analyzed the topographic differences between the two ice regimes. A square-root relation function between sea ice topographic height and thickness was established for ice thickness retrieval (Petty et al., 2016). The results demonstrated a maximum $\pm 2\,\mathrm{m}$ difference between the measured and predicted ice thickness. Note that the measured thickness ranges from $0$ to $8\,\mathrm{m}$ with an initial uncertainty of $0.8\,\mathrm{m}$ (Petty et al., 2016). In Antarctica, Toyota et al. (2011) demonstrated that the mean ice thickness of snow-covered ice is highly correlated with the sea ice topography. Nowadays, precise characterization of sea ice topography is a topic of active research.

The sea ice topography can be measured by various instruments, such as laser altimeters (Dierking, 1995; Schutz et al., 2005; Abdalati et al., 2010; Farrell et al., 2011, 2020) and stereo cameras using photogrammetric techniques (Dotson and Arvesen., 2012, updated 2014; Divine et al., 2016; Nghiem et al., 2018; Li et al., 2019). However, the major limitation of above measurements is the small spatial coverage. Synthetic aperture radar (SAR) achieves a good balance between wide-spatial coverage and high resolution and becomes an invaluable asset for monitoring polar regions thanks to its ability to provide all-weather, day/night imagery. Interferometric SAR (InSAR) offers an opportunity to estimate surface height from two or more image pairs (Rodriguez and Martin, 1992). However, due to the inherent dynamics of sea ice, it is impossible to derive height over sea ice from a single SAR sensor by repeat-pass interferometry, because of its temporal decorrelation. Only single-pass

interferometry offers the possibility to characterize the sea ice topography (Dierking et al., 2017). TanDEM-X is a single-pass
SAR interferometer developed by the German Aerospace Center (DLR) (Krieger et al., 2007) and is providing high resolution
co-registered data on a global scale. Dierking et al. (2017) demonstrated the theoretical potential of generating sea ice height
from single-pass InSAR data and discussed the factors that may impede the accuracy of the retrieval. From TanDEM-X InSAR
acquisitions, the derivation of topography over snow-free multi-year sea ice was demonstrated and verified with laser and
photogrammetric measurements (Yitayew et al., 2018). Until now, the InSAR technique has become one of the most promising
tools for sea ice height estimation.

However, a digital elevation model (DEM) derived with InSAR is affected by the penetration of microwave signals into
dry, frozen snow and ice. In fact, an InSAR DEM is actually a measurement of the radar scattering center height, which can
be below the surface due to the microwave penetration. This height bias leads to inaccuracies of InSAR DEMs, especially
for multi-year sea ice with snow cover. The microwave penetration into snow and ice is described by the electromagnetic
penetration depth $\delta_p$. It is determined by the signal extinction coefficient $\sigma$ in units of decibels per unit length $(\mathrm{dB/m})$, which
indicates the decrease of the signal strength inside the medium. The total electromagnetic loss in a medium consists of both
scattering and absorption losses. Scattering loss results from particles of different relative permittivity embedded in a host
medium. The absorption loss depends on the imaginary part of the relative permittivity $\epsilon''$ (Hallikainen and Winebrenner,
1992). Larger $\delta_p$ are found in multi-year ice due to the smaller $\epsilon''$ attribution to reduced brine compared to first-year ice. For
sensors operating at X-band, experimental penetration depth for sea ice ranges from about $0.05\,\mathrm{m}$ to $1\,\mathrm{m}$, depending on the sea
ice type, salinity, and temperature (Hallikainen and Winebrenner, 1992). Snow on top causes a greater range of $\delta_p$ due to the
high sensitivity of $\epsilon''$ to water content. Dry and fine grained snow can have $\delta_p$ values up to hundreds of wavelengths (Cloude,
2010).

In this study, we observed an elevation discrepancy between the InSAR DEM and the photogrammetric DEMs which were
acquired in a coordinated campaign (Nghiem et al., 2018) conducted with DLR's TanDEM-X satellite and the NASA IceBridge
aircraft over the sea ice in the Western Weddell Sea, Antarctica. The elevation difference reveals the necessity to consider the
penetration depth $\delta_p$ when retrieving sea ice topography from the InSAR imagery. The objective of this study is to compensate
the penetration depth and thereby obtain a more accurate sea ice topographic map with wide spatial coverage. Note that the
studied area is snow-covered sea ice, therefore the term "sea ice topographic height", throughout the paper, refers to the sea ice
height including snow depth above local sea level.

The estimation of the penetration depth of InSAR signals can be inferred from the interferometric volume decorrelation,
which is one of the key components of the interferometric coherence. The volume decorrelation is caused by backscatter
contributions from different depths and can be derived from the integral of an assumed vertical scattering distribution function.
The investigation of vertical distribution functions for various scattering processes, known as the polarimetric-interferometry
SAR (Pol-InSAR) technique (Papathanassiou and Cloude, 2001), is widely applied in retrieving geophysical parameters from
natural volumes, such as forests (Kugler et al., 2015), agriculture (Joerg et al., 2018), ice sheets (Fischer et al., 2018), and
glaciers (Sharma et al., 2012). To the best of our knowledge, few studies have assessed the potential of retrieving geophysical
information by means of the Pol-InSAR technique for sea ice. Dierking et al. (2017) estimated the penetration depth into sea ice

volume under the assumption of a uniform lossy volume with an exponential vertical function. However, for snow-covered sea
ice, the scattering effects from the snow volume and sub-layers, such as the snow-ice interface, also need to be considered. To
achieve an effective estimation of penetration depth, factors including the physical structures, the electromagnetic properties,
as well as the scattering mechanisms within the sea ice volume need to be understood and properly modelled.

Sea ice can be modelled as a multi-layer structure behaving as a mixture of surface, volume, and surface-volume interaction scattering in microwave remote sensing (Nghiem et al., 1995a, b; Albert et al., 2012). When the electromagnetic waves
penetrate the volume, the inhomogeneous materials inside the volume (i.e., a mixture of constituents such as brine, ice, and
air bubbles) excite the occurrence of volume scattering (Nghiem et al., 1995a). Besides, the surface conditions such as rough
interfaces, hummocks, and snow cover can increase the surface scattering at the rough air-snow interface, snow-ice interface,
and ice-water interface (Nghiem et al., 1990). In the Arctic first-year thin ice, snow capillary force gives rise to brine wicking,
and consequently, a layer of high salinity slush ice appears at the snow-ice interface (Reimnitz and Kempema, 1987; Drinkwater and Crocker, 1988; Nghiem et al., 1995a). In the Antarctic, ice-surface flooding widely occurs resulting from the generally
thicker snow layer loading on the thinner ice floes, often followed by freezing of the slush layer at the snow-ice interface (Massom et al., 2001; Jeffries et al., 2001; Maksym and Jeffries, 2000). Even without flooding, the upward wicking of brine from
the ice surface can also form a saline layer at the bottom of the snowpack (Massom et al., 2001; Toyota et al., 2011; Webster
et al., 2018). The slush layer at the snow-ice interface can induce significant surface scattering and thus has been included in
sea ice scattering modelling (Nghiem et al., 1995a, b; Maksym and Jeffries, 2000). Moreover, the surface-volume interaction
components (Albert et al., 2012) further complicate the overall scattering mechanisms.

Huang and Hajnsek (2021) investigated the X-band SAR polarimetric behaviour for several types of ice over the Western
Weddell Sea, including new ice, thin ice, thick ice, and deformed ice with ridges. For the area covered by the thick and deformed
ice, an empirically inverse relation between the elevation difference (i.e., penetration bias) and the co-polarimetric coherence
was observed, indicating that SAR polarimetry carries significant topographic information (Huang and Hajnsek, 2021). Based
on (Huang and Hajnsek, 2021), this study offers a further understanding of the InSAR penetration bias by investigating the
polarimetric behaviour and exploiting the interferometric volume decorrelation. A novel model is proposed to characterize the
scattering processes, and an inversion scheme is developed for height retrieval. Therefore, compared to the previous work, this
study is a crucial step forward towards developing an advanced algorithm for sea ice topographic estimation.

In this study, inspired by multi-layer sea ice models utilized for electromagnetic simulation (Nghiem et al., 1990, 1995a, b;
Albert et al., 2012), a two-layer plus volume model is proposed to relate interferometric coherence to extinction coefficients,
layer depths, and layer-to-volume scattering ratios. The model sensitivity to the variation of several parameters is analyzed,
and the model accuracy is assessed with various baseline configurations. With the goal to develop and invert the model for
sea ice topographic retrieval, the proposed theoretical model is further simplified by reducing the required amount of input
parameters. An inversion scheme for topographic retrieval using both the theoretical and simplified model is established. The
sea ice topographic height is retrieved in different polarizations over around $50\,\mathrm{km} \times 18\,\mathrm{km}$ in the Western Weddell Sea. The
model-retrieved DEM is validated against a photogrammetric DEM, proving the effectiveness of the proposed model and the
inversion scheme.

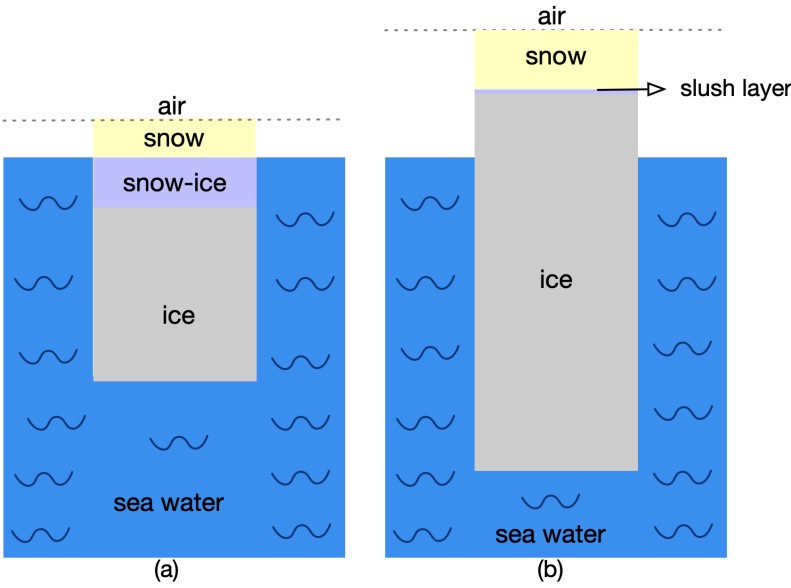

**Figure 1.** Schematic of (a) thinner ice floes flooded by seawater and (b) thicker ice floes without flooding.

The organization of the paper is as follows. Section 2 introduces the basic concepts of Antarctic sea ice conditions and the Pol-InSAR technique. Section 3 introduces the data sets and the preprocessing procedures. A two-layer theoretical model and a simplified model are proposed in Section 4. A model inversion scheme is developed in Section 5 to achieve the sea ice topographic retrieval. In Section 6, the proposed methodology is applied to the study area, and the experimental results are analyzed. More discussions about the model are given in Section 7 and the conclusion is drawn in Section 8.

## 2 Basic concepts

### 2.1 Sea ice conditions

In the Antarctic, the presence of a saline layer at the snow-ice interface due to the flooding or capillary suction of brine from the ice surface has been recognized as a widespread and critical phenomenon (Massom et al., 2001). For thinner ice, flooding may occur when the weight of the snow pushes the ice surface below the water level, yielding a negative freeboard. In this case, as shown in Fig. 1(a), seawater infiltrates into the snowpack, floods the ice surface, and creates a high-saline slush layer which may refreeze into snow ice (Lange et al., 1990; Jeffries et al., 1997; Maksym and Jeffries, 2000). Note that snow ice forms from slush which by definition (and ignoring brine wicking or other second-order processes) locates below the water level, and the freeboard of snow ice is generally considered to be zero. The freeboard of snow ice could become positive if there is subsequent ice growth at the bottom of the ice sheet, without new snow accumulation; however, that is a secondary effect as

snow thickness will generally tend to increase in the meanwhile. The thickness of snow ice was observed to be $\sim 42 - 70\%$ of the total snow accumulation (i.e., the thickness of snow ice plus snow depth) (Jeffries et al., 2001).

For thicker and deformed ice with ridges, less flooding occurs due to the increased buoyancy of the ice mass contained in the ridges (Jeffries et al., 1998). However, even in the absence of flooding, a thin slush layer can also occur due to the capillary suction of brine from the ice surface (Massom et al., 2001; Webster et al., 2018). Besides, the deformed ice in the ridging and rafting area is often poorly consolidated, and thus seawater may reach the snow layer and form a thin slush layer (Maksym and Jeffries, 2000). The sea ice structure for thicker ice without flooding is sketched in Fig. 1(b), including snow on top, the ice volume, and a thin and high-saline layer in between.

The condition of flooding can be quantified by a simple hydrostatic balance (Lange et al., 1990)

$$\rho_w d = \rho_i d + \rho_i f + \rho_s s$$
$$t = d + f$$

(1)

where $\rho_w$ is the seawater density, $\rho_i$ is the ice density, and $\rho_s$ is the snow density; $d$ and $f$ are the thickness of ice below and above the sea level, respectively; $t$ is the thickness of the total ice thickness, and $s$ is the snow depth. In case of a flooding, $f$ should be zero (i.e., $d = t$), and Eq. (1) becomes

$$s/t = (\rho_w - \rho_i)/\rho_s \approx 0.12/\rho_s$$

(2)

by assuming $\rho_w = 1.03\,\mathrm{Mg/m^{-3}}$ and $\rho_i = 0.91\,\mathrm{Mg/m^{-3}}$ (Lange et al., 1990). For snow density $\rho_s$ being $0.3\,\mathrm{Mg/m^{-3}}$ (Lange et al., 1990), the ratio between snow depth and ice thickness $s/t$ is estimated to be $0.4$. The snow depth on Antarctic sea ice during September and November was shown to be below $0.8\,\mathrm{m}$ for $99\%$ of the samples in Webster et al. (2018). This range of snow depth will lead to flooding for ice thickness $< 2\,\mathrm{m}$.

The relation between ice thickness $H_i$ and surface height $h_{\mathrm{sur}}$ (i.e., ice height above sea surface including snow depth) has been discussed over different regions (Petty et al., 2016; Toyota et al., 2011; Ozsoy-Cicek et al., 2013). Ozsoy-Cicek et al. (2013) showed a linear relation $H_i = 2.24 h_{\mathrm{sur}} + 0.228$ fitted from large-scale, survey-averaged data over the Western Weddell Sea, which is the same region as this study. According to this linear relation, $H_i = 2\mathrm{m}$ corresponds to a surface height of $\sim 0.8\,\mathrm{m}$.

This paper focuses on thicker ($> 2\,\mathrm{m}$) and deformed ice (Fig. 1(b)), which is the main ice typology in the studied area. In the following sections, the model and experiments are conducted only for the samples above $\sim 0.8\,\mathrm{m}$ surface height. We assume that samples exceeding this threshold are thicker and deformed ice without flooding. The potential to extend the proposed model to thinner ice scenarios (e.g. Fig. 1 (a)) is discussed in Section 7.3.

## 2.2 PolinSAR technique

The complex interferometric coherence $\tilde{\gamma}_{\mathrm{InSAR}}$ is a measurement of signal correlation between two acquisitions. For single-pass systems, $\tilde{\gamma}_{\mathrm{InSAR}}$ can be decomposed into a product of terms (Cloude, 2010)

$$\tilde{\gamma}_{\mathrm{InSAR}} = e^{i\phi_0} \gamma_s \gamma_{\mathrm{SNR}} \tilde{\gamma}_v$$

(3)

where $\phi_0$ is the topographic phase. $\gamma_s$ is the baseline or surface decorrelation, which depends on the nature of the surface scattering; it can always be removed by employing range spectral filtering and thus is set equal to 1 in this study. $\gamma_{\text{SNR}}$ denotes decorrelation due to additive noise in the signals. $\tilde{\gamma}_v$ refers to the complex volume decorrelation. The topographic phase can be converted to topographic height $h_{\text{topo}}$ by

$$h_{\text{topo}} = \frac{\phi_0}{\kappa_z} = h_a \frac{\phi_0}{2\pi} \tag{4}$$

where $h_a$ is the height of ambiguity and $\kappa_z$ is the vertical wavenumber in free space. Note that $h_a$ corresponds to an interferometric phase change of $2\pi$, and is inversely proportional to the perpendicular baseline between the two acquisitions (Dall, 2007)

$$h_a = \frac{\lambda H \tan\theta}{2 b_\perp} \tag{5}$$

where $\lambda$ is the wavelength, $H$ is orbit height, and $b_\perp$ is the effective perpendicular baseline of the TanDEM-X bi-static mode.

The magnitude of the $\tilde{\gamma}_{\text{InSAR}}$ can be corrected for $\gamma_{\text{SNR}}$ and $\gamma_s$ by rewriting Eq. (3) as

$$\tilde{\gamma}_{\text{InSAR}'} = \frac{\tilde{\gamma}_{\text{InSAR}}}{\gamma_s \gamma_{\text{SNR}}} = e^{i\phi_0} \tilde{\gamma}_v \tag{6}$$

where $\tilde{\gamma}_{\text{InSAR}'}$ is the SNR-removed interferometric coherence. $\gamma_{\text{SNR}}$ can be estimated as a function of $SNR$ (Cloude, 2010)

$$\gamma_{\text{SNR}} = \frac{SNR}{1 + SNR} = \frac{S(\text{dB}) - N(\text{dB})}{1 + S(\text{dB}) - N(\text{dB})} \tag{7}$$

with $S$ being the backscattering signal and $N$ being the noise floor (i.e., the noise equivalent sigma zero (NESZ)) (Eineder et al., 2008).

In the case of pure surface scattering, the interferometric coherence can be approximated to $\tilde{\gamma}_{\text{InSAR}} \approx e^{i\phi_0} \gamma_{\text{SNR}}$, assuming that volume scattering can be neglected (i.e., $\tilde{\gamma}_v \approx 1$). $\gamma_{\text{SNR}}$ only contributes to the magnitude of $\tilde{\gamma}_{\text{InSAR}}$; therefore the InSAR scattering phase center, denoted as $\angle\tilde{\gamma}_{\text{InSAR}}$, purely contains the information of topographic phase $\phi_0$. In this case, $\angle\tilde{\gamma}_{\text{InSAR}}$ can be directly converted to topographic height.

However, in the case of snow-covered, thick and deformed sea ice, when the microwaves penetrate into the snow and ice volume, the inhomogeneous materials inside the volume can excite volume scattering (Nghiem et al., 1995a). Then, the volume decorrelation in Eq. (3) is not 1. Both the topographic phase $\phi_0$ and the complex $\tilde{\gamma}_v$ contribute to the InSAR scattering phase center $\angle\tilde{\gamma}_{\text{InSAR}}$. In this case, in order to obtain an accurate topographic phase $\phi_0$, $\tilde{\gamma}_v$ has to be properly modelled and estimated. The main contribution of this paper is the development of a novel two-layer plus volume model (Section 4) for $\tilde{\gamma}_v$, which is applied for an improved sea ice topographic retrieval.

The volume decorrelation $\tilde{\gamma}_v$ depends on the vertical distribution of backscattering $\sigma_v(z)$ (Cloude, 2010)

$$\tilde{\gamma}_v = \frac{\int_0^D \sigma_v(z) e^{i\kappa_{z\_\text{vol}} z} dz}{\int_0^D \sigma_v(z) dz} \tag{8}$$

where the surface is located at $z = 0$, $D$ is the thickness of volume, and $\kappa_{z\_\mathrm{vol}}$ is the vertical wavenumber in the volume (Sharma et al., 2012; Dall, 2007)

$$\kappa_{z\_\mathrm{vol}} = \frac{2\pi}{h_{a\_\mathrm{vol}}} = \frac{2\pi}{h_a \frac{\sqrt{\epsilon' - \sin^2\theta}}{\epsilon' \cos\theta}} = \frac{\kappa_z}{\frac{\sqrt{\epsilon' - \sin^2\theta}}{\epsilon' \cos\theta}} \tag{9}$$

where $h_{a\_\mathrm{vol}}$ is the height of ambiguity in the volume, $\theta$ is the incidence angle (in air), and $\epsilon'$ is the dielectric constant of the volume and is assumed to be 2.8 (Dierking et al., 2017) throughout this study. The phase of the volume decorrelation $\angle\tilde{\gamma}_v$ can be translated to height $h_{\mathrm{volume}}$ as

$$h_{\mathrm{volume}} = \frac{\angle\tilde{\gamma}_v}{\kappa_{z\_\mathrm{vol}}} \tag{10}$$

In Eq. (8), $\tilde{\gamma}_v$ can be estimated by choosing an appropriate vertical structural function $\sigma_v(z)$ and a suitable InSAR baseline configuration using Eq. (9), and then be substituted into Eq. (3) to obtain the topography of snow-covered sea ice.

## 3 Data sets and preprocessing

This section introduces the campaign, the study area, and the data sets. The InSAR processing and its performance are also described.

### 3.1 Campaign and study area

A coordinated campaign of NASA's Operation IceBridge (OIB) airborne mission and the DLR's TanDEM-X satellite mission was successfully conducted on Oct 29, 2017, named as OIB/TanDEM-X Coordinated Science Campaign (OTASC) (Nghiem et al., 2018). The OTASC data have been successfully used in investigating the topography of icebergs (Dammann et al., 2019) and sea ice (Huang and Hajnsek, 2021).

As presented in Fig. 2(a), the study area is located in the Western Weddell Sea, near the east coast of the Antarctic Peninsula. The TanDEM-X SAR intensity image of the study area is shown in Fig. 2(b) where the optical images and the transect of photogrammetric measurements of the sea ice topography are superimposed. From the optical images of the airborne digital camera, it is visible that the study region consists of snow-covered, thick and deformed ice with ridges.

### 3.2 TanDEM-X

The German TanDEM-X mission is a single-pass SAR interferometer operating at X-band at a wavelength of $3\,\mathrm{cm}$. With nearly no temporal gap, TanDEM-X collects two images of the same footprint seen from slightly different viewing angles to generate the topography of the Earth's surface (Krieger et al., 2007). The studied InSAR images were acquired at UTC 23:41 Oct 29, 2017 in bi-static mode. The InSAR pair is a dual-polarization (HH and VV) StripMap product. The incidence angle of the scene center is $34.8°$, and the pixel spacing is $0.9\,\mathrm{m} \times 2.7\,\mathrm{m}$ in range and azimuth. The effective perpendicular baseline $b_\perp$ is $175.7\,\mathrm{m}$ and the along-track baseline $b_{\mathrm{al}}$ is $201.9\,\mathrm{m}$. The height of ambiguity $h_a$ is $32.5\,\mathrm{m}$.

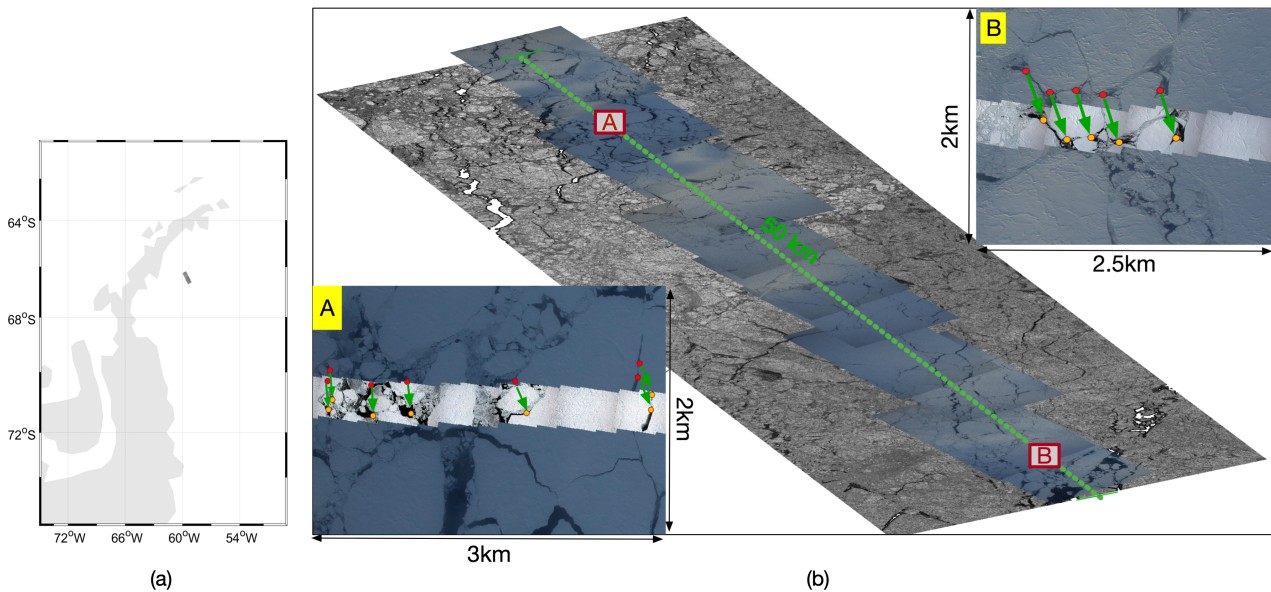

(a)                                    (b)

**Figure 2.** (a) Geo-location of the study area (the grey rectangle). (b) Composite of the SAR intensity image in HH polarization and the airborne DMS digital camera images. The green dashed line indicates the $50\,\mathrm{km}$ transect of the DMS DEM data. Sub-image A and sub-image B are zoom-in of area A and area B with small-scale DMS images (acquired at around UTC 17:50) superimposed on the large-scale DMS images (acquired at around UTC 22:05). The red and yellow dots denote the selected reference pairs from the large- and small-scale DMS image, respectively. The green arrow denotes the 'shift-vector' which is used for data co-registration.

### 3.3  DMS

The digital mapping system (DMS) is one of the OIB instruments acquiring different data from a digital camera system. This study uses the airborne DMS digital camera images (Dominguez, 2010, updated 2018) and the airborne DMS DEMs (Dotson and Arvesen., 2012, updated 2014).

The DMS digital camera captures natural color and panchromatic imagery, hereby named as DMS images for conciseness. The DMS images are geolocated and orthorectified, with a high spatial resolution varying from $0.015\,\mathrm{m}$ to $2.5\,\mathrm{m}$ depending on the flight altitude (Dominguez, 2010, updated 2018). Two types of photography: large- and small-scale DMS images, are obtained during the airborne overflights. The large-scale DMS images over the study area were acquired from UTC 22:01 to 22:07 on Oct 29, 2017, each with a spatial coverage of around $5.8\,\mathrm{km}$ by $8.8\,\mathrm{km}$, shown in Fig. 2(b). The small-scale DMS

images were captured from UTC 17:45 to 17:52 on Oct 29, 2017 with each about $400\,\mathrm{m}$ by $400\,\mathrm{m}$ spatial coverage. The transect of small-scale DMS images is shown in the green dashed line in Fig. 2(b) and enlarged in Fig. 2(b) sub-image A and B, where the details of sea ice structure become visible.

The DMS DEM is generated from the small-scale DMS images by a photogrammetric technique and is calibrated with LIDAR measurements (Dotson and Arvesen., 2012, updated 2014). For the snow-covered sea ice, the DMS DEM measures

sea ice height including snow depth. The data are acquired along a $50\,\mathrm{km}$ transect with a swath width of $400\,\mathrm{m}$ (Dotson and Arvesen., 2012, updated 2014) and $40 \times 40\,\mathrm{cm}$ spatial resolution. Note that the temporal gap between the DMS DEM and TanDEM-X SAR acquisitions is about 6 hours.

In this study, the DMS DEMs are further processed following four steps: reprojection, mosaicing, geocoding, and calibration. First, the DMS DEMs are reprojected from Antarctic Polar Stereographic to WGS84 spatial reference. The second step is the

mosaicing of adjacent files into the $50\,\mathrm{km}$ transect with a swath width of $400\,\mathrm{m}$. Third, using the GAMMA software, the merged DMS DEM is geocoded into the SAR coordinate system and re-sampled into the same resolution (i.e., $\sim 10 \times 10\,\mathrm{m}$ in range and azimuth) as the multilook TanDEM-X image. Finally, as the DMS DEMs are given in meters above the WGS-84 ellipsoid, the sea ice topographic height in this paper is calibrated to the local sea level by selecting the water-surface reference from DMS images. In total, we label nine points as water-surface reference according to the DMS images. Since the interferometric

coherence magnitude over water is very low, it can also be used to classify open water (Dierking et al., 2017). All the nine points have an interferometric coherence magnitude below $0.3$, which is the threshold of the open-water area mask in (Huang and Hajnsek, 2021). The average height of the open-water points is subtracted from the DMS DEMs to obtain the sea ice topographic height relative to the local sea level.

### 3.4   Data co-registration

Due to the inherent dynamics of sea ice and the temporal gap between the DMS DEM and TanDEM-X acquisitions, data co-registration is employed to cancel the shift and thereby ensure a valid pixel-by-pixel comparison.

The large- and small-scale DMS images, although acquired at different times, both clearly reveal the shape and size of ice floes; therefore they are used to match identical sea ice features, referred to as 'reference pair' in the following. The co-registration is performed by tracking the movement of the selected reference pairs. Specifically, we manually label several

pairs of distinguishable features (i.e. the ice floes of a particular shape, or leads) on both the large-scale DMS image and the small-scale DMS image, which is acquired about 4 hours 15 minutes later. By extracting the two geo-locations of the reference pair, a 'shift-velocity vector' can be derived. As the temporal difference between the DMS DEM and the SAR acquisition is 5 hours $49 - 56$ minutes, the shift-distance can be estimated, assuming constant sea ice motion, based on the 'shift-velocity vector' and utilized for the co-registration.

In the studied image, the focus is the $50\,\mathrm{km}$ transect (green dashed line in Fig. 2(b)) of the DMS flight track. The transect is divided into 50 segments, and each segment contains $11 \times 100$ pixels in range and azimuth corresponding to about $110 \times 1000\,\mathrm{m}$ area. For each segment, several reference pairs are selected and labelled (marked in red and yellow dots on the large- and small-scale DMS images, respectively, in Fig. 2(b) sub-image A and B). The 'shift-velocity vectors' are calculated and annotated by the green arrows. Then, data co-registration is conducted by multiplying the derived 'shift-velocity vector' with the temporal

gap for each segment respectively. The co-registered results of all segments are confirmed by the visualization of the DMS images and the SAR images. Among the 50 segments, 12 segments which still contain residual mis-coregistration induced by the sea ice non-linear movement or rotation are excluded and will not be used in the following experiments. $76\%$ of the segments from the whole SAR scene are accepted as correctly co-registered segments in this study.

## 3.5 InSAR processing

The TanDEM-X InSAR pair is already co-registered and common spectral band filtered in range and azimuth (Duque et al., 2012). The remaining InSAR processing includes interferogram generation, flat earth removal, interferogram filtering, low-coherence area mask, phase unwrapping, and phase-to-height conversion, see details in (Huang and Hajnsek, 2021). All steps are carried out with the GAMMA software.

In single-pass interferometry, two simultaneous observations, denoted as $s_1$ and $s_2$, are made. The complex interferogram $\gamma$ and interferometric phase $\phi_\gamma$ can be calculated as (Cloude, 2010)

$$\gamma = s_1 s_2^* \tag{11}$$

$$\phi_\gamma = \arg\{s_1 s_2^*\} \tag{12}$$

where symbol $(^*)$ denotes the complex conjugate.

The interferometric coherence is estimated by (Cloude, 2010)

$$\tilde{\gamma}_{\mathrm{InSAR}} = \frac{< s_1 s_2^* >}{\sqrt{< s_1 s_1^* > < s_2 s_2^* >}} \tag{13}$$

where the symbol $< . >$ denotes an ensemble average. Here, a $4 \times 12$ window in azimuth and slant range, corresponding to about $10\,\mathrm{m} \times 10\,\mathrm{m}$ spatial size, is applied to estimate $\tilde{\gamma}_{\mathrm{InSAR}}$ for four polarizations: HH, VV, Pauli-1 (HH+VV), and Pauli-2 (HH-VV). Areas with $|\tilde{\gamma}_{\mathrm{InSAR}}|$ less than $0.3$ are masked out and will not be considered in the following processing. For conciseness, only the interferometric coherence in HH polarization is shown in Fig. 3(a). The $|\tilde{\gamma}_{\mathrm{InSAR}}|$ histograms for the four polarizations are plotted in Fig. 4, from which the interferometric decorrelation varying among different polarizations can be observed. $|\tilde{\gamma}_{\mathrm{InSAR}}|$ for HH and VV polarizations show small differences, mainly lying in a range of $0.6 - 0.8$. The Pauli-1 polarization has the highest $|\tilde{\gamma}_{\mathrm{InSAR}}|$ of $0.7 - 0.8$ with a narrow distribution; whereas the Pauli-2 polarization shows the lowest values with a wider spread of the coherence, which is mainly due to the lower signal-to-noise ratio (SNR). The observed interferometric decorrelation indicates the necessity to consider the volume scattering contributing to the InSAR decorrelation.

The InSAR DEM $h_{\mathrm{InSAR}}$ is derived for the four polarizations. Again, only the HH polarization is shown in Fig. 3(b) for conciseness. The comparison between $h_{\mathrm{InSAR}}$ for the four polarizations and the DMS DEM $h_{\mathrm{DMS}}$ along the flight track (the green dashed line in Fig. 2) is shown in Fig. 5, with a maximum elevation difference around $2\,\mathrm{m}$. The differences of $h_{\mathrm{InSAR}}$ across the four polarizations are illustrated in Fig. 6, where the height differences mostly lie in the range of $-0.5\,\mathrm{m}$ to $0.5\,\mathrm{m}$. The InSAR-derived heights from Pauli-1 and HH polarizations reveal the most similar values; while Pauli-2 and HH polarizations show the largest height difference, which is accordant with the wider spread distribution of InSAR coherence in Pauli-2 channel (see Fig. 4). The root-mean-square errors $RMSE$ between $h_{\mathrm{InSAR}}$ and $h_{\mathrm{DMS}}$ are averaged to be $\sim 1.10\,\mathrm{m}$ for the four polarizations, indicating that the penetration of electromagnetic waves into snow and ice should be properly considered and corrected for sea ice topographic retrieval, at least for the deformed thick ice with snow cover in this study.

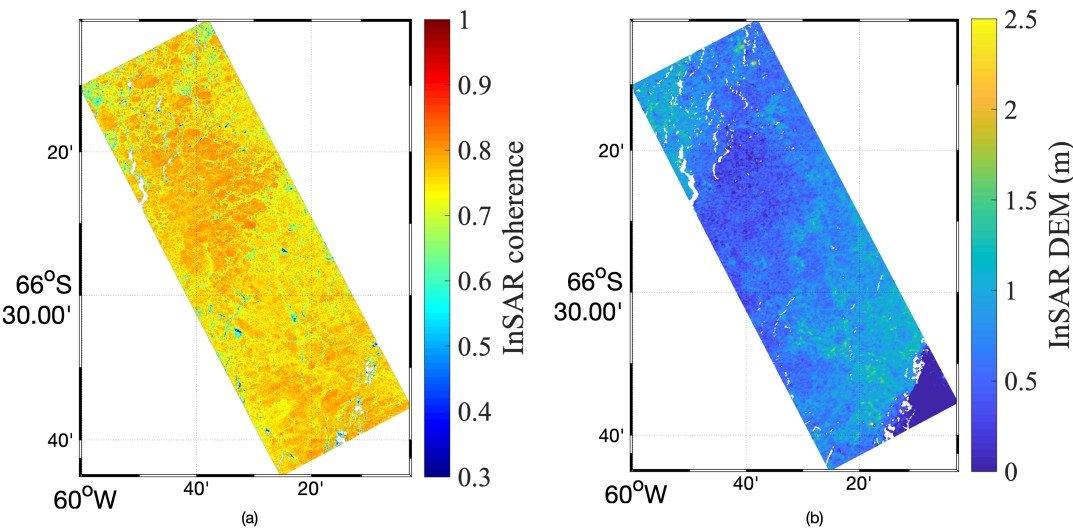

**Figure 3.** (a) Magnitude of the interferometric coherence $|\tilde{\gamma}_{\mathrm{InSAR}}|$ and (b) InSAR DEM $h_{\mathrm{InSAR}}$ for HH polarization.

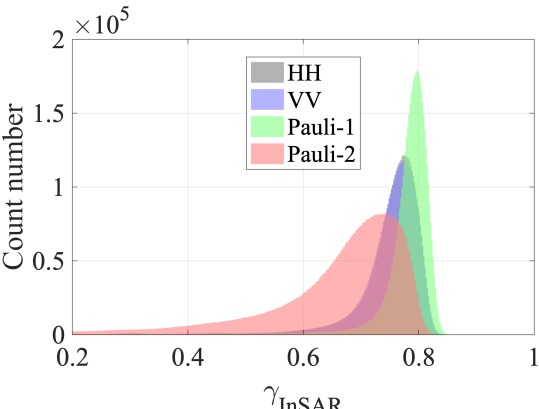

**Figure 4.** Magnitude of the interferometric coherence $|\tilde{\gamma}_{\mathrm{InSAR}}|$.

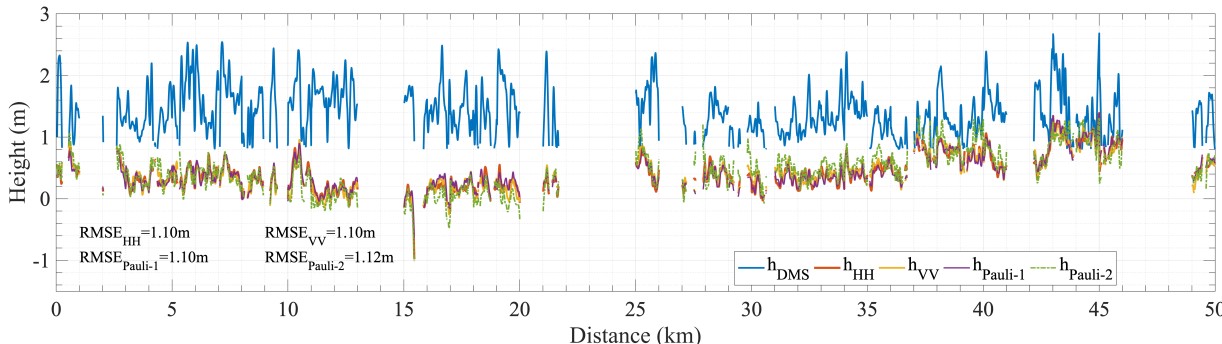

**Figure 5.** The InSAR-derived height profiles ($h_{\mathrm{InSAR}}$) and the DMS DEM ($h_{\mathrm{DMS}}$). Each profile represents the height along a $1 \times 5000$-pixel section at the center of the co-registered segment. The mis-coregistered and $h_{\mathrm{DMS}}$ below $0.8\,\mathrm{m}$ samples are excluded from the plots.

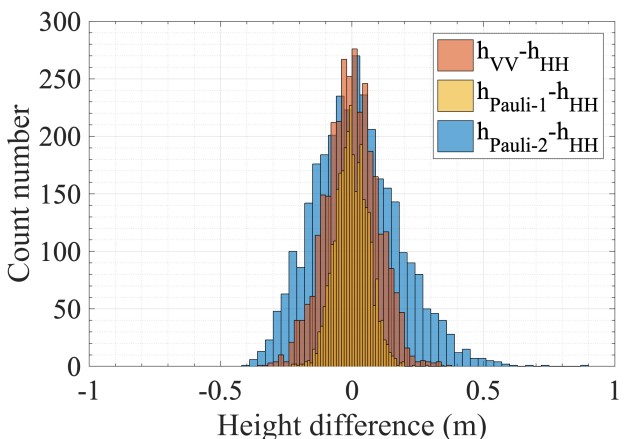

**Figure 6.** Height difference between the InSAR-derived height ($h_{\mathrm{InSAR}}$) in HH polarization and other three polarizations.

It can be summarized that both the interferometric decorrelation and the elevation difference between the InSAR DEM and the DMS DEM highlight the necessity for developing an appropriate method aimed at an accurate sea ice topography retrieval.

## 4   Model development

This section proposes a two-layer plus volume model to describe the interferometric coherence of sea ice and snow on top. Simulations are performed to analyze the model sensitivity and accuracy by varying parameter sets and baseline configurations.
In a separate step, the model is further simplified for the practical purpose of deriving sea ice topography with limited a priori parameter knowledge.

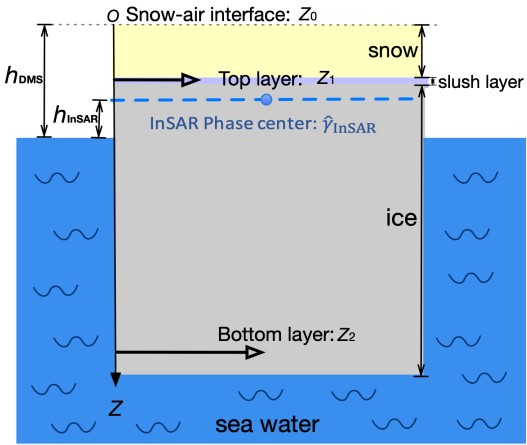

**Figure 7.** Schematic of the proposed two-layer plus volume model for sea ice.

### 4.1 Composite coherence model for sea ice

The sea ice volume has been modelled as a multi-layer structure in microwave remote sensing (Nghiem et al., 1990; Albert et al., 2012). We propose a two-layer plus volume model considering snow cover, ice volume, and seawater, illustrated in Fig.

7, behaving as a mix of surface and volume scattering under radar illumination. The uppermost surface (i.e., snow-air interface) is located at $z_0$.

Surface scattering is considered to originate mainly from two interfaces, named as the top layer and the bottom layer, respectively. The top layer located at $z_1$ is the snow-ice interface, which can induce significant surface scattering due to a slush layer with high permittivity (Hallikainen and Winebrenner, 1992; Maksym and Jeffries, 2000). This slush layer is widespread

on the Antarctic sea ice, and increases the radar backscattering as well as limits the signal penetration compared to a smooth and dry snow-ice interface. As long as the slush layer has a small vertical extent, it is irrelevant for the Pol-InSAR scattering structure model, whether the top layer represents the snow-ice interface, the snow-slush interface or both. The position of the bottom layer ($z_2$) could be somewhere inside the ice volume or at the ice-water interface. The vertical distributions of the top and bottom surface can be modelled as two Dirac delta functions at the specific layer position. An additional parameter, the

layer-to-volume scattering ratio, accounts for the (relative) scattering from these interfaces, depending e.g. on roughness and dielectric contrast (Fischer et al., 2018). Hence, the surface scattering component in the context of interferometry is modelled as (Fischer et al., 2018)

$$\tilde{\gamma}_{\text{Layer}} = e^{i\phi_0} \frac{m_1 e^{i\kappa_{z\_\text{vol}}z_1} + m_2 e^{i\kappa_{z\_\text{vol}}z_2}}{m_1 + m_2} \tag{14}$$

where $m_1$ and $m_2$ are the layer-to-volume ratio of the top and bottom layer, respectively.

The volume scattering is attributed to the constituents in the snow (from $z_0$ to $z_1$) and ice (from $z_1$ to $z_2$) volumes (Hallikainen and Winebrenner, 1992). Both volumes are assumed to be uniform volumes, which means that the scattering coefficient per unit volume and the extinction coefficient $\sigma$ have no spatial variation. In this case, the vertical structure function $\sigma_v(z)$ becomes

exponential. The $\tilde{\gamma}_v$ for a uniform volume model can be formulated as (Papathanassiou and Cloude, 2001)

$$\tilde{\gamma}_v(\sigma, D) = \frac{\int_0^D e^{\frac{2\sigma z}{\cos \theta_r}} e^{i\kappa_{z\_vol} z} dz}{\int_0^D e^{\frac{2\sigma z}{\cos \theta_r}} dz} \tag{15}$$

where $\theta_r$ is the incidence angle in the volume, $\sigma$ is the constant extinction coefficient, and D is the volume thickness. The corresponding volume coherences can be derived according to Eq. (15), denoted as $\tilde{\gamma}_v(\sigma_1, z_{01})$ and $\tilde{\gamma}_v(\sigma_2, z_{12})$ for the snow and ice volumes, respectively, where $z_{01} = z_0 - z_1$ is the thickness of the snow volume and $z_{12} = z_1 - z_2$ is the thickness of the ice volume.

For the overall two-layer plus volume model, the interferometric coherence can be given as a combination of volume and
surface effects which are described by Eq. (14) and Eq. (15), respectively. As represented in Fig. 7, if we set $z_0$ as the origin of the coordinate and thus to be 0, $z_1$ and $z_2$ being the position of two layers with negative values, the composite interferometric coherence is postulated to be

$$
\begin{aligned}
\tilde{\gamma}_{\text{InSAR}'} \\
&= e^{i\phi_0} \frac{\alpha \tilde{\gamma}_v(\sigma_1, z_{01}) + e^{i\phi_1}(1-\alpha)\tilde{\gamma}_v(\sigma_2, z_{12}) + m_1 e^{i\phi_1} + m_2 e^{i\phi_2}}{1 + m_1 + m_2} \\
&= e^{i\phi_0} \tilde{\gamma}_{\text{mod\_T}}(\sigma_1, \sigma_2, \alpha, m_1, m_2, z_1, z_2)
\end{aligned}
\tag{16}
$$

where $\phi_1 = \kappa_{z\_vol} z_1$, $\phi_2 = \kappa_{z\_vol} z_2$, $\sigma_1$ and $\sigma_2$ are extinction coefficients of snow and ice volume, respectively, in the unit of
Np/m. Note that $\sigma(\text{dB/m}) = \frac{10}{\ln 10}\sigma(\text{Np/m}) = 4.343\,\sigma(\text{Np/m})$. The volume coherences of snow $\tilde{\gamma}_v(\sigma_1, z_{01})$ and ice $\tilde{\gamma}_v(\sigma_2, z_{12})$ can be obtained according to Eq. (15). Weight parameter $\alpha$ ($\in [0,1]$) represents the proportion of the snow volume scattering in the combined (snow and ice) volume scattering.

## 4.2   Analysis of model sensitivity

The prediction of $\tilde{\gamma}_{\text{mod\_T}}$ from the proposed two-layer plus volume model by Eq. (16) requires seven parameters: extinction
coefficients $\sigma_1$, $\sigma_2$ (dB/m), layer-to-volume ratio $m_1$, $m_2$, layer-position $z_1$, $z_2$ (m), and weight parameter $\alpha$. However, it is impossible to estimate all unknowns based on only two observables: the phase and magnitude of $\tilde{\gamma}_{\text{InSAR}'}$. Therefore, the necessary simplification in terms of model parameters should be considered for the model inversion, which is addressed in Sections 4.4 and 5.

The simulation in this section aims at reducing the number of unknowns of the model by selecting the parameters which
induce minor variance of $\tilde{\gamma}_{\text{mod\_T}}$. The sensitivity of $\tilde{\gamma}_{\text{mod\_T}}$ to various parameters is presented in Fig. 8, where the radius and angular rotation corresponds to the coherence magnitude and phase, respectively. The phase can be translated to height via Eq. (4). It shows the complex $\tilde{\gamma}_{\text{mod\_T}}$ as a function of the ice-volume height ($h_v = z_1 - z_2$, ranging from 0 to $-5$ m) by varying only one parameter and keeping the others constant. The $\kappa_z$ for the studied image is $0.28\,\text{rad/m}$ following Eq. (9).

Figure 8(a) and Fig. 8(b) show the loci obtained for $\alpha = 0.5$, $m_1 = 0.5$, $m_2 = 0.5$, $z_1 = -0.15$ m with increasing $\sigma_1$ and
$\sigma_2$, respectively. The snow extinction coefficient $\sigma_1$ depends on the electromagnetic wave's frequency, snow temperature, volumetric water content, snow density, and the shape of the ice particles. At $10\,\text{GHz}$ frequency, the snow extinction coefficient

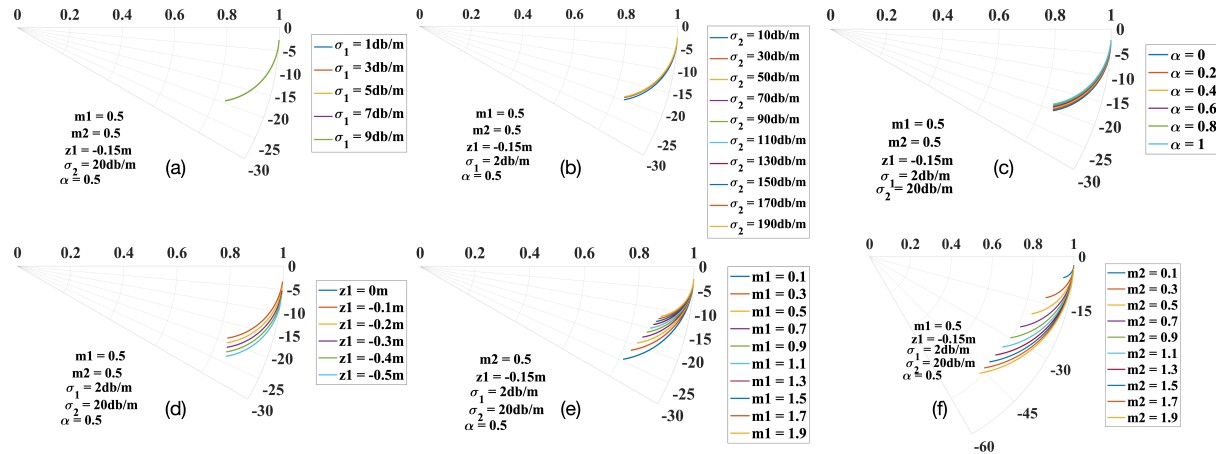

**Figure 8.** Simulation of $\tilde{\gamma}_{\mathrm{mod\_T}}$ for the proposed model by varying (a) snow extinction $\sigma_1$, (b) ice extinction $\sigma_2$, (c) weight parameter $\alpha$, (d) top-layer position $z_1$, (e) top-layer layer-to-volume ratio $m_1$, and (f) bottom-layer layer-to-volume ratio $m_2$.

was measured to be $1-10\,\mathrm{db/m}$ (Haykin et al., 1994). This range of values is considered for $\sigma_1$ in Fig. 8(a). Sea ice consists of pure ice, brine inclusions, and air bubbles. The properties and geometry of these constituents together with the environmental conditions influence the sea ice extinction coefficient $\sigma_2$. Experimental values of sea ice extinction coefficient at $10\,\mathrm{GHz}$ are

given in (Hallikainen and Winebrenner, 1992), ranging from 10 to $200\,\mathrm{dB/m}$ covering different types of sea ice (cf. Fig. 8(b)). Figure 8(c) shows the loci obtained for $m_1 = 0.5$, $m_2 = 0.5$, $\sigma_1 = 2\,\mathrm{dB/m}$, and $\sigma_2 = 20\,\mathrm{dB/m}$ with $\alpha$ varying from 0 to 1.

As illustrated in Fig. 8(a)-(c), the simulated $\tilde{\gamma}_{\mathrm{mod\_T}}$ is marginally sensitive to the variance of $\sigma_1$, $\sigma_2$, and $\alpha$, suggesting the possibility to fix them as constant to reduce the model complexity. This marginal sensitivity of both volume contributions can be understood by looking at their individual contributions to Eq. (16). The complex coherence of the snow volume $\tilde{\gamma}_{\mathrm{v}}(\sigma_1, z_{01})$

can be calculated by Eq. (15) with thickness $z_{01} = 15\,\mathrm{cm}$, and its magnitude and phase can be denoted as $|\tilde{\gamma}_{\mathrm{v}}(\sigma_1, z_{01})|$ and $\angle\tilde{\gamma}_{\mathrm{v}}(\sigma_1, z_{01})$, respectively. Then, the phase center location of the snow volume alone can be calculated by Eq. (10). Across the range of $\sigma_1$ (i.e., $1-10\,\mathrm{db/m}$), the snow volume has an individual coherence magnitude (i.e., $|\tilde{\gamma}_{\mathrm{v}}(\sigma_1, z_{01})|$) close to unity and phase center height varying from $-6$ to $-7\,\mathrm{cm}$. Therefore, it acts almost as a constant contribution, which is additionally located close to the Dirac delta of the top layer $m_1 e^{i\phi_1}$ in the complex unit circle. Similarly, the ice volume $\tilde{\gamma}_{\mathrm{v}}(\sigma_2, z_{12})$ has an

individual coherence magnitude of almost unity and a phase center height between $-15$ to $-33\,\mathrm{cm}$ for the investigated range of ice extinction coefficients. Therefore, its effects have a limited variablity and are also more or less aligned with the top layer. These observations are only valid for the investigated $\kappa_{z\_\mathrm{vol}}$ and might differ for baselines larger than usual for TanDEM-X.

Figure 8(d) shows the loci obtained for $\alpha = 0.5$, $m_1 = 0.5$, $m_2 = 0.5$, $\sigma_1 = 2\,\mathrm{dB/m}$, and $\sigma_2 = 20\,\mathrm{dB/m}$ with snow depth $z_1$ varying from 0 to $-0.5\,\mathrm{m}$. The influence of the snow depth on $\tilde{\gamma}_{\mathrm{mod\_T}}$ is not negligible. One way to address this is by using

a priori knowledge from external sources. With fixed values of $\alpha$, $\sigma_1$, $\sigma_2$, and $z_1$, the loci with different $m_1$ and $m_2$ values are illustrated in Fig. 8(e) and Fig. 8(f), respectively. The simulated $\tilde{\gamma}_{\mathrm{mod\_T}}$ shows sensitivity to the layer-to-volume ratio of the top layer in Fig. 8(e). However, estimations of $m_1$ from observations are challenging due to the insufficient measurements of

the sea ice conditions over the study area. Therefore, $m_1$ is approximated to be a constant value in the proposed theoretical model. A simplified model, which avoids estimating $m_1$, will be introduced in Section 4.4. For the layer-to-volume ratio of the bottom layer, $m_2$ induces significant variance to $\tilde{\gamma}_{\mathrm{mod\_T}}$, indicating $m_2$ as the most deterministic parameter which should be properly estimated to ensure the accuracy of model inversion.

## 4.3 Assessment of model accuracy

The observed interferometric coherence $\tilde{\gamma}_{\mathrm{InSAR}'}$ can be biased by a residual non-volumetric decorrelation component $\gamma_{\mathrm{res}}$, even after accounting for $\gamma_{\mathrm{s}}$ and $\gamma_{\mathrm{SNR}}$ by means of Eq. (6). $\gamma_{\mathrm{res}}$ can induce further errors when performing the model inversion for height estimation (Kugler et al., 2015). Therefore, based on Eq. (16), this potential error term is considered as

$$\tilde{\gamma}_{\mathrm{InSAR}'} = e^{i\phi_0} \tilde{\gamma}_{\mathrm{mod\_T}} \gamma_{\mathrm{res}} \tag{17}$$

In the inversion, the height uncertainty depends on the magnitude of $\tilde{\gamma}_{\mathrm{InSAR}'}$ (i.e., $|\tilde{\gamma}_{\mathrm{InSAR}'}|$) and the InSAR baseline configuration (i.e., $\kappa_{z\_\mathrm{vol}}$) . In this subsection, a Monte-Carlo simulation is performed to assess the height uncertainty with various $|\tilde{\gamma}_{\mathrm{InSAR}'}|$ and $\kappa_{z\_\mathrm{vol}}$ values. The estimation of coherence itself has a variance due to its stochastic nature and the number of looks (i.e. the size of the coherence estimation window). In other words, the estimation accuracy of $\tilde{\gamma}_{\mathrm{InSAR}'}$ depends on the standard deviation of its magnitude and phase which are defined by the statistical distribution and the number of looks for multilook SAR data (Kugler et al., 2015; Touzi and Lopes, 1996; Lopes et al., 1992). The statistical distribution of coherence magnitude and phase can be given as follows.

The probability density function (pdf) of coherence magnitude $\gamma$ is obtained as (Touzi and Lopes, 1996)

$$P(\gamma) = 2(N-1)(1-D^2)^N \gamma (1-\gamma^2)^{N-2} F(N,N;1,D^2\gamma^2) \tag{18}$$

where $N$ is the number of looks, $F$ is a hypergeometric function, and $D$ is the expectation value of coherence level.

The pdf of sample coherence phase $\phi$ follows (Lopes et al., 1992)

$$P(\phi) = \frac{(1-D^2)^N}{2\pi} [{}_3F_2(1,N,N;0.5,N;D^2\cos^2(\phi-\beta)) + k'D\cos(\phi-\beta)) \times_3 F_2(1.5,N+0.5,N+0.5; \\ 1.5,N+0.5;D^2\cos^2(\phi-\beta))] \tag{19}$$

where $k' = \Gamma(0.5)\Gamma(N+0.5)/\Gamma(N)$, ${}_3F_2$ is a generalized hypergeometric function, and $\beta$ is the mean phase difference.

The simulation is a four-step procedure. First, the complex value of $\tilde{\gamma}_{\mathrm{mod\_T}}$ is calculated for the designed two-layer plus volume model with specific parameters $(\sigma_1, \sigma_2, \alpha, m_1, m_2, z_1, z_2)$ and a given $\kappa_{z\_\mathrm{vol}}$. The surface phase $\phi_0$ is assumed to be 0. Second, $\tilde{\gamma}_{\mathrm{sim}}$ is obtained by $\tilde{\gamma}_{\mathrm{sim}} = \tilde{\gamma}_{\mathrm{mod\_T}} \gamma_{\mathrm{res}}$ with $\gamma_{\mathrm{res}} = 0.98$ (according to (Kugler et al., 2015)). Next, a set of $(N_s = 10000)$ $\tilde{\gamma}_{\mathrm{sim}_i}$ complex samples is generated via Eq. (18) (for magnitude) and Eq. (19) (for phase) with $D = |\tilde{\gamma}_{\mathrm{sim}}|$ and $\beta = \angle\tilde{\gamma}_{\mathrm{sim}}$. Finally, for each simulated $\tilde{\gamma}_{\mathrm{sim}_i}$, the volume height $h_{vi}$ is estimated by the inversion of Eq. 16 with the specific parameters and $\kappa_{z\_\mathrm{vol}}$ of the simulation and compared with the input $h_v = z_1 - z_2$ in the first step. The bias $B_{\Delta_h} = |\mathbb{E}(h_{vi}) - h_v|$ and the standard deviation $\sigma_{\Delta_h} = \sqrt{\mathbb{E}[(h_{vi} - h_v)^2]}$ are calculated to quantify the estimation accuracy of the model.

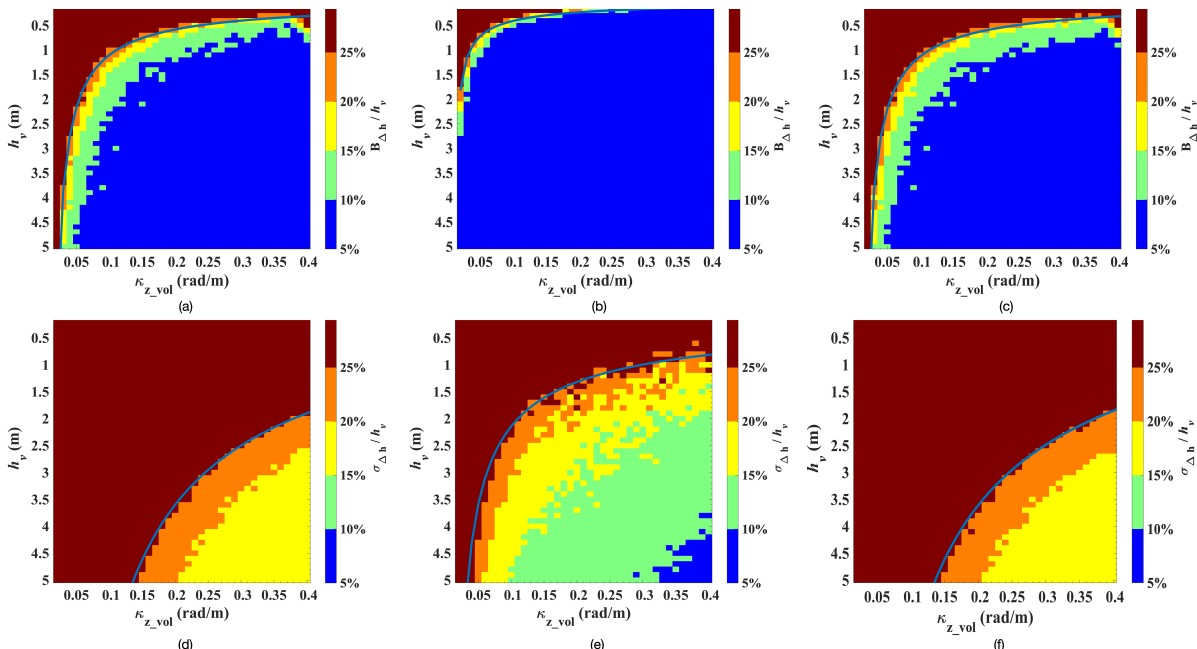

**Figure 9.** (a)-(c) Relative Bias $B_{\delta_h}/h_v$ and (d)-(f) relative standard deviation $\sigma_{\delta_h}/h_v$ of the obtained ice volume $h_v$ for the proposed model. Fixed parameters: $\sigma_1 = 2\,\mathrm{dB/m}$, $m_1 = 0.3$, $z_1 = -0.15\,\mathrm{m}$, $\alpha = 0.5$, variable parameters: (a) and (d) $\sigma_2 = 20\,\mathrm{dB/m}$, $m_2 = 0.5$, (b) and (e) $\sigma_2 = 20\,\mathrm{dB/m}$, $m_2 = 2$, (c) and (f) $\sigma_2 = 100\,\mathrm{dB/m}$, $m_2 = 0.5$. Blue line indicates the threshold of $B_{\delta_h}/h_v = 25\%$ or $\sigma_{\delta_h}/h_v = 25\%$.

By fixing the model parameters $(\sigma_1, \sigma_2, \alpha, m_1, m_2, z_1)$ and varying $z_2$, $B_{\Delta_h}$ and $\sigma_{\Delta_h}$ are functions of ice volume height $h_v = z_1 - z_2$ and volume-corrected vertical wavenumber $\kappa_{z\_\mathrm{vol}}$. The simulation procedure is performed for $h_v$ ranging from 0 to $5\,\mathrm{m}$ and $\kappa_{z\_\mathrm{vol}}$ ranging from 0.02 to $0.4\,\mathrm{rad/m}$. The number of looks $N$ is set to be the same value as the experimental data. The bias and the standard deviation relative to volume height are shown in Fig. 9. The plots illustrate that for a specific baseline geometry with a given $\kappa_{z\_\mathrm{vol}}$, the model performance is superior for a certain range of volume height, shown in the blue curve, indicating the 25% threshold, in Fig. 9. For volume heights lower than at the blue curve, the bias and variance are larger than 25%, leading to a lower precision of model inversion. With different $\sigma_2$ values, there are no obvious distinctions of $B_{\Delta_h}/h_v$ between the Fig. 9(a) and 9(c), as well as $\sigma_{\Delta_h}/h_v$ between Fig. 9(d) and 9(f), respectively. It suggests that the model accuracy is marginally sensitive to ice extinction coefficient $\sigma_2$. The layer-to-volume ratio of the bottom layer $m_2$ plays a key role in model accuracy. From Fig. 9(b) and Fig. 9(e), the $B_{\Delta_h}/h_v$ and $\sigma_{\Delta_h}/h_v$ are smaller than those from Fig. 9(a) and Fig. 9(d) due to the larger $m_2$, indicating the higher model accuracy in this case. The 25%-error threshold provides a criteria for selecting the best baseline geometry for the application. For example, with a specific parameter set as Fig. 9(b) and Fig. 9(e), the $\kappa_{z\_\mathrm{vol}}$ needs to be larger than $0.40\,\mathrm{rad/m}$ to ensure an effective inversion for ice-volume thickness less than $0.85\,\mathrm{m}$. Since the $\kappa_{z\_\mathrm{vol}}$ of the studied SAR image is $0.28\,\mathrm{rad/m}$, in order to achieve an 25%-error accuracy, the ice volume needs to be thicker than a certain value depending on $m_2$. This certain value ranges from 1.1 to $2.7\,\mathrm{m}$ for $m_2$ being $0.5 - 2$, see Fig. 9(d) and (e). The above assessment also indicates the potential of applying the proposed model to achieve a more accurate result using a larger

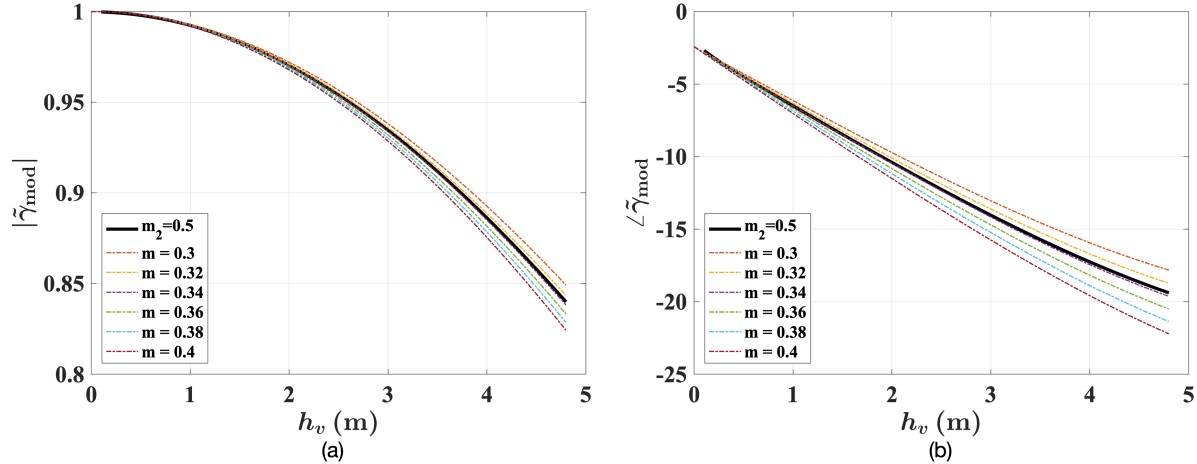

**Figure 10.** Comparison of complex coherence $\tilde{\gamma}_{\mathrm{mod}}$ from the theoretical model ($\tilde{\gamma}_{\mathrm{mod\_T}}$, thick black line) and the simplified model ($\tilde{\gamma}_{\mathrm{mod\_S}}$, colored lines). (a) Magnitude of the modelled coherence. (b) Phase ($°$) of the modelled coherence.

baseline configuration. Note that the baseline should not be too large since it results in stronger interferometric decorrelation which contaminates the topographic information.

## 4.4 Model simplification

The proposed theoretical model $\tilde{\gamma}_{\mathrm{mod\_T}}$ given in Eq. (16) contains seven parameters: $\sigma_1, \sigma_2, \alpha, m_1, m_2, z_1, z_2$, requiring necessary a priori knowledge of the test site. However, such a priori knowledge is scarce due to the sparse ground measurement of the Antarctic sea ice, therefore impeding the practical application of the proposed model. As described in Section 4.2, the contributions of the snow volume $\tilde{\gamma}_{\mathrm{v}}(\sigma_1, z_{01})$ and the ice volume $\tilde{\gamma}_{\mathrm{v}}(\sigma_2, z_{12})$ to the theoretical model $\tilde{\gamma}_{\mathrm{mod\_T}}$ show a limited sensitivity for the TanDEM-X acquisition geometry within the investigated range of extinction coefficients $\sigma_1$ and $\sigma_2$. Additionally, their individual coherence loci are located close to the Dirac delta of the top layer $m_1 e^{i\phi_1}$ in the unit circle. Therefore, the theoretical model can be approximated by merging the contributions of the snow volume, the ice volume, and the top layer into one Dirac delta. This simplified model can be given as

$$
\begin{aligned}
&\tilde{\gamma}_{\mathrm{InSAR}'} \\
&= e^{i\phi_0} \frac{\alpha \tilde{\gamma}_{\mathrm{v}}(\sigma_1, z_{01}) + e^{i\phi_1}(1-\alpha)\tilde{\gamma}_{\mathrm{v}}(\sigma_2, z_{12}) + m_1 e^{i\phi_1} + m_2 e^{i\phi_2}}{1 + m_1 + m_2} \\
&\approx e^{i\phi_0} \frac{1 \cdot e^{i\phi_1} + m \cdot e^{i\phi_2}}{1 + m} \\
&= e^{i\phi_0} \tilde{\gamma}_{\mathrm{mod\_S}}(m, z_1, z_2)
\end{aligned}
\tag{20}
$$

where $\phi_1 = \kappa_{z\_\mathrm{vol}} z_1$, $\phi_2 = \kappa_{z\_\mathrm{vol}} z_2$, $z_1$ and $z_2$ are the position of the top layer and the bottom layer, respectively, and $m$ is the layer-to-layer ratio.

Compared to the theoretical model in Eq. (16), the simplified model in Eq. (20) only has three parameters, remarkably improving the applicability in practice. Figure 10 illustrates the simulations of $\tilde{\gamma}_{\mathrm{mod\_T}}$ and $\tilde{\gamma}_{\mathrm{mod\_S}}$ according to Eq. (16) and Eq. (20), respectively. For $\tilde{\gamma}_{\mathrm{mod\_T}}$, the parameters are set as $\sigma_1 = 2\,\mathrm{dB/m}$, $\sigma_2 = 20\,\mathrm{dB/m}$, $z_1 = -0.15\,\mathrm{m}$, $m_1 = m_2 = 0.5$. For $\tilde{\gamma}_{\mathrm{mod\_S}}$, $z_1$ is also set to be $-0.15\,\mathrm{m}$, and $m$ varies from $0.3$ to $0.4$. As we can see, for both coherence magnitude and phase, the simplified model can achieve comparable results to the theoretical model by assuming appropriate $m$ values.

## 5 Model inversion

In order to apply the simplified model and theoretical model to geophysical parameter retrieval, a methodology is developed for the model inversion. The objective is to estimate the topographic phase $\phi_0$ and thus generate the sea ice topographic height with snow depth for the whole SAR image. The model inversion includes three main steps, illustrated in Fig. 11. As explained in Section 2.1, in order to select the ice that is deformed and thick without seawater flooding, the samples with height above $0.8\,\mathrm{m}$, which are $83\%$ of the co-registered data set, are selected.

As shown in Step-1 in Fig. 11, the $h_{\mathrm{DMS}}$ is converted to phase $\phi_{\mathrm{DMS}}$ via Eq. (4) and used as a priori knowledge. For the simplified model, $z_1$ is set to be $-0.18\,\mathrm{m}$ according to the snow depth provided by the AMSR-E/AMSR2 Unified Level-3 Daily data set (Meier, W. N., T. Markus, and J. C. Comiso, 2018). The AMSR-E/AMSR2 data set provides snow depth over sea ice as five day running averages. For the studied area on the campaign date, the averaged snow depth was measured to be $18\,\mathrm{cm}$. For the theoretical model, as discussed in Section 4.2, since the simulated $\tilde{\gamma}_{\mathrm{mod\_T}}$ shows marginal sensitivity to the variance of snow layer extinction coefficient $\sigma_1$, ice layer extinction coefficient $\sigma_2$, and weight parameter $\alpha$, these three parameters are fixed to constants. For the snow-covered sea ice of the studied area, the $\sigma_1$ and $\sigma_2$ are assumed to be $2\,\mathrm{dB/m}$ and $20\,\mathrm{dB/m}$, respectively, referring to experimental values (Cox and Weeks, 1974; Hallikainen and Winebrenner, 1992). The snow depth $z_1$ is also set to $-0.18\,\mathrm{m}$. The layer-to-volume ratio of the top layer $m_1$ is set to $0.3$. With above specific parameters, the $m_2$ (also $m$ for the simplified model) values can be derived by the inversion of the proposed model according to Eq. (16) or Eq. (20) for the theoretical and the simplified model, respectively.

Next, since $m_2$ or $m$ is the most deterministic parameter in the respective models, the aim is to estimate $m_2$ or $m$ from the SAR observations (Step-2 in Fig. 11). In addition to interferometry, which provides topographic information, polarimetry reveals information on the scattering processes and is a useful tool to characterize sea ice properties. Among several polarimetric signatures, the co-polarization (coPol) coherence $\gamma_{\mathrm{coPol}}$ is a measurement of the degree of electromagnetic wave depolarization between HH and VV polarizations caused by both the rough surface scattering and the volume scattering (Kasilingam et al., 2001). $\gamma_{\mathrm{coPol}}$ was demonstrated to be a crucial signature in sea ice characterization (Kim et al., 2011; Wakabayashi et al., 2004; Huang and Hajnsek, 2021). $\gamma_{\mathrm{coPol}}$ can be calculated as (Lee and Pottier, 2009)

$$\tilde{\gamma}_{\mathrm{coPol}} = \gamma_{\mathrm{coPol}} \cdot e^{i\phi_{\mathrm{coPol}}} = \frac{< s_{\mathrm{VV}} s_{\mathrm{HH}}^* >}{\sqrt{< s_{\mathrm{VV}} s_{\mathrm{VV}}^* >< s_{\mathrm{HH}} s_{\mathrm{HH}}^* >}} \tag{21}$$

where $s_{\mathrm{HH}}$ and $s_{\mathrm{VV}}$ are single look complex images in HH and VV polarization, respectively. The symbol $< . >$ denotes an ensemble average. A $4 \times 12$ window in azimuth and slant range is applied to estimate $\gamma_{\mathrm{coPol}}$. It is found that $m_2$ (also $m$) is

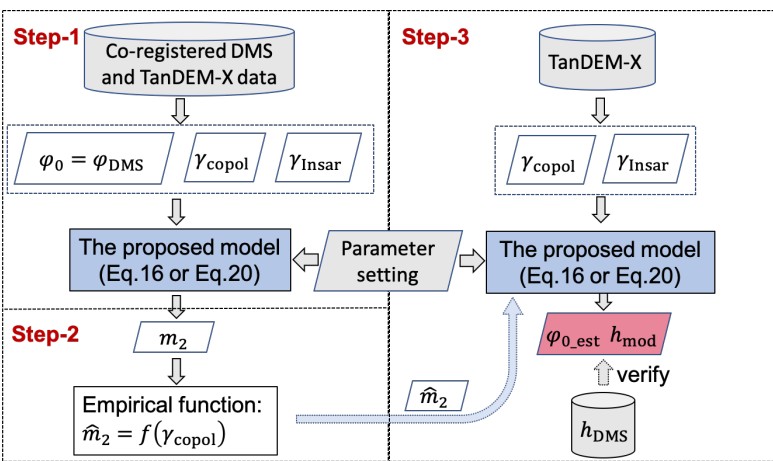

**Figure 11.** Flow chart of the proposed inversion method.

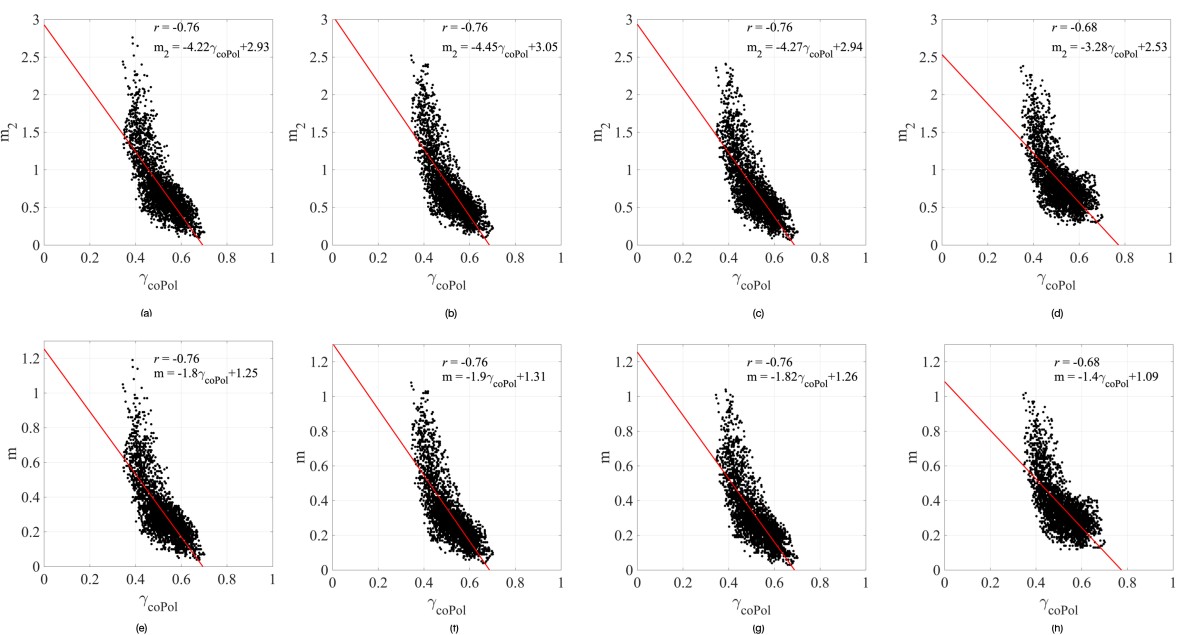

**Figure 12.** The relation between the bottom-layer layer-to-volume ratio $m_2$ (respectively $m$) and the coPol coherence $\gamma_{\mathrm{coPol}}$ for the theoretical model (first row) and the simplified model (second row). (a) and (d) HH polarization, (b) and (f) VV polarization, (c) and (g) Pauli-1: HH+VV polarization, and (d) and (h) Pauli-2: HH-VV polarization. Note the different y-axis scaling for $m_2$ and $m$.

inversely related to $\gamma_{\text{coPol}}$ for the four polarizations, shown in Fig. 12, enlightening us to derive an empirical function between the parameter $m_2$ (also $m$) and $\gamma_{\text{coPol}}$. As shown in Step-2 in Fig. 11, the linear functions for the different polarizations are derived by least-squares fitting and are detailed in Fig. 12. Note the almost identical correlation coefficients of the fitted linear function for the theoretical model (first row) and the simplified model (second row) in Fig. 12. This underlines that the theoretical model can be approximated by the simplified model. The parameters of the linear functions for $m_2$ and $m$ are of course different. Then, the fitted functions are applied to estimate $\hat{m}_2$ (also $\hat{m}$), which will be used as an input to perform the model inversion for the whole image, including the area without DMS measurements.

Finally, as shown in the Step-3 in Fig. 11, for the TanDEM-X data without a priori knowledge of DMS measurement, the $\gamma_{\text{coPol}}$ together with the derived linear function is utilized to estimate $\hat{m}_2$ (also $\hat{m}$) for each pixel. With the specific parameter setting, the estimated $\hat{m}_2$ (also $\hat{m}$), and the $\tilde{\gamma}_{\text{InSAR}'}$ from InSAR pairs, the topographic phase $\phi_{0\_\text{est}}$ can be retrieved by solving Eq. (16) or Eq. (20), and then converted to height $h_{\text{mod}}$ in meter via Eq. (4). The area overlaid by the DMS flight track is used to verify the model-inversion result quantitatively and visually, which will be illustrated in the next section.

## 6  Experimental results

In this section, the proposed two-layer plus volume model and its simplified version are inverted to estimate sea ice topography following the developed scheme. Note that the retrieved sea ice topography refers to the sea ice height including the snow depth above the local sea level. Both visual and quantitative analyses are given to evaluate the retrieval performance.

### 6.1  Retrieval performance of the simplified model

The sea ice topographic retrievals based on the simplified model (Fig. 11) are performed for the four polarizations (HH, VV, Pauli-1, and Pauli-2), respectively. Because of the marginal visual distinction among HH, VV, and Pauli-1 polarizations, only the Pauli-1 polarization result is presented for conciseness. The quantitative evaluation is given for the four polarizations.

The model-retrieved sea ice topography in Pauli-1 polarization is shown in Fig. 13. The strip between the grey lines is the area covered by the DMS DEM which is superimposed on the model-retrieved result ($h_{\text{mod\_S}}$) with the same colormap. In general, the retrieved height varies from $0.8$ to $3\,\text{m}$ across the whole image, showing a good agreement with the height range obtained by DMS measurements. Three areas are selected and enlarged for detailed analyses. Each area contains $40 \times 500$ pixels (corresponding to $400 \times 5000\,\text{m}$ area) along the range and azimuth direction, respectively. Figure 13(b) is the zoom-in of Area 1, where several sea ice areas are higher than $2.5\,\text{m}$. The model-retrieved height (outside the grey lines) shows good continuity with the DMS measurements (between the grey lines), indicating the effectiveness of the proposed method. Area 2 is mainly covered with ice lower than $2\,\text{m}$. The sea ice topographic retrieval of Area 2 is shown in Fig. 13(c), where the consistency between the model-retrieved Pol-InSAR DEM and the DMS DEM is again visually verified. Area 3 (Fig. 13(d)), including sea ice in the range of $2 - 2.5\,\text{m}$, shows the preservation of continuous sea ice features as well. Besides, the relative retrieval bias $\epsilon$, which can be calculated as $\epsilon = |h_{\text{mod\_S}} - h_{\text{DMS}}|/h_{\text{DMS}}$, is used to quantify the retrieval accuracy. In Fig. 13(e)-(g), $\epsilon$ over area 1-3 are below $25\%$ for most parts, whereas only a few parts, often near to the masked-out regions (transparent pixels), present

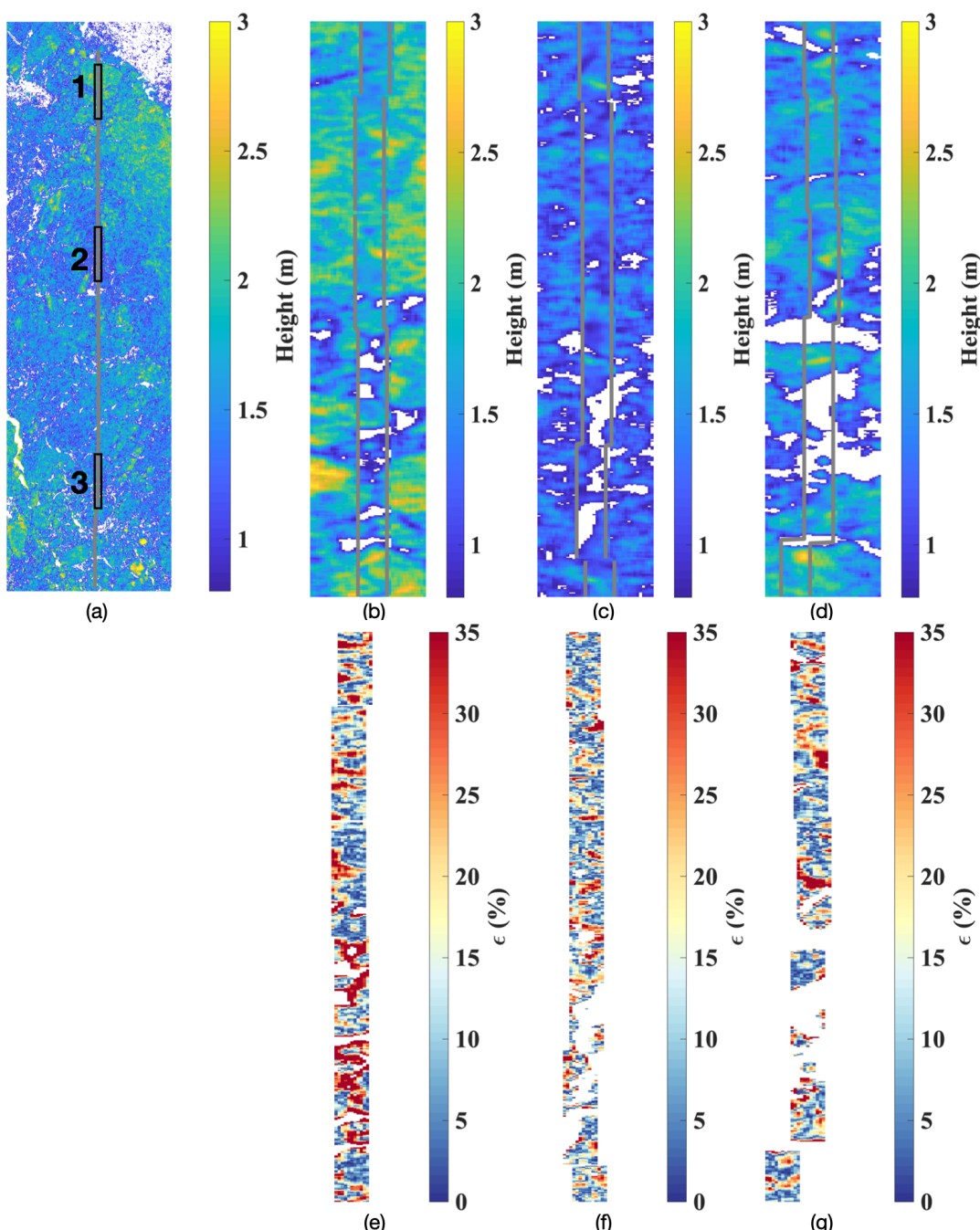

**Figure 13.** Sea ice topographic retrieval with the simplified model ($h_{\mathrm{mod\_S}}$). The transect from the DMS DEM is plotted between grey lines. Note that the heights below $0.8\,\mathrm{m}$ are set to be transparent. (a) The whole studied SAR image. (b)-(d) Zoom-in of Areas 1-3. (e)-(g) Relative retrieval bias $\epsilon = |h_{\mathrm{mod\_S}} - h_{\mathrm{DMS}}|/h_{\mathrm{DMS}}$ of area 1-3.

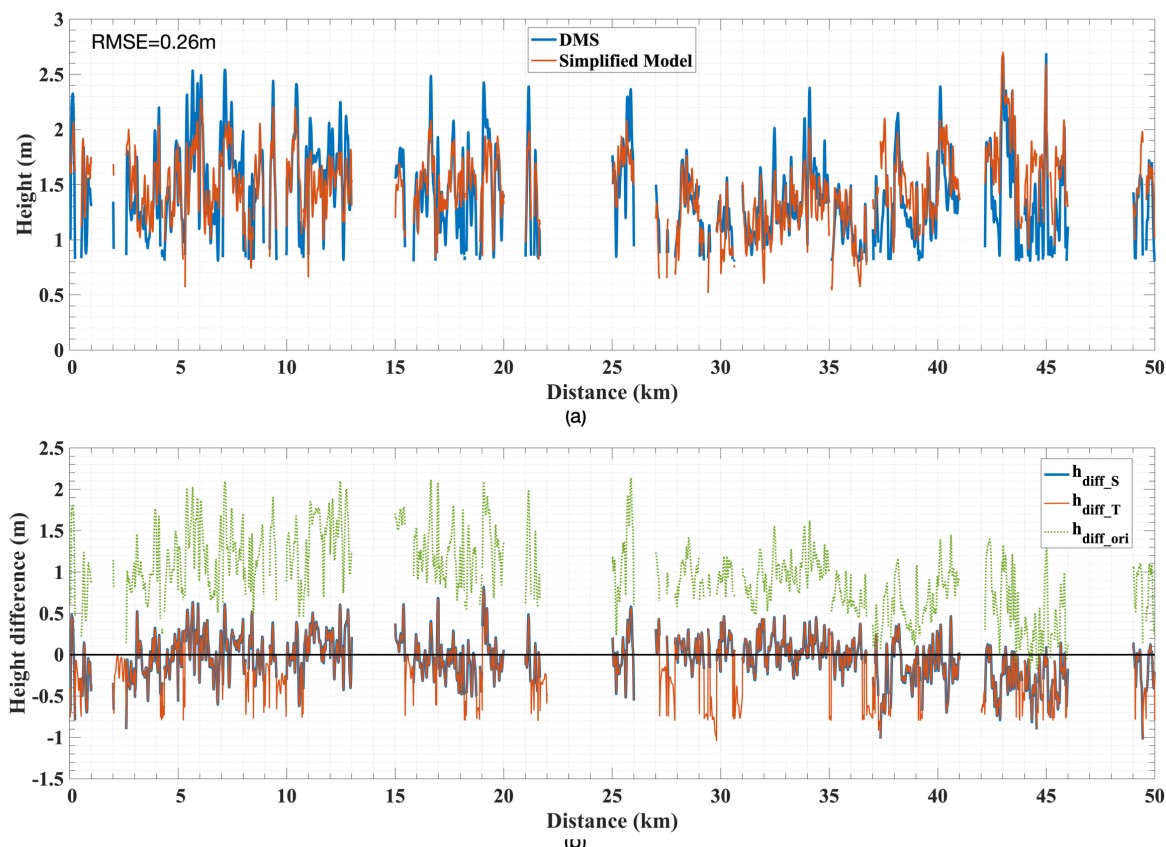

**Figure 14.** (a) Sea ice height profiles from DMS measurement (blue) and model inversion (red). Each profile represents the height along a $1 \times 5000$-pixel section at the center of the co-registered segment. (b) Height difference between the DMS measurement and the simplified model-derived height (blue), theoretical model-derived height (red), or original InSAR-derived height (green). The mis-coregistered and $h_{\mathrm{DMS}}$ below $0.8\,\mathrm{m}$ samples are excluded from the plots. $0\,\mathrm{m}$ is plotted in the black line as a reference.

higher $\epsilon$. Note that the masked-out regions refer to water and thinner ice areas with height less than $0.8\,\mathrm{m}$. The averaged $\epsilon$ over area 1-3 are about $19\%$, $14\%$, and $15\%$, respectively, and is $18\%$ for the whole image, achieving the theoretical $25\%$-error accuracy derived in Section 4.3. Therefore, from the comparison with photogrammetric measurements, the model-retrieved Pol-InSAR DEM demonstrates a good visual and quantitative agreement with the DMS DEM.

The comparison of the height profiles in Pauli-1 polarization along the DMS strip is shown in Fig. 14(a). The model-retrieved DEM $h_{\mathrm{Model}}$ and the DMS DEM $h_{\mathrm{DMS}}$ along the transect are plotted in the red and blue line, respectively. As observed from Fig. 14(a), the model-retrieved height has, in general, a good capture of the topographic variation. The height difference ($h_{\mathrm{diff\_S}}$) between the $h_{\mathrm{DMS}}$ and the simplified model-retrieved height is shown in Fig. 14(b) (blue). Compared with the larger height difference ($h_{\mathrm{diff\_ori}}$) between the $h_{\mathrm{DMS}}$ and the initial InSAR DEM in Fig. 14(b) (green), the surface-elevation bias

is properly compensated by using the simplified model. In Pauli-1 polarization, the $RMSE$ of the model-retrieved height

**Table 1.** The $RMSE$ of model-retrieved height $h_{\mathrm{Model}}$ for the four polarizations.

| Polarization | HH | VV | Pauli-1 | Pauli-2 |
|---|---|---|---|---|
| The InSAR method $RMSE$(m) | 1.1012 | 1.1050 | 1.0951 | 1.1183 |
| The simplified model $RMSE$(m) | 0.2757 | 0.2764 | 0.2637 | 0.4013 |
| The theoretical model $RMSE$(m) | 0.2745 | 0.2754 | 0.2631 | 0.3983 |

is $\sim 0.26\,\mathrm{m}$ relative to the DMS DEM. Compared to the original InSAR-derived height with $RMSE$ of $\sim 1.10\,\mathrm{m}$, Fig. 14 reveals the pronounced improvement of applying the proposed model to estimate sea ice topographic height, considering that the $RMSE$ in the DMS DEM is already $0.2\,\mathrm{m}$ to start with (Dotson and Arvesen., 2012, updated 2014).

Table 1 summarizes the performances between the retrieved height from the simplified model and $h_{\mathrm{DMS}}$ for the four polar-
izations. The $RMSE$ values across the HH, VV, and Pauli-1 polarizations are similar, ranging from $0.2637\,\mathrm{m}$ to $0.2764\,\mathrm{m}$. The larger $RMSE$ values in the Pauli-2 channel likely result from the lower SNR values. The polarization-independent performances in HH, VV, and Pauli-1 further reveal the inherent property of the studied sea ice to be a polarization-independent volume among co-pol channels in X-band radar frequencies.

## 6.2 Retrieval performance of the theoretical model

For the theoretical model which requires necessary a priori knowledge (e.g., snow density, ice salinity, air temperature) to determine the input parameters, the inversion performance depends on the study area and the sea ice structural characteristics. In this subsection, the inversion of the theoretical model is achieved by fixing parameters which have marginal effects on the model predictions and by estimating the layer-to-volume ratio of the bottom layer from a polarimetric signature: The coPol coherence. With the specific parameter set assumed in Section 5, the theoretical model is inverted according to the proposed
method (Fig. 11) to retrieve the sea ice topography. The height difference ($h_{\mathrm{diff\_T}}$) between the $h_{\mathrm{DMS}}$ and the theoretical model-retrieved height is shown in Fig. 14(b) (red), visualizing the similar performance of the theoretical and the simplified model. The results in Table 1 also show that the retrieval accuracy in terms of $RMSE$ of the theoretical model is almost identical to the simplified model. It demonstrates that the theoretical model can adequately correct the penetration bias of InSAR signals and achieve an effective sea ice height retrieval from dual-pol single-pass interferometric data, fulfilling the primary goal of
this study.

The comparable performance of the theoretical and simplified model convinces the effectiveness of employing the simplified model to achieve an accurate sea ice topographic retrieval. In the cases when the ground measurements are sparse, the simplified model requires only one parameter (i.e., snow depth over sea ice), significantly reducing the model complexity and improving the applicability in practice.

 ## 7 Discussion

### 7.1 Model complexity and observation space

The proposed two-layer plus volume model includes seven parameters with the assumption of their independence of polarization. In the case of a polarization-dependent volume, two more parameters are introduced for each volume and thus further complicate the inversion. In order to achieve the model inversion, one method is to develop a simplified model as presented in Section 6.1, which accurately approximates the behavior of the theoretical model and requires only the snow depth as an input parameter. Another method to achieve the theoretical model inversion is to increase the observation space to full polarization and/or multi-baseline configurations. Acquisitions of full-pol data improve the inversion capability over single-pol or dual-pol configurations. For instance, dual-baseline quad-pol data provide 12 independent observables and thereby offer an opportunity to theoretically invert a model with a maximum of 12 parameters. In Section 4.3, we have illustrated the theoretical performance of the proposed model with various baseline configurations and obtained a certain range which can ensure high inversion accuracy. It reveals the potential to establish an inversion scheme by combining observations from a range of different $\kappa_z$ (i.e., the vertical wavenumber in free space), where larger values of the effective perpendicular baseline $b_\perp$, corresponding to larger $\kappa_z$ values, are expected to improve height retrieval accuracy. With quad-pol and multi-baseline data acquired over sea ice in the future, developing a refined inversion scheme for more diverse scattering scenarios and thinner sea ice heights will be promising.

### 7.2 Influence of snow depth on experimental result

In Section 4.2, we demonstrated that the influence of snow depth on the simulated coherences is not negligible, and stated that external data of snow measurements should be used in the model. In this study, snow depth is assumed to be invariant across the scene due to the limited spatial resolution of available snow measurements. Therefore, a constant value of $z_1 = -0.18$ is used in the retrieval. Actually, snow on sea ice undergoes temporal- and spatial-variant processes and is strongly coupled with atmospheric, oceanic, and ice conditions. Thus, a single value is not representative of the actual spatial snow depth distribution. In order to assess the impact of the snow depth on the experimental results, we perform the whole inversion scheme with various inputs of $z_1$. During September and November, the snow depth on Antarctic sea ice is reported to be maximum $\sim 1\,\mathrm{m}$ and mainly $0 - 0.8\,\mathrm{m}$ (Webster et al., 2018). Therefore, $z_1$ values ranging from $-0.05\,\mathrm{m}$ to $-0.75\,\mathrm{m}$ are selected. For each pixel, we retrieve heights using this range of $z_1$ values, shown as the yellow area in Fig. 15. $\Delta h_{\mathrm{mod\_S}}$ is defined as the difference between the maximum and the minimum retrieved height of every pixel. The distribution of $\Delta h_{\mathrm{mod\_S}}$ along the transect is presented in Fig. 16, where $\Delta h_{\mathrm{mod\_S}}$ has a range of $0.07 - 1.09\,\mathrm{m}$ with an average of $0.31\,\mathrm{m}$, indicating the fluctuation of model-retrieved height by using different snow depths. This analysis with various snow depth assumptions can help to constrain possible model-retrieved topographies, and $\Delta h_{\mathrm{mod\_S}}$ can be a quantitative indicator for the uncertainty of the retrieved height in the absence of high-resolution snow depth data.

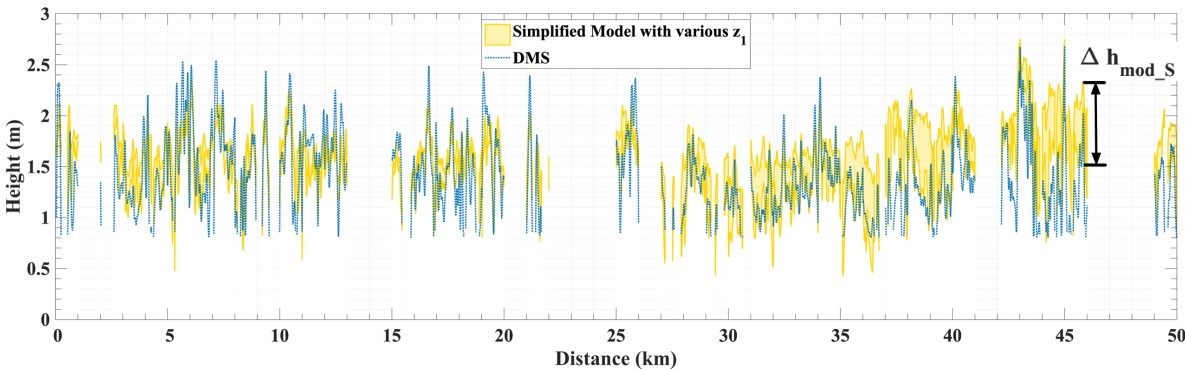

**Figure 15.** Yellow area: sea ice height profiles from the simplified model in Pauli-1 polarization with $z_1$ from $-0.05\,\text{m}$ to $-0.75\,\text{m}$. Blue dash line: sea ice height profiles from DMS measurement. Each profile represents the height along a $1 \times 5000$-pixel section at the center of the co-registered segment. The mis-coregistered and $h_{\text{DMS}}$ below $0.8\,\text{m}$ samples are excluded from the plot.

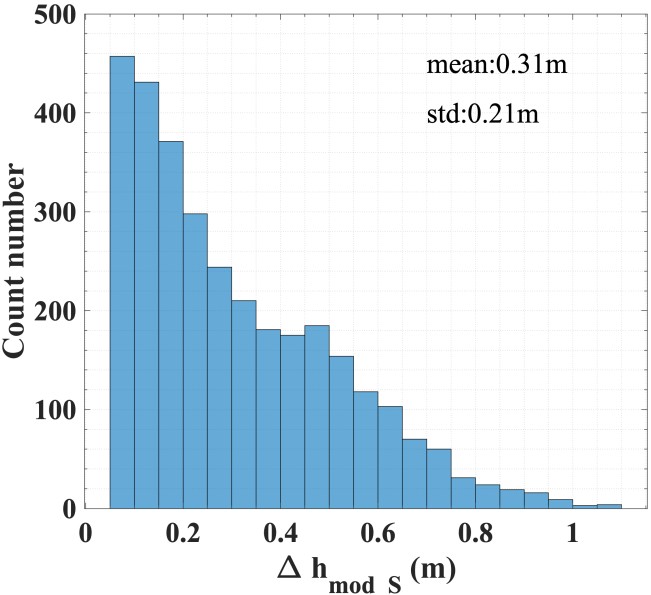

**Figure 16.** The distribution of $\Delta h_{\text{mod\_S}}$ along the transect.

## 7.3 Model extension to other ice conditions

The proposed model was proven to be effective in a specific area covered by thick and deformed ice with snow cover in the Western Weddell Sea. The extension of the proposed model to other ice types under different environmental conditions needs further research and suitable data.

In order to apply the model over younger and thinner sea ice, the first challenge is the severe misregistration between SAR images and reference measurements due to the stronger dynamics of thinner sea ice. Reduced SAR backscattering intensity corresponding to thinner and smoother sea ice further complicates the data co-registration. Besides, the achievable height sensitivity for thin ice is also a major limitation of InSAR/Pol-InSAR derived sea ice DEMs with current SAR systems. In this study, the proposed method can achieve sea ice topographic retrieval with an $RMSE$ of $0.26\,\mathrm{m}$ for thick and deformed

ice; however, this accuracy is insufficient for thinner ice whose height above sea level is only tens-of-centimeters or even less. Moreover, an additional volume, i.e., snow ice formed by flooding, should be considered when extending the proposed model to a thinner ice area. The presence of snow ice is a challenging retrieval scenario, not only for Pol-InSAR applications. The zero freeboard of snow ice, or the negative freeboard of ice in case of a slush cover, can cause significant difficulties when converting radar or laser altimeter measurements of snow or ice freeboard to ice thickness because the isostatic equations

fail in such a scenario. Past studies showed that snow ice contributes an average of $8\%$ of the total volume in the Weddell Sea (Lange et al., 1990). A greater amount of snow ice, which accounts for $12-36\%$ of the total mass, was reported in the Ross, Amundsen, and Bellingshausen Seas (Jeffries et al., 2001). Although the snow ice has a higher salinity than the ice below, there could be still some penetration into the ice volume below. A three-layer model, which includes snow, snow ice, and ice layers, could be feasible to correct the InSAR phase center and retrieve surface height for thinner ice in the Antarctic

(as illustrated in Fig. 1 (a)). However, the three-layer model involving more parameters than the proposed theoretical model can only be inverted by increasing the observation space to full polarization and multi-baseline configurations. In the case of single-baseline or dual-polarization configurations where observables are limited, an intelligent model with fewer parameters is worthy of investigation in the future.

    In order to assess the transferability of the proposed model to the Arctic regions, further validations, including co-registered

SAR images and topographic reference (e.g., optical/LIDAR measurements), are needed considering the significant difference of ice and snow properties between the Arctic and Antarctic. Ancillary measurements (e.g., snow depth, temperature, ice and snow salinity) at a wide ranger of ice conditions in both polar regions are crucial to understand the properties of various typologies of ice, and therefore are valuable for extending the model to general applicability. Part of the ancillary data (i.e., snow depth and ice freeboard height) would be available in OTASC Level-4 products in the future, offering us an opportunity

to interpret sea ice electromagnetic properties.

    Given the sea ice scenario in this study, we assume that the snow condition over the thick and deformed ice is dry, and the X-band microwaves penetrate both the snow and ice layer. For wet-snow covered sea ice, the penetration capability of X-band is limited (Hallikainen and Winebrenner, 1992), and therefore the proposed approach in this study cannot be applied. In the

case of wet snow, as well as medium wet snow, the snow-air interface influenced by the surface roughness needs to be taken into account as it changes the X-band SAR backscatter (Nandan et al., 2016; Dufour-Beauséjour et al., 2020).

In this study, the InSAR pair in StripMap mode covers $19\,\mathrm{km} \times 50\,\mathrm{km}$ in SAR ground range and azimuth direction, respectively, providing a unique 3-dimensional (3D) topographic map rather than a narrow transect from LIDAR or photogrammetric measurements. TanDEM-X has a regular revisit cycle of 11 days over the Arctic and a larger revisit time due to the particular satellite position configuration required over Antarctica. The current SAR satellites, such as X-band TanDEM-X and COSMO-SkyMed, C-band Sentinel-1 and Radarsat Constellation, as well as the future X-band LOTUSat, L/S-band NISAR, and L-band ROSE-L, will together achieve a long-term sea ice topographic monitoring in both polar regions. Synergistic use of different SAR satellites offers more extensive spatial coverage and shorter revisit times than a single platform. In the future, the joint use of multi-frequency SAR imagery could develop a better understanding of sea ice properties and processes (Dierking and Davidson, 2021), which would be indispensable for retrieving sea ice topography at a more comprehensive range of ice conditions.

## 8  Conclusions

In this study, the potential to retrieve sea ice topography with the Pol-InSAR technique was validated with single-pass interferometric SAR data and airborne photogrammetric measurements over the thick ($> 2\,\mathrm{m}$) and deformed sea ice with snow cover in the Western Weddell sea. The DMS DEM reveals that the sea ice topography along the flight track varies from $0$ to $2.68\,\mathrm{m}$ with the average height being $1.27\,\mathrm{m}$. The average elevation difference between the conventional InSAR DEM and the DMS DEM is $\sim 1\,\mathrm{m}$ in the four investigated polarizations (HH, VV, Pauli-1, and Pauli-2), suggesting the demand for a valid method to obtain sea ice topography and to correct for the penetration of the microwave signals into the sea ice. By exploiting the interferometric coherence, a two-layer plus volume model was proposed to characterize the sea ice vertical scattering structure and an inversion scheme was developed for height retrieval. The model's theoretical accuracy was assessed for various vertical wavenumber values to ensure $25\%$-error accuracy at the employed baseline configuration with $\kappa_z = 0.28\,\mathrm{rad/m}$. The assessment of model's theoretical accuracy showed the potential to apply the model to multi-baseline configurations, giving the ability to adjust a sensor to the particular type of sea ice. For instance, a configuration with $\kappa_z = 0.40\,\mathrm{rad/m}$ ensures an effective inversion for ice-volume thickness less than $0.85\,\mathrm{m}$.

The proposed theoretical model requires seven input parameters depending on the environmental conditions over the test site, which are unavailable in many practical applications. In order to reduce the model complexity and improve the model applicability, a simplified model requiring only the input of snow depth was proposed based on the analyses of the model sensitivity to different parameterizations. For the theoretical and simplified model, the layer-to-volume ratio of the bottom layer, respectively the layer-to-layer ratio, were observed to be inversely correlated to an essential polarimetric signature: the coPol coherence. This relationship was exploited in the inversion scheme by estimating those parameters from the coPol coherence with a fitted linear function. Note that the proposed models and inversion scheme in this study were developed over a specific area containing thick and deformed ice; therefore they can not be directly applied to sea ice areas covering various ice

types. With more co-registered data acquired from SAR and reference DEMs in the future, extension of the model to regions covered by different sea ice types is worthy of further investigation.

The effectiveness of both the theoretical and simplified models and the proposed inversion scheme were verified with the DMS measurements for the sea ice height above $0.8\,\mathrm{m}$, corresponding to the thick ($> 2\,\mathrm{m}$) and deformed ice without flooding. The model-retrieved sea ice topography achieved a $RMSE$ as low as $0.26\,\mathrm{m}$, which is significantly better than the $RMSE$ of $1.10\,\mathrm{m}$ of the conventional InSAR DEM. This indicates the capability to correct for the microwave signal penetration and to generate a precise wide-swath topographic map from dual-pol single-pass InSAR data. The polarization-independent volume property of sea ice in the co-pol channels in X-band radar frequency, which was concluded from the similar retrieval

performance across HH, VV, and Pauli-1 polarizations, gave insights to develop superior models for height retrieval in the future. Next work will include investigating the possibility of sea ice topographic retrieval for various types of sea ice, such as thin ice and newly formed ice.

*Data availability.* TanDEM-X data can be obtained from the German Aerospace Center (DLR) and downloaded on the website (https://eoweb.dlr.de). DMS data can be obtained from the National Snow and Ice Data Center and downloaded on the website (https://nsidc.org/data/icebridge)

*Author contributions.* LH conducted the PolInSAR processing, ran the models, and prepared the manuscript. GF provided useful suggestions for designing the theoretical model, proposed the mathematical formulation of the simplified model, and revised the manuscript. LH and GF jointly discussed the model simulations and experimental results. IH provided valuable comments on all aspects of the modelling and experiments, and contributed to the improvements of the manuscript.

*Competing interests.* The authors declare that they have no conflict of interest.

*Acknowledgements.* The authors would like to thank everyone involved in the OTASC campaign, which was conducted by DLR and NASA. The authors would like to thank the editor Dr. Christian Haas for handling our manuscript and the two anonymous referees who provided detailed and constructive comments.

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
