# Peer review of "Antarctic snow-covered sea ice topography derivation from TanDEM-X using polarimetric SAR interferometry"

_The Cryosphere, 2021_

## Author Comment (AC1)

**Response to the comments of Reviewer 1**

First of all, we would like to thank the anonymous reviewer for the careful review and valuable suggestions. We carefully revised the manuscript following the suggestions. Hereby we give a point-by-point reply to address the comments. In this document, the words in *italics are the reviewers' comments*, the words in blue are the modifications we have made in the revision, and others are our responses.

**Q1:** *Summary. This is an interesting, detailed, well prepared paper that has important implications for deriving sea ice topography using a unique approach, with single-pass interferometry. I have some requests on clarifications that I will write down in Details. The methods are largely clearly described. I think I have three main points about the paper as a whole that I suggest the authors consider.*

**A1:** We thank the reviewer for the positive comment about our research. We have carefully revised the manuscript based on the following comments.

**Q2:** *First, the paper is presented as being applicable to both polar sea ice covers. However, particularly in the Introduction and Basic Concepts, they really don't distinguish sufficiently between the main differences in the sea ice between the Arctic and Antarctic. I have added some suggested references to include. So please add some more text about differences in ice type and snow layer.*

**A2:** We have included all suggested references. Specifically, more discussions about the main differences of ridges between the Arctic and Antarctic have been added in the Introduction:

"Timco and Burden, (1997) estimated the ratio of the keel-depth (i.e., depth of ice below the seawater) to sail-height (i.e., height of ice above the seawater) for both first-year and multi-year sea ice ridges in the Beaufort Sea and highlighted their differences in ridge height and shape. Tin and Jeffries (2003) indicated that first-year ridges in the Antarctic are flatter and less massive than those in the Arctic. Sea ice ridging height is a crucial parameter to evaluate total ice mass in both polar regions (Hibler et al., 1974; Melling and Riedel, 1995; Lytle et al., 1998; Tin et al., 2003). In the Antarctic, the mean height of the ridges in the Weddell Sea was found to be $\sim 1.1\,\mathrm{m}$ which is similar to the ridging statistics from the Ross Sea (Lytle and Ackley, 1991), whereas it is considerably less than the height from the Arctic (Lytle and Ackley, 1991; Dierking, 1995)."

The differences of snow layer and sea ice properties between Arctic and Antarctic have been added in a new paragraph in the Introduction:

"On top of the sea ice, the snow cover, which is redistributed by wind, can also distort the sea-ice topography (Webster et al., 2018). The snow layer and ice properties in the Arctic and Antarctic are significantly different due to the diverse growing conditions in the two polar regions (Gloersen, 1992; Walsh, 2009; Sturm and Massom, 2009). In the Antarctic, the snow depth is reported to be thicker than in the Arctic (Jeffries et al., 1997; Massom et al., 2001; Willatt et al., 2009). When thick enough, the snow will overburden the ice floe and be flooded by seawater, resulting in higher salinity of the snow layer in the Antarctic. Besides, compared to the Arctic, snow on the Antarctic sea ice comprises more heterogeneous layers resulting from highly variable temperature (Massom et al., 2001). The layer heterogeneity in types, density, salinity, and wetness would determines the electromagnetic characteristics of the snow. As for ice properties in general, Antarctic sea ice is reported to be thinner (Worbyet al., 2008; Kurtz and Markus, 2012; Lindsay and Schweiger, 2015), younger (Webster et al., 2018), and more saline than in the Arctic at comparable age and thickness (Gow et al., 1982, 1987). Quantitatively, the mean salinity of the Antarctic

first-year ice and multi-year ice profiles are $4.6‰$ and $3.5‰$ , respectively, whereas the average values are $3‰$ for the first-year ice and $2 - 2.5‰$ for the multi-year ice in the Arctic (Coxand Weeks, 1974). These variable properties of sea ice, its ridges and snow cover, at both small and large spatial scales, highlight the challenge and necessity for accurate sea ice topographic information with large spatial coverage and high resolution."

**Q3:** *Next, it's not clear to me that a two-layer model is sufficient, with both layers considered to be uniform, to correctly identify the phase center. In Arctic first year ice, at the snow-ice surface particularly for young first year there is often a significant layer of high salinity slush ice that may also include frost flowers. There wil be some penetration below this thin layer where the salinity is much lower. Then of course in the Antarctic, flooding at the snow-ice layer occurs due to relatively deeper snow loading on the generally thinner ice layer, as compared to the Arctic. This flooded layer has a higher salinity than the ice below but is still not likely sufficient to minimize further penetration. Plus of course there is increase in salinity near the ice-ocean boundary in all winter ice growth conditions. The slush layer is referred to in the paper, but I maintain that it is not sufficient to dismiss the possibility of a 3-layer without demonstrating otherwise which I suggest they do, as it may impact correct estimation of the phase center and therefore a modeled-derived height.*

**A3:** We agree that the slush layer is important for Arctic first-year ice and Antarctic flooded sea ice. First, in the revision, we have added more references in Section 1 (Introduction), introducing the slush layer in both polar regions:

"In the Arctic first-year thin ice, snow capillary force gives rise to brine wicking, and consequently, a layer of high salinity slush ice appears at the snow-ice interface (Reimnitz and Kempema, 1987; Drinkwater and Crocker, 1988; Nghiem et al., 1995a). In the Antarctic, ice-surface flooding widely occurs resulting from the generally thicker snow layer loading on the thinner ice floes, often followed by freezing of the slush layer at the snow-ice interface (Massom et al., 2001; Jeffrieset al., 2001; Maksym and Jeffries, 2000). Even without flooding, the upward wicking of brine from the ice surface can also form a saline layer at the bottom of the snowpack (Massom et al., 2001; Toyota et al., 2011; Webster et al., 2018). The slush layer at the snow-ice interface would induce significant surface scattering and thus has been included in the sea ice scattering modelling (Nghiem et al., 1995a, b; Maksym and Jeffries, 2000)."

Focusing on the Antarctic sea ice, we have added more explanations about the slush layer for both thinner and thicker ice conditions in Section 2 (Basic concepts). Different sea ice structures for thinner ice and thicker ice have been plotted in Fig. 1(a) and (b), respectively. We have also emphasized that in this study, the experiments are conducted based on the thicker and deformed ice with height $> 0.8\,\mathrm{m}$ above the sea level. This height threshold is estimated from hydrostatic balance, and we assume that sea ice higher than this threshold do not suffer seawater flooding. Note that even without flooding, the upward wicking of brine from the ice surface can also form a thin and high saline layer at snow-ice interface (Massom et al., 2001; Toyota et al., 2011; Webster et al., 2018). Therefore, the surface scattering from the slush layer at the snow-ice interface has been considered in the model. Below Fig. 1 and texts have been added in the new Section 2.1:

"In the Antarctic, the presence of a saline layer at the snow-ice interface due to the flooding or capillary suction of brine from the ice surface has been recognized as a widespread and critical phenomenon (Massom et al., 2001). For thinner ice, flooding may occur when the weight of the snow pushes the ice surface below the water level, yielding a negative freeboard. In this case, as shown in Fig. 1(a), seawater infiltrates into the snowpack, floods the ice surface, and creates a high-saline slush layer which may refreeze into snow ice (Lange et al., 1990; Jeffries et al., 1997; Maksym and Jeffries, 2000). The thickness of snow ice was observed to be $\sim 42 - 70\%$ of the total snow accumulation (i.e., the thickness of snow ice plus snow depth) (Jeffries et al., 2001).

[Figure]

Fig. 1: Schematic of (a) thinner ice floes flooded by seawater and (b) thicker ice floes without flooding.

For thicker and deformed ice with ridges, less flooding occurs due to the increased buoyancy of the ice mass contained in the ridges (Jeffries et al., 1998). However, even in the absence of flooding, a thin slush layer can also occur due to the capillary suction of brine from the ice surface (Massom et al., 2001; Webster et al., 2018). Besides, the deformed ice in the ridging and rafting area is often poorly consolidated, and thus seawater may reach the snow layer and form a thin slush layer (Maksymand Jeffries, 2000). The sea ice structure for thicker ice without flooding is sketched in Fig. 1(b), including snow on top, the ice volume, and a thin and high-saline layer in between.

The condition of flooding can be quantified by a simple hydrostatic balance (Lange et al., 1990)

$$\rho_w d = \rho_i d + \rho_i f + \rho_s s$$
$$t = d + f \tag{1}$$

where $\rho_w$, $\rho_i$, and $\rho_s$ are the densities of seawater, ice, and snow, respectively; $d$, $f$, $t$ and $s$ are the thickness of ice below and above the sea level, the total ice thickness, and the snow depth on top, respectively. For flooding to occur, $f$ should be zero (i.e., $d = t$), and Eq. (1) becomes

$$s/t = (\rho_w - \rho_i)/\rho_s \approx 0.12/\rho_s \tag{2}$$

by assuming $\rho_w = 1.03\,\mathrm{Mg/m^{-3}}$ and $\rho_i = 0.91\,\mathrm{Mg/m^{-3}}$ (Lange et al., 1990). For snow density $\rho_s$ being $0.3\,\mathrm{Mg/m^{-3}}$ (Lange et al., 1990), the ratio between snow depth and ice thickness $s/t$ is estimated to be $0.4$. The snow depth on Antarctic sea ice during September and November was shown to be below $0.8\,\mathrm{m}$ for $99\%$ of the samples in Webster et al., (2018). This range of snow depth will lead to flooding for ice thickness $< 2\,\mathrm{m}$.

The relation between ice thickness $H_i$ and surface height $h_{\mathrm{sur}}$ (i.e., ice height above sea surface including snow depth) has been discussed over different regions (Petty et al., 2016; Toyota et al., 2011; Ozsoy-Cicek et al., 2013). Ozsoy-Cicek et al., (2013) showed a linear relation $H_i = c_0 h_{\mathrm{sur}} + c_1$ with $c_0 = 2.24$ and $c_1 = 0.228$ fitted from large-scale, survey-averaged data over the Western Weddell Sea, which is the same region as this study. According to this linear relation, $H_i = 2\mathrm{m}$ corresponds

to a surface height of $\sim 0.8\,\mathrm{m}$.

This paper focuses on thicker ($> 2\,\mathrm{m}$) and deformed ice (Fig. 1(b)), which is the main ice typology in the studied area. In the following sections, the model and experiments are conducted only for the samples above $\sim 0.8\,\mathrm{m}$ surface height. We assume that samples exceeding this threshold are thicker and deformed ice without flooding. The potential to extend the proposed model to thinner ice scenarios (e.g. Fig. 1 (a)) is discussed in Section 7.3."

The Figure: Schematic of the proposed two-layer plus volume model for sea ice, has been updated with Fig. 2 and some sentences have been added in Section 4.2:

[Figure]

Fig. 2: Schematic of the proposed two-layer plus volume model for the thicker and deformed sea ice.

"The top layer located at $z_1$ is the snow-ice interface, which can induce significant surface scattering due to a slush layer with high permittivity (Hallikainen and Winebrenner, 1992; Maksym and Jeffries, 2000). This slush layer is widespread on the Antarctic sea ice, and increases the radar backscattering as well as limits the signal penetration compared to a smooth and dry snow-ice interface. As long as the slush layer has a small vertical extent, it is irrelevant for the Pol-InSAR scattering structure model, whether the top layer represents the snow-ice interface, the snow-slush interface or both."

To extend the proposed model for thinner ice where flooding often occurs, an additional layer, i.e., snow ice formed from the slush layer when air is cold, has been discussed. A three-volume model incorporated with the snow ice could be a promising approach to correct the InSAR phase, and it will be investigated in future study. Below texts has been added in the new discussion Section 7.3:

"The proposed model was proven to be effective in a specific area covered by thick and deformed ice with snow cover in the Western Weddell Sea. The extension of the proposed model to other ice types under different environmental conditions needs further research and suitable data.

In order to apply the model over younger and thinner sea ice, the first challenge is the severe misregistration between SAR images and reference measurements due to the stronger dynamics of thinner sea ice. Reduced SAR backscattering intensity corresponding to thinner and smoother sea ice further complicates the data co-registration. Besides, the achievable height sensitivity for thin ice is also a major limitation of InSAR/Pol-InSAR derived sea ice DEMs with current SAR systems. In this study, the proposed method can achieve sea ice topographic retrieval with an $RMSE$ of $0.26\,\mathrm{m}$ for thick and deformed ice; however, this accuracy is insufficient for thinner ice whose height above sea level is only tens-of-centimeters or even less. Last but not least, an additional volume, i.e., snow ice formed by flooding, should be considered when extending the proposed model to a thinner ice area. Past studies showed that snow ice contributes an average of $8\%$ of the total volume in

the Weddell Sea (Lange et al., 1990). A greater amount of snow ice, which accounts for $12 - 36\%$ of the total mass, was reported in the Ross, Amundsen, and Bellingshausen Seas (Jeffries et al., 2001). Although the snow ice has a higher salinity than the ice below, there could be still some penetration into the ice volume below. Therefore, in order to correct the InSAR phase center and retrieve surface height for snow-covered thin ice in the Antarctic (as illustrated in Fig. 1 (a)), a three-volume model, including snow, snow ice, and ice, would be worthy of further investigations."

**Q4:** *Finally, last main point has two components – first while they compare with a DMS height as a narrow 2D transect, which is what is available and appropriate, they also show 3D output (Figure 13). However, there isn't much discussion about the 3d output – do these appear to be representative of what might be expected or compared to other possible data or studies? There are papers on ridge/sail characteristics plus a nice example in Tucker chapter 2 in Carsey sea ice microwave book. Second part of last point is that these 3D maps are really unique because they are not just narrow transects. How often for example could these 3D maps be generated with Tandem-X, spatially and temporally? I think Tandem-X is pretty limited by its duty cycle at least and perhaps storage/downlink too. It would be good to hear about longer term capabilities for deriving this product and what might be required to validate Arctic products, for example.*

**A4:** We thank the reviewer for recommending Tucker chapter 2 in Carsey sea ice microwave book. We agree it is a great idea to extract ridge/sail characteristics from the 3D output, which is a distinct advantage of SAR out of other measurements. We have worked on these for some months. The statistical features extracted from SAR-retrieved topographic map have been analyzed and related to the Antarctic geophysical environments. We are now preparing another manuscript to present these interesting results. In the revision, based on the 3D output, we have added more discussions on the relative retrieval bias $\epsilon = (h_{\mathrm{mod\_S}} - h_{\mathrm{DMS}})/h_{\mathrm{DMS}}$ of zoom-in area 1-3. The texts and Fig. 3 (e)-(g) have been added in Section 6.1:

"The relative retrieval bias $\epsilon$, which can be calculated as $\epsilon = (h_{\mathrm{mod\_S}} - h_{\mathrm{DMS}})/h_{\mathrm{DMS}}$, is used to quantify the retrieval accuracy. In Fig. 3(e)-(g), $\epsilon$ over Area 1-3 are below $25\%$ for most parts, whereas only a few parts, often near to the masked-out regions (pixels in transparent), present higher $\epsilon$. Note that the masked-out regions refer to water and thinner ice areas with height less than $0.8\,\mathrm{m}$. The averaged $\epsilon$ over Area 1-3 are about $19\%$, $14\%$, and $15\%$, respectively, and is $18\%$ for the whole image, achieving the theoretical $25\%$-error accuracy derived in Section 4.4."

Besides, we have added more discussions regarding the spatial coverage and temporal resolution for the unique InSAR-derived 3D map. The long-term capability of SAR sensor for sea ice monitoring and the requirement to validate Arctic products have been included in the new discussion Section 7.3:

"In this study, the InSAR pair in StripMap mode covers $19\,\mathrm{km} \times 50\,\mathrm{km}$ in SAR ground range and azimuth direction, respectively, providing a unique 3-dimensional (3D) topographic map rather than a narrow transect from LIDAR or photogrammetric measurements. TanDEM-X has a regular revisit cycle of 11 days over the Arctic and a larger revisit time due to the particular satellite position configuration required over Antarctica. The current SAR satellites, such as X-band TanDEM-X and COSMO-SkyMed, C-band Sentinel-1 and Radarsat Constellation, as well as the future X-band LOTUSat, L/S-band NISAR, and L-band ROSE-L, will together achieve a long-term sea ice topographic monitoring in both polar regions. Synergistic use of different SAR satellites offers more extensive spatial coverage and shorter revisit times than a single platform (Dierking and Davidson, 2021). In the future, the joint use of multi-frequency SAR imagery could develop a better understanding of sea ice properties and processes, which would be indispensable for retrieving sea ice topography at a more comprehensive range of ice conditions.

In order to assess the transferability of the proposed model to the Arctic regions, further validations, including co-registered

[Figure]

Fig. 3: Sea ice topographic retrieval with the simplified model ($h_{\mathrm{mod\_S}}$). The transect from the DMS DEM is plotted between grey lines. Note that the heights below $0.8\,\mathrm{m}$ are set to be transparent. (a) The whole studied SAR image. (b)-(d) Zoom-in of Areas 1-3. (e)-(g) Relative retrieval bias $\epsilon = (h_{\mathrm{mod\_S}} - h_{\mathrm{DMS}})/h_{\mathrm{DMS}}$ of Area 1-3.

SAR images and topographic reference (e.g., optical/LIDAR measurements), are needed considering the significant difference of ice and snow properties between the Arctic and Antarctic. Ancillary measurements (e.g., snow depth, temperature, ice and snow salinity) at a wide ranger of ice conditions in both polar regions are crucial to understand the properties of various typologies of ice, and therefore are valuable for extending the model to general applicability. Part of the ancillary data (i.e., snow depth and ice freeboard height) would be available in OTASC Level-4 products in the future, offering us an opportunity to interpret sea ice electromagnetic properties."

**Q5:** *Lines16-18. Add references that discuss ridge characteristics, etc in addition to Rampal reference for both poles, for example for Antarctic, Lytle et al. Annals Glaciology 1998, two Tin and Jeffries papers in 2003/04, plus Timco and Burden 1997 for Arctic.*

**A5:** The references that discuss ridge characteristics in both polar regions have been added as suggested in the revision:

"Timco and Burden, (1997) estimated the ratio of the keel-depth (i.e., depth of ice below the seawater) to sail-height (i.e., height of ice above the seawater) for both first-year and multi-year sea ice ridges in the Beaufort Sea and highlighted their differences in ridge height and shape. Tin and Jeffries (2003) indicated that first-year ridges in the Antarctic are flatter and less massive than those in the Arctic. Sea ice ridging height is a crucial parameter to evaluate total ice mass in both polar regions (Hibler et al., 1974; Melling and Riedel, 1995; Lytle et al., 1998; Tin et al., 2003). In the Antarctic, the mean height of the ridges in the Weddell Sea was found to be $\sim 1.1\,\mathrm{m}$ which is similar to the ridging statistics from the Ross Sea (Lytle and Ackley, 1991), whereas it is considerably less than the height from the Arctic (Lytle and Ackley, 1991; Dierking, 1995)."

**Q6:** *Line 24, Tucker et al reference is for first year ice only. please clarify.*

**A6:** It has been clarified in the revision:

"Focusing on first-year sea ice in the Alaska region, Tucker and Govoni (1981) observed a square-root relation between the ridge height and thickness, which is further validated by additional in situ observations in (Tucker et al., 1984)."

**Q7:** *Line 25. Petty reference discusses both FY and MY and differences. Please mention in text. Also following Toyota paper, there is a really good chapter on Snow by Sturm and Massom in Sea Ice book edited by Thomas and a recent chapter by Webster et al Nature Climate Change 2018.*

**A7:** In the revision, we have added that Petty's reference discusses both FY and MY and differences:

"Petty et al. (2016) presented a detailed characterization of Arctic sea ice topography across both first-year and multi-year sea ice and analyzed the topographic differences between the two ice regimes."

As suggested, the references have been added in line 18 and in a new paragraph before line 19:

"On top of the sea ice, the snow cover, which is redistributed by wind, can also distort the sea-ice topography (Webster et al., 2018). The snow layer and ice properties in the Arctic and Antarctic are significantly different due to the diverse growing conditions in the two polar regions (Gloersen, 1992; Walsh, 2009; Sturm and Massom, 2009)."

**Q8:** *Lines 30-31. Add journal papers that utilize DMS data in addition to Dotson and Aversen references.*

**A8:** We have add an important conference paper (Nghiem et al.,2018) from OTASC team. Another journal paper using both DMS and SAR data for iceberg topographic retrieval (Dammann et al., 2019) was given in Section 3.1, Line 143.

- Nghiem, S., Busche, T., Kraus, T., Bachmann, M., Kurtz, N., Sonntag, J., Woods, J., Ackley, S., Xie, H., Maksym, T., et al.: RemoteSensing of Antarctic Sea Ice with Coordinated Aircraft and Satellite Data Acquisitions, in: Proc. IGARSS., pp. 8531–8534, IEEE,https://doi.org/10.1109/IGARSS.2018.8518550, 2018.
- Dammann, D. O., Eriksson, L. E. B., Nghiem, S. V., Pettit, E. C., Kurtz, N. T., Sonntag, J. G., Busche, T. E., Meyer, F. J., andMahoney, A. R.: Iceberg topography and volume classification using TanDEM-X interferometry, The Cryosphere, 13, 1861–1875,https://doi.org/10.5194/tc-13-1861-2019, 2019.

In the revision Line 30-31 has been updated:

"The sea ice topography can be measured by various instruments, such as laser altimeters (Dierking, 1995; Schutz et al.,2005; Abdalati et al., 2010; Farrell et al., 2011, 2020) and stereo cameras using photogrammetric techniques (Dotson and Arvesen.,

2012, updated 2014; Divine et al., 2016; Nghiem et al., 2018; Li et al., 2019)."

**Q9:** *Line 53. Substitute 'deficient brine' for 'reduced brine'*

**A9:** It has been changed as suggested.

**Q10:** *Line 61. Substitute 'obtain an' for 'obtain a more' accurate.*

**A10:** It has been changed as suggested.

**Q11:** *Lines 77-83. This paragraph should be expanded to discuss thin salinity layers at snowice interface as mentioned in the summary with references*

**A11:** The paragraph has been expanded by adding more discussions of the high salinity layer at the snow-ice interface in the revision:

"In the Arctic first-year thin ice, snow capillary force gives rise to brine wicking, and consequently, a layer of high salinity slush ice appears at the snow-ice interface (Reimnitz and Kempema, 1987; Drinkwater and Crocker, 1988; Nghiem et al., 1995a). In the Antarctic, ice-surface flooding widely occurs resulting from the generally thicker snow layer loading on the thinner ice floes, often followed by freezing of the slush layer at the snow-ice interface (Massom et al., 2001; Jeffrieset al., 2001; Maksym and Jeffries, 2000). Even without flooding, the upward wicking of brine from the ice surface can also form a saline layer at the bottom of the snowpack (Massom et al., 2001; Toyota et al., 2011; Webster et al., 2018). The slush layer at the snow-ice interface would induce significant surface scattering and thus has been included in the sea ice scattering modelling (Nghiem et al., 1995a, b; Maksym and Jeffries, 2000)."

**Q12:** *Figure 8. I guess I really don't understand these figures. I looked and looked at how one might determine that these graphs suggest phase centers of 6-7cm and 15-33 cm as described in the text – Lines 312-315. I would appreciate an explanation of what information they are using from these figures and how they are deriving the phase centers. I also hope the editors are getting a review from another person who has a lot more INSAR and radar modeling expertise than me.*

**A12:** Sorry for the unclear statement. In the preprint manuscript, Fig. 8 shows the complex coherence, modelled with Eq. (12), in the unit circle. The radius corresponds to the coherence magnitude, the angular rotation to the phase. The phase can be translated to height via

$$h_{\text{volume}} = \frac{\angle \tilde{\gamma}_v}{\kappa_{\text{z\_vol}}} \tag{3}$$

where $\tilde{\gamma}_v$ can be substituted with $\tilde{\gamma}_{\text{mod\_T}}$ derived from Eq. (12) (in the preprint manuscript).

The above equation has been added in Section 2, and the following texts has been added in Section 4.3:

"The sensitivity of $\tilde{\gamma}_{\text{mod\_T}}$ to various parameters is presented in Fig. 9, where the radius and angular rotation corresponds to the coherence magnitude and phase, respectively. The phase can be translated to height via Eq. (3)."

Second, we would like to clarify that the volume-only phase centers are not directly derived from the plots but from Eq. (11) (in the preprint manuscript). The complex value of $\tilde{\gamma}_v(\sigma_1, z_{01})$ can be obtained according to Eq. (11) (in the preprint manuscript), and the phase part is denoted as $\angle \tilde{\gamma}_v(\sigma_1, z_{01})$. The derived phase can be converted to height by Eq. (3). As the range of $\sigma_1$ is $1 - 10\,\text{db/m}$, the corresponding phase center height is calculated to be $-6$ to $-7\,\text{cm}$. Similarly, the phase $\angle \tilde{\gamma}_v(\sigma_2, z_{12})$ can be obtained according to Eq. (11) (in the preprint manuscript) at different values of $\sigma_2$ and converted to height by Eq. (3). Across the range of $\sigma_2$ (i.e., $10 - 200\,\text{db/m}$), the phase center height varies from $-15$ to $-33\,\text{cm}$. These explanations have been added in the revision:

"The complex coherence of the snow volume $\tilde{\gamma}_v(\sigma_1, z_{01})$ can be calculated by Eq. (15) with thickness $z_{01} = 15\,\mathrm{cm}$, and its magnitude and phase can be denoted as $|\tilde{\gamma}_v(\sigma_1, z_{01})|$ and $\angle\tilde{\gamma}_v(\sigma_1, z_{01})$, respectively. Then, the phase center location of the snow volume alone can be calculated by Eq. (3). Across the range of $\sigma_1$ (i.e., $1 - 10\,\mathrm{db/m}$), the snow volume has an individual coherence magnitude (i.e., $|\tilde{\gamma}_v(\sigma_1, z_{01})|$) close to unity and phase center height varying from $-6$ to $-7\,\mathrm{cm}$. Similarly, the ice volume $\tilde{\gamma}_v(\sigma_2, z_{12})$ has an individual coherence magnitude of almost unity and a phase center height between $-15$ to $-33\,\mathrm{cm}$ for the investigated range of ice extinction coefficients."

**Q13:** *Figure 14. I think the grey lines for the removed sections are too distracting from both a) and b) and perhaps just not included in the graph. Are the grey removed sections the same segments as described in Figure 4 as being mis-registered and set at 0 height? If so, not sure why they need to be included in Figure 14 at all.*

**A13:** As explained in A3, in this study, we focus on the snow-covered sea ice thicker than $\sim 2\,\mathrm{m}$, corresponding to $0.8\,\mathrm{m}$ height above seawater. Therefore, the mis-registered segments and pixels with DMS height below $0.8\,\mathrm{m}$ were removed. In the revision, all these removed pixels have been set to be NaN values, and the plot is updated without grey lines (see Fig 4 below):

[Figure]

Fig. 4: (a) Sea ice height profiles from DMS measurement (blue) and model (red). (b) Height difference between the DMS measurement and the simplified model-derived height (blue), theoretical model-derived height (red), or original InSAR-derived height (green). The mis-coregistered and $h_{\mathrm{DMS}}$ below $0.8\,\mathrm{m}$ samples are excluded from the plots.

**Q14:** *Lines 474. Snow depth is not well correlated with thickness or even FY or MY ice and in both polar regions.*

**A14:** In this study, snow depth is assumed as a constant value across the scene due to the limited spatial resolution of available snow measurements. We admit that this single value is likely not representative of the actual spatial snow depth distribution, which is related to ice thickness and ice condition. Therefore, in the revision, we have performed the whole inversion scheme and retrieved heights assuming various $z_1$ values from $-0.05\,\mathrm{m}$ to $-0.75\,\mathrm{m}$, according to the snow depth on Antarctic sea ice reported in Webster et al., (2018). Using varying snow depths, we have demonstrated how the retrieved height would agree with the DMS DEM and discussed the impact of the snow depth parameter on the experimental results. Below texts and figures have been added in the new Section 7.2:

"In Section 4.3, we demonstrated that the influence of snow depth on the simulated coherences is not negligible, and stated

that external data of snow measurements should be used in the model. In this study, snow depth is assumed to be invariant across the scene due to the limited spatial resolution of available snow measurements. Therefore, a constant value of $z_1 = -0.18$ is used in the retrieval. Actually, snow on sea ice undergoes temporal- and spatial-variant processes and is strongly coupled with atmospheric, oceanic, and ice conditions. Thus, a single value is not representative of the actual spatial snow depth distribution. In order to assess the impact of the snow depth on the experimental results, we perform the whole inversion scheme with various inputs of $z_1$. During September and November, the snow depth on Antarctic sea ice is reported to be maximum $\sim 1\,\mathrm{m}$ and mainly between $0$ and $0.8\,\mathrm{m}$ (Webster et al., 2018). Therefore, $z_1$ values ranging from $-0.05\,\mathrm{m}$ to $-0.75\,\mathrm{m}$ are selected. For each pixel, we retrieve heights using this range of $z_1$ values, shown as the yellow area in Fig. 5. $\Delta h_{\mathrm{mod\_S}}$ is defined as the difference between the maximum and the minimum retrieved height of every pixel. The distribution of $\Delta h_{\mathrm{mod\_S}}$ along the transect is presented in Fig. 6, where $\Delta h_{\mathrm{mod\_S}}$ has a range of $0.07 - 1.09\,\mathrm{m}$ with an average of $0.31\,\mathrm{m}$, indicating the fluctuation of model-retrieved height by using different snow depths. This analysis with various snow depth assumptions can help to constrain possible model-retrieved topographies, and $\Delta h_{\mathrm{mod\_S}}$ can be a quantitative indicator for the uncertainty of the retrieved height in the absence of high-resolution snow depth data."

[Figure]

Fig. 5: Yellow area: sea ice height profiles from the simplified model in Pauli-1 polarization with $z_1$ from $-0.05\,\mathrm{m}$ to $-0.75\,\mathrm{m}$. Blue dash line: sea ice height profiles from DMS measurement. The mis-coregistered and $h_{\mathrm{DMS}}$ below $0.8\,\mathrm{m}$ samples are excluded from the plot.

**Q15:** *Lines 509-511. This is a good sentence and touches back at some of my points in the summary and the need to clarify some references in the introduction and Basic Concept section, the idea of model that may need to improvement etc as well as how these type of products could be expanded in Tandem-X acquisitions and products.*

**A15:** The reply to this comment is a repetition from earlier answers (**A2 and A3**). Specifically, we have clarified more references in the introduction about the different characteristics of snow and ice in the Arctic and Antarctic, **see A2**. The idea of the model and how the model can be extended to other ice conditions have been further discussed in the revision, **see A3**.

**Q16:** *Basic concepts or Model Development or Discussion. I really do think it's important to consider a third thin high salinity layer at the snow-ice interface, whether on thin first year ice or flooded ice. I realize this might be a lot of extra work and at this stage of your study, it may not be of primary importance. This could also be added to the Model section or Discussion section too at minimum, as a topic for further research and what you think the impact might be on the model. Of course, the authors could tell me that they don't think it's a worthy topic at all and won't make any difference. I do think firmly that their two-layer is not universally applicable to all the major ice types and conditions for both polar regions, based on my understanding of their model. New and young ice are often the trickiest anyway to deal with any radar algorithm.*

[Figure]

Fig. 6: The distribution of $\Delta h_{\mathrm{mod\_S}}$ along the transect.

*Throughout the paper as I was reviewing it, I kept thinking about those two thin salinity layers and differences between first year and multiyear etc in both poles and how this should all be considered in a model of radar penetrating sea ice.*

**A16:** The reply to this comment is a repetition from earlier answers (**A3**). We agree with the review that a third thin high salinity layer at the snow-ice interface, whether on thin first year ice or flooded ice, is important. The discussions of slush layer have been added in Section 1 (Introduction), Section 2 (Basic conception), and the new Section 7 (Discussion), see **A3** for details.

Again, we sincerely thank the editor and reviewers for helping us improving the manuscript.

---

## Author Comment (AC2)

**Response to the comments of Reviewer 2**

First of all, we would like to thank the anonymous reviewer for the careful review and valuable suggestions. We carefully revised the manuscript following the suggestions. Hereby we give a point-by-point reply to address the comments. In this document, the words in *italics are the reviewers' comments*, the words in blue are the modifications we have made in the revision, and others are our responses.

**Q1:** *The manuscript Antarctic snow-covered sea ice topography derivation from TanDEM-X using polarimetric SAR interferometry by Huang et al. presents the development and validation of a new two-layer plus volume sea ice model with the aim to correct for the height bias associated with InSAR penetration into the snow pack. This model is able to represent the sea ice/snow stratigraphy and associated scattering, and, when simplified and inverted, allows for the estimation of the sea ice plus snow surface topography from TanDEM-X. This retrieval technique shows strong agreement to an Operation IceBridge optical (DMS) DEM that was collected contemporaneously as part of the OIB/TanDEM-X Coordinated Science Campaign. This manuscript is well-written and thoroughly presents novel methods and results that could be useful to the broader sea ice community. I have a few relatively minor comments and suggestions that should be considered, found in the general and specific comments below. The main comments I have on the manuscript deal with (1) the height threshold used (2) X-band scattering/slush layers and (3) the snow depth parameter.*

**A1:** We thank the reviewer for the positive comment about our research. We have carefully revised the manuscript based on the following comments.

**Q2:** *GC1: To me, it appears there is some mix-up with the height threshold used to keep model-error accuracy to within 25volume (z1-z2) needs to be thicker than 1.5m to achieve this accuracy. However, in later sections only ice+snow heights above the local sea surface (effectively the total freeboard) above 1.5m are used. Doing so filters out ice volumes much thicker than 1.5m, since most of the ice volume is below the waterline. I would suggest the authors confirm that the 1.5m threshold is indeed for the ice volume, and recommend that they filter the InSAR retrieved heights accordingly (which should result in a much lower height-above-sea-surface threshold).*

**A2:** We confirm that the $\sim 1.5\,\mathrm{m}$ threshold is for the ice volume to achieve a $\leq 25\%$-error inversion accuracy. In the revision, the applied height threshold was change to $0.8\,\mathrm{m}$ in order to select ice that is deformed and thick without seawater flooding (see Fig. 1(b)). In this case, ice thickness should exceed $2\,\mathrm{m}$ (**see details in A3**), corresponding to surface height of $\sim 0.8\,\mathrm{m}$ (Ozsoy-Cicek et al., 2013). Therefore, the samples with height above $0.8\,\mathrm{m}$ are selected for processing. In the revision, experimental parts have been updated as suggested, and the sentences below have been added in the new Section 2.1:

"The relation between ice thickness $H_i$ and surface height $h_{\mathrm{sur}}$ (i.e., ice height above sea surface including snow depth) has been discussed over different regions (Petty et al., 2016; Toyota et al., 2011; Ozsoy-Cicek et al., 2013). Ozsoy-Cicek et al., (2013) showed a linear relation $H_i = c_0 h_{\mathrm{sur}} + c_1$ with $c_0 = 2.24$ and $c_1 = 0.228$ fitted from large-scale, survey-averaged data over the Western Weddell Sea, which is the same region as this study. According to this linear relation, $H_i = 2\mathrm{m}$ corresponds to a surface height of $\sim 0.8\,\mathrm{m}$."

**Q3:** *GC2: (This is similar to that from reviewer 1) While the scattering impacts of a slush layer are briefly mentioned, I feel that their impact should either be discussed further or/and incorporated into the model in some way. A slush layer at*

*the snow-ice interface would surely effect the radar return differently than if the snow-ice interface was smooth and dry. Also, some mention of the effects of surface roughness would be beneficial, as snow surface/interface roughness has been found to influence X-band backscatter (Nandan et al. 2016, Remote Sens. Of Envir., https://doi.org/10.1016/j.rse.2016.10.004). Finally, while surface melt may not be present in this particular region or season, a wet snow surface could also influence the X-band backscatter (Dufour-Beauséjour et al. 2020, The Cryosphere, https://doi.org/10.5194/tc-14-1595-2020). This would need to be taken into account if applying this technique to other regions and/or seasons.*

**A3:** We agree that the slush layer is important for Antarctic sea ice. In the revision, we have added more references introducing the slush layer in Section 1 (Introduction):

"In the Arctic first-year thin ice, snow capillary force gives rise to brine wicking, and consequently, a layer of high salinity slush ice appears at the snow-ice interface (Reimnitz and Kempema, 1987; Drinkwater and Crocker, 1988; Nghiem et al., 1995a). In the Antarctic, ice-surface flooding widely occurs resulting from the generally thicker snow layer loading on the thinner ice floes, often followed by freezing of the slush layer at the snow-ice interface (Massom et al., 2001; Jeffrieset al., 2001; Maksym and Jeffries, 2000). Even without flooding, the upward wicking of brine from the ice surface can also form a saline layer at the bottom of the snowpack (Massom et al., 2001; Toyota et al., 2011; Webster et al., 2018). The slush layer at the snow-ice interface would induce significant surface scattering and thus has been included in the sea ice scattering modelling (Nghiem et al., 1995a, b; Maksym and Jeffries, 2000)."

Besides, focusing on the Antarctic sea ice, we have added more explanations about the slush layer for both thinner and thicker ice conditions in Section 2 (Basic concepts). Different sea ice structures for thinner ice and thicker ice have been plotted in Fig. 1(a) and (b), respectively. We have also emphasized that in this study, the experiments are conducted based on the thicker and deformed ice with height $> 0.8\,\mathrm{m}$ above the sea level. This height threshold is estimated from hydrostatic balance, and we assume that sea ice higher than this threshold do not suffer seawater flooding. Note that even without flooding, the upward wicking of brine from the ice surface can also form a thin and high saline layer at snow-ice interface (Massom et al., 2001; Toyota et al., 2011; Webster et al., 2018). Therefore, the surface scattering from the slush layer at the snow-ice interface has been considered in the model. Below Fig. 1 and texts have been added in the new Section 2.1:

[Figure]

Fig. 1: Schematic of (a) thinner ice floes flooded by seawater and (b) thicker ice floes without flooding.

"In the Antarctic, the presence of a saline layer at the snow-ice interface due to the flooding or capillary suction of brine from the ice surface has been recognized as a widespread and critical phenomenon (Massom et al., 2001). For thinner ice, flooding may occur when the weight of the snow pushes the ice surface below the water level, yielding a negative freeboard. In this case, as shown in Fig. 1(a), seawater infiltrates into the snowpack, floods the ice surface, and creates a high-saline slush layer which may refreeze into snow ice (Lange et al., 1990; Jeffries et al., 1997; Maksym and Jeffries, 2000). The thickness of snow ice was observed to be $\sim 42 - 70\%$ of the total snow accumulation (i.e., the thickness of snow ice plus snow depth) (Jeffries et al., 2001).

For thicker and deformed ice with ridges, less flooding occurs due to the increased buoyancy of the ice mass contained in the ridges (Jeffries et al., 1998). However, even in the absence of flooding, a thin slush layer can also occur due to the capillary suction of brine from the ice surface (Massom et al., 2001; Webster et al., 2018). Besides, the deformed ice in the ridging and rafting area is often poorly consolidated, and thus seawater may reach the snow layer and form a thin slush layer (Maksymand Jeffries, 2000). The sea ice structure for thicker ice without flooding is sketched in Fig. 1(b), including snow on top, the ice volume, and a thin and high-saline layer in between.

The condition of flooding can be quantified by a simple hydrostatic balance (Lange et al., 1990)

$$\rho_w d = \rho_i d + \rho_i f + \rho_s s$$
$$t = d + f$$

(1)

where $\rho_w$, $\rho_i$, and $\rho_s$ are the densities of seawater, ice, and snow, respectively; $d$, $f$, $t$ and $s$ are the thickness of ice below and above the sea level, the total ice thickness, and the snow depth on top, respectively. For flooding to occur, $f$ should be zero (i.e., $d = t$), and Eq. (1) becomes

$$s/t = (\rho_w - \rho_i)/\rho_s \approx 0.12/\rho_s$$

(2)

by assuming $\rho_w = 1.03\,\mathrm{Mg/m^{-3}}$ and $\rho_i = 0.91\,\mathrm{Mg/m^{-3}}$ (Lange et al., 1990). For snow density $\rho_s$ being $0.3\,\mathrm{Mg/m^{-3}}$ (Lange et al., 1990), the ratio between snow depth and ice thickness $s/t$ is estimated to be $0.4$. The snow depth on Antarctic sea ice during September and November was shown to be below $0.8\,\mathrm{m}$ for $99\%$ of the samples in Webster et al., (2018). This range of snow depth will lead to flooding for ice thickness $< 2\,\mathrm{m}$.

The relation between ice thickness $H_i$ and surface height $h_{\mathrm{sur}}$ (i.e., ice height above sea surface including snow depth) has been discussed over different regions (Petty et al., 2016; Toyota et al., 2011; Ozsoy-Cicek et al., 2013). Ozsoy-Cicek et al., (2013) showed a linear relation $H_i = c_0 h_{\mathrm{sur}} + c_1$ with $c_0 = 2.24$ and $c_1 = 0.228$ fitted from large-scale, survey-averaged data over the Western Weddell Sea, which is the same region as this study. According to this linear relation, $H_i = 2\mathrm{m}$ corresponds to a surface height of $\sim 0.8\,\mathrm{m}$.

This paper focuses on thicker ($> 2\,\mathrm{m}$) and deformed ice (Fig. 1(b)), which is the main ice typology in the studied area. In the following sections, the model and experiments are conducted only for the samples above $\sim 0.8\,\mathrm{m}$ surface height. We assume that samples exceeding this threshold are thicker and deformed ice without flooding. The potential to extend the proposed model to thinner ice scenarios (e.g. Fig. 1 (a)) is discussed in Section 7.3."

The Figure: Schematic of the proposed two-layer plus volume model for sea ice, has been updated with Fig. 2 and some sentences have been added in Section 4.2:

"The top layer located at $z_1$ is the snow-ice interface, which can induce significant surface scattering due to a slush layer

[Figure]

Fig. 2: Schematic of the proposed two-layer plus volume model for the thicker and deformed sea ice.

with high permittivity (Hallikainen and Winebrenner, 1992; Maksym and Jeffries, 2000). This slush layer is widespread on the Antarctic sea ice, and increases the radar backscattering as well as limits the signal penetration compared to a smooth and dry snow-ice interface. As long as the slush layer has a small vertical extent, it is irrelevant for the Pol-InSAR scattering structure model, whether the top layer represents the snow-ice interface, the snow-slush interface or both."

To extend the proposed model for thinner ice where flooding often occurs, an additional layer, i.e., snow ice formed from the slush layer when air is cold, has been discussed. A three-volume model incorporated with the snow ice could be a promising approach to correct the InSAR phase, and it will be investigated in future study. Below texts has been added in the new discussion Section 7.3:

"The proposed model was proven to be effective in a specific area covered by thick and deformed ice with snow cover in the Western Weddell Sea. The extension of the proposed model to other ice types under different environmental conditions needs further research and suitable data.

In order to apply the model over younger and thinner sea ice, the first challenge is the severe misregistration between SAR images and reference measurements due to the stronger dynamics of thinner sea ice. Reduced SAR backscattering intensity corresponding to thinner and smoother sea ice further complicates the data co-registration. Besides, the achievable height sensitivity for thin ice is also a major limitation of InSAR/Pol-InSAR derived sea ice DEMs with current SAR systems. In this study, the proposed method can achieve sea ice topographic retrieval with an $RMSE$ of $0.26\,\mathrm{m}$ for thick and deformed ice; however, this accuracy is insufficient for thinner ice whose height above sea level is only tens-of-centimeters or even less. Last but not least, an additional volume, i.e., snow ice formed by flooding, should be considered when extending the proposed model to a thinner ice area. Past studies showed that snow ice contributes an average of $8\%$ of the total volume in the Weddell Sea (Lange et al., 1990). A greater amount of snow ice, which accounts for $12-36\%$ of the total mass, was reported in the Ross, Amundsen, and Bellingshausen Seas (Jeffries et al., 2001). Although the snow ice has a higher salinity than the ice below, there could be still some penetration into the ice volume below. Therefore, in order to correct the InSAR phase center and retrieve surface height for snow-covered thin ice in the Antarctic (as illustrated in Fig. 1 (a)), a three-volume model, including snow, snow ice, and ice, would be worthy of further investigations."

Roughness plays an important role for the backscattering from surfaces and interfaces. Nandan et al., (2016) demonstrated the roughness effects through modelling, where they assume the same roughness for the snow-air surface, the snow-snow interfaces, and the snow-ice interface, for a brine-wetted snow cover on smooth FYI ice. While the scope of our Pol-InSAR

model is less detailed than their ground-based study, we account for roughness induced surface/interface backscatter effects, in relative terms, by the layer-to-volume ratios and respectively the layer-to-layer ratios. However, in contrast, we neglect an air-snow surface scattering contribution due to the required simplifications for the satellite based Pol-InSAR application. Given the quite different sea ice scenario in our study, we consider this appropriate. The following explanation has been added in Section 4.2:

"An additional parameter, the layer-to-volume scattering ratio, accounts for the (relative) scattering from these interfaces, depending e.g. on roughness and dielectric contrast (Fischer et al., 2018)."

Some mentions and references describing the effects from snow surface, especially a wet snow surface effects during melting season has been added in the new discussion Section 7.3:

"To apply the model to other seasons, the snow-air surface may also be incorporated into the model since snow roughness has been found to influence X-band SAR backscatter (Nandan et al., 2016), especially for wet snow surface which often occurs during melting season (Dufour-Beauséjour et al., 2020)."

**Q4:** *GC3: The paper states that the influence of snow depth on $\gamma_{mod_T}$ is not negligible, and that a priori data from external sources must be used in the simplified model. If I understand correctly, the passive-microwave-derived snow depth data used as the sole model parameter results in a single snow depth value (18 cm) for each pixel across the scene. While I understand that high-resolution snow depth data is generally not available, this single value is likely not representative of the actual spatial snow depth distribution (and perhaps not realistic for heights ¿1.5m, as a quick hydrostatic calculation of ice thickness with this snow depth yields abnormally thick ice). Therefore, I'm curious as to the impact of the snow depth parameter on the experimental results (beyond what is shown in the simulated results of figure 8), and if/how the retrieved heights would agree with the DMS DEM under e.g. spatially-varying snow depths.*

**A4:** In this study, snow depth is assumed as a constant value across the scene due to the limited spatial resolution of available snow measurements. We agree with the reviewer that this single value is likely not representative of the actual spatial snow depth distribution. Therefore, in the revision, we have performed the whole inversion scheme and retrieved heights with various inputs of $z_1$. According to the snow depth on Antarctic sea ice reported in Webster et al., (2018), various $z_1$ are selected from $-0.05\,\mathrm{m}$ to $-0.75\,\mathrm{m}$. Using varying snow depths, we have demonstrated how the retrieved height would agree with the DMS DEM and discussed the impact of the snow depth parameter on the experimental results. Below texts and figures have been added in the revision:

"In Section 4.3, we demonstrated that the influence of snow depth on the simulated coherences is not negligible, and have stated that external data of snow measurements should be used in the model. In this study, snow depth is assumed to be invariant across the scene due to the limited spatial resolution of available snow measurements. Therefore, a constant value of $z_1 = -0.18$ is used in the retrieval. Actually, snow on sea ice undergoes temporal- and spatial-variant processes and is strongly coupled with atmospheric, oceanic, and ice conditions. Thus, a single value is not representative of the actual spatial snow depth distribution. In order to assess the impact of the snow depth on the experimental results, we perform the whole inversion scheme with various inputs of $z_1$. During September and November, the snow depth on Antarctic sea ice is reported to be maximum $\sim 1\,\mathrm{m}$ and mainly between $0$ and $0.8\,\mathrm{m}$ (Webster et al., 2018). Therefore, $z_1$ values ranging from $-0.05\,\mathrm{m}$ to $-0.75\,\mathrm{m}$ are selected. For each pixel, we retrieve heights using this range of $z_1$ values, shown as the yellow area in Fig. 3. $\Delta h_{\mathrm{mod\_S}}$ is defined as the difference between the maximum and the minimum retrieved height of every pixel. The distribution of $\Delta h_{\mathrm{mod\_S}}$ along the transect is presented in Fig. 4, where $\Delta h_{\mathrm{mod\_S}}$ has a range of $0.07 - 1.09\,\mathrm{m}$ with an average of $0.31\,\mathrm{m}$,

indicating the fluctuation of model-retrieved height by using different snow depths. This analysis with various snow depth assumptions can help to constrain possible model-retrieved topographies, and $\Delta h_{\mathrm{mod\_S}}$ can be a quantitative indicator for the uncertainty of the retrieved height in the absence of high-resolution snow depth data."

[Figure]

Fig. 3: Yellow area: sea ice height profiles from the simplified model in Pauli-1 polarization with $z_1$ from $-0.05\,\mathrm{m}$ to $-0.75\,\mathrm{m}$. Blue dash line: sea ice height profiles from DMS measurement. The mis-coregistered and $h_{\mathrm{DMS}}$ below $0.8\,\mathrm{m}$ samples are excluded from the plot.

[Figure]

Fig. 4: The distribution of $\Delta h_{\mathrm{mod\_S}}$ along the transect.

**Q5:** *Lines 23-25: I find this sentence slightly confusing as it's written, especially since Petty et al. 2016 also mention the "close correspondence" between the predicted (surface height+square root relation) and OIB-measured thickness. Just noting the +/-2m difference makes it sound like a poor retrieval.*

**A5:** $\pm 2\,\mathrm{m}$ is the maximum differences between measured and predicted ice thickness. Considering that the thickness in the study area varies from $0$ to $8\,\mathrm{m}$ and the IceBridge measurements of ice thickness readily include an inherent uncertainty of $0.8\,\mathrm{m}$, this study (Petty et al. 2016) provided a useful method of understanding ice topography and thickness variability. The sentences have been changed in the revision:

"Petty et al. (2016) presented a detailed characterization of Arctic sea ice topography across both first-year and multi-year sea ice and analyzed the topographic differences between the two ice regimes. A square-root relation function between sea

ice topographic height and thickness was established for ice thickness retrieval (Petty et al., 2016). The results demonstrated a maximum $\pm 2\,\mathrm{m}$ difference between the measured and predicted ice thickness. Note that the measured thickness ranges from $0$ to $8\,\mathrm{m}$ with an initial uncertainty of $0.8\,\mathrm{m}$ (Petty et al., 2016)."

**Q6:** *Lines 29-31: Since you mention that characterization of sea ice topography is an active area of research (line 28), I would suggest citing more recent studies using laser altimetry and photogrammetry (e.g. Farrell et al. 2020, https://doi.org/10.1029/2020GL090708; Li et al. 2019 https://doi.org/10.3390/rs11070784; and/or others).*

**A6:** Thanks for the references. We have added more recent references as suggested.

- Farrell, S., Duncan, K., Buckley, E., Richter-Menge, J., and Li, R.: Mapping sea ice surface topography in high fidelity with ICESat-2,Geophysical Research Letters, 47, e2020GL090 708, https://doi.org/10.1029/2020GL090708, 2020.

- Li, T., Zhang, B., Cheng, X., Westoby, M. J., Li, Z., Ma, C., Hui, F., Shokr, M., Liu, Y., Chen, Z., Zhai, M., and Li, X.: Resolving Fine-Scale Surface Features on Polar Sea Ice: A First Assessment of UAS Photogrammetry Without Ground Control, Remote Sensing, 11,https://doi.org/10.3390/rs11070784, 2019.

- Nghiem, S., Busche, T., Kraus, T., Bachmann, M., Kurtz, N., Sonntag, J., Woods, J., Ackley, S., Xie, H., Maksym, T., et al.: RemoteSensing of Antarctic Sea Ice with Coordinated Aircraft and Satellite Data Acquisitions, in: Proc. IGARSS., pp. 8531–8534, IEEE, https://doi.org/10.1109/IGARSS.2018.8518550, 2018.

The sentence in Line 29-31 has been updated as:

"The sea ice topography can be measured by various instruments, such as laser altimeters (Dierking, 1995; Schutz et al.,2005; Abdalati et al., 2010; Farrell et al., 2011, 2020) and stereo cameras using photogrammetric techniques (Dotson and Arvesen., 2012, updated 2014; Divine et al., 2016; Nghiem et al., 2018; Li et al., 2019)."

**Q7:** *Line 87: By previous work, do you mean Huang and Hajnsek (2021)? Or previous studies in general?*

**A7:** Sorry for the vague statement. The previous work refers to the work in Huang and Hajnsek (2021). This has been clarified in the revision.

**Q8:** *Line 91: Same as above comment. If previous work is referring to Huang and Hajnsek (2021), I would suggest writing that explicitly.*

**A8:** It has been clarified in the revision.

**Q9:** *Line 179-180: How are water-surface points selected? And how many pixels/points are used in this scene? More information would be useful to ensure that these reference surface elevations are not biased due to e.g. newly frozen leads.*

**A9:** In this study, 9 points/pixels are labeled from DMS images as the water surface. In addition to the information from the optical photo (i.e., DMS images), we also use the information from InSAR coherence magnitude. Since the interferometric coherence measured over water is very low, a coherence map can be used to detect the water in the sea ice cover. All 9 points are verified with InSAR coherence magnitude below $0.3$, a threshold of open-water area mask used in (Huang and Hajnsek, 2021). Then, the heights of 9 points are averaged and subtracted from the DMS DEMs to obtain the sea ice topographic height relative to the local sea level.

These explanations have been added in the revision:

"In total, we label nine points as water-surface reference according to the DMS images. Since the interferometric coherence magnitude over water is very low, it can also be used to classify open water (Dierking et al., 2017). All the nine points have an interferometric coherence magnitude below $0.3$, which is the threshold of the open-water area mask in (Huang and Hajnsek,

2021). The average height of the open-water points is subtracted from the DMS DEMs to obtain the sea ice topographic height relative to the local sea level."

**Q10:** *Line 199: How many segments are removed vs used due to mis-coregistration? A percentage of rejected or accepted segments would be useful here.*

**A10:** The percentage of accepted segments is $76\%$, which means $12$ segments are removed due to mis-coregistration, and the rest $38$ segments are accepted for the experiments. The specific numbers have been added in the revision:

"Among the $50$ segments, $12$ segments which still contain residual mis-coregistration induced by the sea ice non-linear movement or rotation are excluded and will not be used in the following experiments. $76\%$ segments from the whole SAR scene are accepted as correctly co-registered segments in this study."

**Q11:** *Figure 8: This figure should have subplots labeled (a-f) on the figure, since they are referenced as such in the text. I agree with reviewer 1 that it is not apparent how phase centers are derived from these figures.*

**A11:** The labels (a-f) have been added in the revision. In the preprint manuscript, Fig. 8 shows the complex coherence, modelled with Eq. (12), in the unit circle. The radius corresponds to the coherence magnitude, the angular rotation to the phase. The phase can be translated to height via

$$h_{\text{volume}} = \frac{\angle \tilde{\gamma}_v}{\kappa_{\text{z\_vol}}} \tag{3}$$

where $\tilde{\gamma}_v$ can be substituted with $\tilde{\gamma}_{\text{mod\_T}}$ derived from Eq. (12) (in the preprint manuscript).

The above equation has been added in Section 2, and the following texts has been added in Section 4.3:

"The sensitivity of $\tilde{\gamma}_{\text{mod\_T}}$ to various parameters is presented in Fig. 9, where the radius and angular rotation corresponds to the coherence magnitude and phase, respectively. The phase can be translated to height via Eq. (3)."

Second, we would like to clarify that the volume-only phase centers are not directly derived from the plots but from Eq. (11) (in the preprint manuscript). The complex value of $\tilde{\gamma}_{\text{v}}(\sigma_1, z_{01})$ can be obtained according to Eq. (11) (in the preprint manuscript), and the phase part is denoted as $\angle\tilde{\gamma}_{\text{v}}(\sigma_1, z_{01})$. The derived phase can be converted to height by Eq. (3). As the range of $\sigma_1$ is $1 - 10 \, \text{db/m}$, the corresponding phase center height is calculated to be $-6$ to $-7 \, \text{cm}$. Similarly, the phase $\angle\tilde{\gamma}_{\text{v}}(\sigma_2, z_{12})$ can be obtained according to Eq. (11) (in the preprint manuscript) at different values of $\sigma_2$ and converted to height by Eq. (3). Across the range of $\sigma_2$ (i.e., $10 - 200 \, \text{db/m}$), the phase center height varies from $-15$ to $-33 \, \text{cm}$. These explanations have been added in the revision:

"The complex coherence of the snow volume $\tilde{\gamma}_{\text{v}}(\sigma_1, z_{01})$ can be calculated by Eq. (15) with thickness $z_{01} = 15 \, \text{cm}$, and its magnitude and phase can be denoted as $|\tilde{\gamma}_{\text{v}}(\sigma_1, z_{01})|$ and $\angle\tilde{\gamma}_{\text{v}}(\sigma_1, z_{01})$, respectively. Then, the phase center location of the snow volume alone can be calculated by Eq. (3). Across the range of $\sigma_1$ (i.e., $1 - 10 \, \text{db/m}$), the snow volume has an individual coherence magnitude (i.e., $|\tilde{\gamma}_{\text{v}}(\sigma_1, z_{01})|$) close to unity and phase center height varying from $-6$ to $-7 \, \text{cm}$. Similarly, the ice volume $\tilde{\gamma}_{\text{v}}(\sigma_2, z_{12})$ has an individual coherence magnitude of almost unity and a phase center height between $-15$ to $-33 \, \text{cm}$ for the investigated range of ice extinction coefficients."

**Q12:** *Line 393: Similar to above points, what percentage of pixels are processed (i.e. heights above 1.5m) vs not? With a scene-average height of 1.27m along the DMS DEM (line 494), I suspect that a large portion has been removed.*

**A12:** As we explained in A2, in the revision, pixels with height above $0.8 \, \text{m}$ have been processed into the model, which take up $83\%$ of the total co-registered pixels. This is added in the revision:

"In order to select the ice that is deformed and thick without seawater flooding, the samples with height above $0.8\,\mathrm{m}$, which are $83\%$ of the co-registered data set, are selected."

**Q13:** *Line 398: I assume 18cm is the average snow depth of the whole region, including ice <1.5m? If only samples >1.5m are selected for processing (line 394) I am curious how your results would look if you were able to use snow depth on just the ice with elevation >1.5m. While I know this information may not be available, using some type of spatially varying snow depth assumption may help to constrain possible retrieved topographies.*

**A13:** Yes, $18\,\mathrm{cm}$ is the average snow depth of the whole region due to the limited spatial resolution of available snow measurements. The reply to this comment is a repetition from an earlier answer (**A4**). In the revision, we have performed the whole inversion scheme and retrieved heights with various snow depths from $-0.05\,\mathrm{m}$ to $-0.75\,\mathrm{m}$. We also have emphasized that this type of varying snow depth assumption will help to constrain possible retrieved topographies, see **A4** for details.

**Q14:** *Figure 13: It's fairly tough to see the DMS DEM in between grey lines in (b)-(d) and draw any conclusion about its agreement with the SAR data. I would recommend making the lines thinner or reducing the width of the zoomed sections, if possible, so that more of the DMS heights are shown. If inclined, a difference map (InSAR height – DMS height) would be useful to provide a more quantitative 2-D verification.*

**A14:** The width of the zoomed sections has been reduced, and Figure 13 has been updated in the revision, see Fig. 5(b)-(d) in this document. Besides, the relative retrieval bias $\epsilon = (h_{\mathrm{mod\_S}} - h_{\mathrm{DMS}})/h_{\mathrm{DMS}}$ of zoom-in area 1-3 have been included in the revision to provide a more quantitative 2-D verification, see texts and Fig. 5 (e)-(g):

"The relative retrieval bias $\epsilon$, which can be calculated as $\epsilon = (h_{\mathrm{mod\_S}} - h_{\mathrm{DMS}})/h_{\mathrm{DMS}}$, is used to quantify the retrieval accuracy. In Fig. 5(e)-(g), $\epsilon$ over Area 1-3 are below $25\%$ for most parts, whereas only a few parts, often near to the masked-out regions (pixels in transparent), present higher $\epsilon$. Note that the masked-out regions refer to water and thinner ice areas with height less than $0.8\,\mathrm{m}$. The averaged $\epsilon$ over Area 1-3 are about $19\%$, $14\%$, and $15\%$, respectively, and is $18\%$ for the whole image, achieving the theoretical $25\%$-error accuracy derived in Section 4.4."

**Q15:** *Figure 13 also: How are heights less than 1.5m calculated in this SAR image if not selected for processing with this model? Subplot (d) in particular appears to have regions of 0m height that I suspect are not entirely physical.*

**A15:** In the revision, the height threshold has been changed to be $0.8\,\mathrm{m}$, and the height below $0.8\,\mathrm{m}$ have been masked out. Therefore, in the updated retrieval map (see Fig. 5 in this document), the retrieved height and DMS measurements below $0.8\,\mathrm{m}$ have been set to be transparent and would not be considered in the analyses.

**Q16:** *Line 456: I would suggest clarifying that $h_{Model}$ in this case is the simplified model. While I understand it is in the "simplified model" section, to me the third row in Table 1 is the Theoretical model row (as it is the third method). The fact that the RMSE ranges between 0.22 and 0.27 for both models further adds to the confusion.*

**A16:** Sorry for the confusion. The statements have been modified as suggested in the revision:

"Table 1 summarizes the performances between the retrieved height from the simplified model and $h_{\mathrm{DMS}}$ for the four polarizations."

**Q17:** *Line 487: This line (particularly "larger baselines respectively larger kz values") doesn't quite sound correct as written. Do you perhaps mean the possessive "baselines"?*

**A17:** The larger baseline means a larger value of the effective perpendicular baseline $b_\perp$ (Eq.4 in the preprint manuscript)

[Figure]

Fig. 5: Sea ice topographic retrieval with the simplified model ($h_{\mathrm{mod\_S}}$). The transect from the DMS DEM is plotted between grey lines. Note that the heights below $0.8\,\mathrm{m}$ are set to be transparent. (a) The whole studied SAR image. (b)-(d) Zoom-in of Areas 1-3. (e)-(g) Relative retrieval bias $\epsilon = (h_{\mathrm{mod\_S}} - h_{\mathrm{DMS}})/h_{\mathrm{DMS}}$ of Area 1-3.

in the interferometric SAR configuration. It has been clarified in the revision:

"It reveals the potential to establish an inversion scheme by combining observations from a range of different $\kappa_z$ (i.e., the vertical wavenumber in free space), where larger values of the effective perpendicular baseline $b_\perp$, corresponding to larger $\kappa_z$ values, are expected to improve height retrieval accuracy."

**Q18:** *Line 499: Should be "25%-error accuracy" to be consistent with previous sections*

**A18:** It has been corrected to be "25%-error accuracy".

**Q19:** *Technical Corrections:*

- *Line 59: Icebridge -> IceBridge*

- *Line 143: iceberg -> icebergs*

- *Figure 1 caption: rectangular -> rectangle*

- *Line 215: 'flat-earth removed' should be written as 'flat-earth-removed'*

- *Line 254: Provide full names of TDF and TSX at first mention*

- *Line 470: (grammar) well correct -> e.g. adequately/sufficiently/suitably correct*

- *Line 501: comma after "For instance"*

**A19:** All above points haven been corrected in the revision.

Again, we sincerely thank the editor and reviewers for helping us improving the manuscript.

---

## Author Response (AR1)

**Reply to the comments**

**I. REPLY TO THE EDITOR**

Dear Dr. Haas,

Thank you very much for handling our manuscript! We have carefully revised the manuscript following the suggestions. Hereby we give a point-by-point reply to address the comments. In this document, the words in *italics are the reviewers' comments*, the words in blue are the modifications we have made in the revision, and others are our responses. Besides adjustments requested by the reviewers, we have checked the manuscript carefully for typos and conciseness, and have marked all the modifications in the revised manuscript.

Sincerely,

Lanqing Huang,

on behalf of all the authors

**II. REPLY TO THE COMMENTS OF REVIEWER 1**

**R1Q1:** *Summary. This is an interesting, detailed, well prepared paper that has important implications for deriving sea ice topography using a unique approach, with single-pass interferometry. I have some requests on clarifications that I will write down in Details. The methods are largely clearly described. I think I have three main points about the paper as a whole that I suggest the authors consider.*

**R1A1:** We thank the reviewer for the positive comment about our research. We have carefully revised the manuscript based on the following comments.

**R1Q2:** *First, the paper is presented as being applicable to both polar sea ice covers. However, particularly in the Introduction and Basic Concepts, they really don't distinguish sufficiently between the main differences in the sea ice between the Arctic and Antarctic. I have added some suggested references to include. So please add some more text about differences in ice type and snow layer.*

**R1A2:** We have included all suggested references. Specifically, more discussions about the main differences of ridges between the Arctic and Antarctic have been added in the Introduction:

"Timco and Burden, (1997) estimated the ratio of the keel-depth (i.e., depth of ice below the seawater) to sail-height (i.e., height of ice above the seawater) for both first-year and multi-year sea ice ridges in the Beaufort Sea and highlighted their differences in ridge height and shape. Haas et al. (1999) presented the pressure ridge frequencies to be $3 - 30$ ridges per kilometre over Bellingshausen, Amundsen, and Weddell Seas in Antarctica. Tin and Jeffries (2003) indicated that first-year ridges in the Antarctic are flatter and less massive than those in the Arctic. Sea ice ridging height is a crucial parameter to evaluate total ice mass in both polar regions (Hibler et al., 1974; Melling and Riedel, 1995; Lytle et al., 1998; Tin et al., 2003). In the Antarctic, the mean height of the ridges in the Weddell Sea was found to be $\sim 1.1\,\mathrm{m}$ which is similar to the ridging statistics from the Ross Sea (Lytle and Ackley, 1991), whereas it is considerably less than the height from the Arctic (Lytle and Ackley, 1991; Dierking, 1995)."

The differences of snow layer and sea ice properties between Arctic and Antarctic have been added in a new paragraph in the Introduction:

"The snow layer and ice properties in the Arctic and Antarctic are significantly different due to the diverse growing conditions in the two polar regions (Gloersen, 1992; Walsh, 2009; Sturm and Massom, 2009; Webster et al., 2018). In the Antarctic, the snow depth is reported to be thicker than in the Arctic (Jeffries et al., 1997; Massom et al., 2001; Willatt et al., 2009). When thick enough, the snow will overburden the ice floe and be flooded by seawater, resulting in higher salinity of the snow layer in the Antarctic. Besides, compared to the Arctic, snow on the Antarctic sea ice comprises more heterogeneous layers resulting from highly variable temperature (Massom et al., 2001). The layer heterogeneity in types, density, salinity, and wetness would determines the electromagnetic characteristics of the snow. As for ice properties in general, Antarctic sea ice is reported to be thinner (Worbyet al., 2008; Kurtz and Markus, 2012; Lindsay and Schweiger, 2015), younger (Webster et al., 2018), and more saline than in the Arctic at comparable age and thickness (Gow et al., 1982). Quantitatively, the mean salinity of the Antarctic first-year ice and multi-year ice profiles are $4.6‰$ and $3.5‰$ , respectively, whereas the average values are $3‰$ for the first-year ice and $2 - 2.5‰$ for the multi-year ice in the Arctic (Coxand Weeks, 1974). These variable properties of sea ice, its ridges and snow cover, at both small and large spatial scales, highlight the challenge and necessity for accurate sea ice topographic information with large spatial coverage and high resolution."

**R1Q3:** *Next, it's not clear to me that a two-layer model is sufficient, with both layers considered to be uniform, to correctly identify the phase center. In Arctic first year ice, at the snow-ice surface particularly for young first year there is often a significant layer of high salinity slush ice that may also include frost flowers. There wil be some penetration below this thin layer where the salinity is much lower. Then of course in the Antarctic, flooding at the snow-ice layer occurs due to relatively deeper snow loading on the generally thinner ice layer, as compared to the Arctic. This flooded layer has a higher salinity than the ice below but is still not likely sufficient to minimize further penetration. Plus of course there is increase in salinity near the ice-ocean boundary in all winter ice growth conditions. The slush layer is referred to in the paper, but I maintain that it is not sufficient to dismiss the possibility of a 3-layer without demonstrating otherwise which I suggest they do, as it may impact correct estimation of the phase center and therefore a modeled-derived height.*

**R1A3:** We agree that the slush layer is important for Arctic first-year ice and Antarctic flooded sea ice. First, in the revision, we have added more references in Section 1 (Introduction), introducing the slush layer in both polar regions:

"In the Arctic first-year thin ice, snow capillary force gives rise to brine wicking, and consequently, a layer of high salinity slush ice appears at the snow-ice interface (Reimnitz and Kempema, 1987; Drinkwater and Crocker, 1988; Nghiem et al., 1995a). In the Antarctic, ice-surface flooding widely occurs resulting from the generally thicker snow layer loading on the thinner ice floes, often followed by freezing of the slush layer at the snow-ice interface (Massom et al., 2001; Jeffrieset al., 2001; Maksym and Jeffries, 2000). Even without flooding, the upward wicking of brine from the ice surface can also form a saline layer at the bottom of the snowpack (Massom et al., 2001; Toyota et al., 2011; Webster et al., 2018). The slush layer at the snow-ice interface would induce significant surface scattering and thus has been included in the sea ice scattering modelling (Nghiem et al., 1995a, b; Maksym and Jeffries, 2000)."

Focusing on the Antarctic sea ice, we have added more explanations about the slush layer for both thinner and thicker ice conditions in Section 2 (Basic concepts). Different sea ice structures for thinner ice and thicker ice have been plotted in Fig. 1(a) and (b), respectively, in this document. We have also emphasized that in this study, the experiments are conducted based on the thicker and deformed ice with height $> 0.8\,\mathrm{m}$ above the sea level. This height threshold is estimated from hydrostatic balance, and we assume that sea ice higher than this threshold do not suffer seawater flooding. Note that even without flooding, the upward wicking of brine from the ice surface can also form a thin and high saline layer at snow-ice interface (Massom

et al., 2001; Toyota et al., 2011; Webster et al., 2018). Therefore, the surface scattering from the slush layer at the snow-ice interface has been considered in the model. Below Fig. 1 and texts have been added in the new Section 2.1:

[Figure]

Fig. 1: Schematic of (a) thinner ice floes flooded by seawater and (b) thicker ice floes without flooding.

"In the Antarctic, the presence of a saline layer at the snow-ice interface due to the flooding or capillary suction of brine from the ice surface has been recognized as a widespread and critical phenomenon (Massom et al., 2001). For thinner ice, flooding may occur when the weight of the snow pushes the ice surface below the water level, yielding a negative freeboard. In this case, as shown in Fig. 1(a), seawater infiltrates into the snowpack, floods the ice surface, and creates a high-saline slush layer which may refreeze into snow ice (Lange et al., 1990; Jeffries et al., 1997; Maksym and Jeffries, 2000). The thickness of snow ice was observed to be $\sim 42 - 70\%$ of the total snow accumulation (i.e., the thickness of snow ice plus snow depth) (Jeffries et al., 2001).

For thicker and deformed ice with ridges, less flooding occurs due to the increased buoyancy of the ice mass contained in the ridges (Jeffries et al., 1998). However, even in the absence of flooding, a thin slush layer can also occur due to the capillary suction of brine from the ice surface (Massom et al., 2001; Webster et al., 2018). Besides, the deformed ice in the ridging and rafting area is often poorly consolidated, and thus seawater may reach the snow layer and form a thin slush layer (Maksymand Jeffries, 2000). The sea ice structure for thicker ice without flooding is sketched in Fig. 1(b), including snow on top, the ice volume, and a thin and high-saline layer in between.

The condition of flooding can be quantified by a simple hydrostatic balance (Lange et al., 1990)

$$\rho_w d = \rho_i d + \rho_i f + \rho_s s$$
$$t = d + f \tag{1}$$

where $\rho_w$ is the seawater density, $\rho_i$ is the ice density, and $\rho_s$ is the snow density; $d$ and $f$ are the thickness of ice below and above the sea level, respectively; $t$ is the thickness of the total ice thickness, and $s$ is the snow depth. In case of a flooding, $f$ should be zero (i.e., $d = t$), and Eq. (1) becomes

$$s/t = (\rho_w - \rho_i)/\rho_s \approx 0.12/\rho_s \tag{2}$$

by assuming $\rho_w = 1.03\,\mathrm{Mg/m^{-3}}$ and $\rho_i = 0.91\,\mathrm{Mg/m^{-3}}$ (Lange et al., 1990). For snow density $\rho_s$ being $0.3\,\mathrm{Mg/m^{-3}}$ (Lange et al., 1990), the ratio between snow depth and ice thickness $s/t$ is estimated to be $0.4$. The snow depth on Antarctic sea ice during September and November was shown to be below $0.8\,\mathrm{m}$ for $99\%$ of the samples in Webster et al., (2018). This range of snow depth will lead to flooding for ice thickness $< 2\,\mathrm{m}$.

The relation between ice thickness $H_i$ and surface height $h_\mathrm{sur}$ (i.e., ice height above sea surface including snow depth) has been discussed over different regions (Petty et al., 2016; Toyota et al., 2011; Ozsoy-Cicek et al., 2013). Ozsoy-Cicek et al., (2013) showed a linear relation $H_i = 2.24 h_\mathrm{sur} + 0.228$ fitted from large-scale, survey-averaged data over the Western Weddell Sea, which is the same region as this study. According to this linear relation, $H_i = 2\mathrm{m}$ corresponds to a surface height of $\sim 0.8\,\mathrm{m}$.

This paper focuses on thicker ($> 2\,\mathrm{m}$) and deformed ice (Fig. 1(b)), which is the main ice typology in the studied area. In the following sections, the model and experiments are conducted only for the samples above $\sim 0.8\,\mathrm{m}$ surface height. We assume that samples exceeding this threshold are thicker and deformed ice without flooding. The potential to extend the proposed model to thinner ice scenarios (e.g. Fig. 1 (a)) is discussed in Section 7.3."

The Figure: Schematic of the proposed two-layer plus volume model for sea ice (Fig. 7 in the revised manuscript), has been updated with below Fig. 2 and some sentences have been added in Section 4.1:

[Figure]

Fig. 2: Schematic of the proposed two-layer plus volume model for the thicker and deformed sea ice.

"The top layer located at $z_1$ is the snow-ice interface, which can induce significant surface scattering due to a slush layer with high permittivity (Hallikainen and Winebrenner, 1992; Maksym and Jeffries, 2000). This slush layer is widespread on the Antarctic sea ice, and increases the radar backscattering as well as limits the signal penetration compared to a smooth and dry snow-ice interface. As long as the slush layer has a small vertical extent, it is irrelevant for the Pol-InSAR scattering structure model, whether the top layer represents the snow-ice interface, the snow-slush interface or both."

To extend the proposed model for thinner ice where flooding often occurs, an additional layer, i.e., snow ice formed from the slush layer when air is cold, has been discussed. A three-volume model incorporated with the snow ice could be a promising approach to correct the InSAR phase. Below texts has been added in the new discussion Section 7.3:

"The proposed model was proven to be effective in a specific area covered by thick and deformed ice with snow cover in the Western Weddell Sea. The extension of the proposed model to other ice types under different environmental conditions needs further research and suitable data.

In order to apply the model over younger and thinner sea ice, the first challenge is the severe misregistration between SAR images and reference measurements due to the stronger dynamics of thinner sea ice. Reduced SAR backscattering intensity corresponding to thinner and smoother sea ice further complicates the data co-registration. Besides, the achievable height sensitivity for thin ice is also a major limitation of InSAR/Pol-InSAR derived sea ice DEMs with current SAR systems. In this study, the proposed method can achieve sea ice topographic retrieval with an $RMSE$ of $0.26\,\mathrm{m}$ for thick and deformed ice; however, this accuracy is insufficient for thinner ice whose height above sea level is only tens-of-centimeters or even less. Last but not least, an additional volume, i.e., snow ice formed by flooding, should be considered when extending the proposed model to a thinner ice area. Past studies showed that snow ice contributes an average of $8\%$ of the total volume in the Weddell Sea (Lange et al., 1990). A greater amount of snow ice, which accounts for $12-36\%$ of the total mass, was reported in the Ross, Amundsen, and Bellingshausen Seas (Jeffries et al., 2001). Although the snow ice has a higher salinity than the ice below, there could be still some penetration into the ice volume below. A three-layer model, which includes snow, snow ice, and ice layers, could be feasible to correct the InSAR phase center and retrieve surface height for thinner ice in the Antarctic (as illustrated in Fig. 1 (a)). However, the three-layer model involving more parameters than the proposed theoretical model can only be inverted by increasing the observation space to full polarization and multi-baseline configurations. In the case of single-baseline or dual-polarization configurations where observables are limited, an intelligent model with fewer parameters is worthy of investigation in the future."

**R1Q4:** *Finally, last main point has two components – first while they compare with a DMS height as a narrow 2D transect, which is what is available and appropriate, they also show 3D output (Figure 13). However, there isn't much discussion about the 3d output – do these appear to be representative of what might be expected or compared to other possible data or studies? There are papers on ridge/sail characteristics plus a nice example in Tucker chapter 2 in Carsey sea ice microwave book. Second part of last point is that these 3D maps are really unique because they are not just narrow transects. How often for example could these 3D maps be generated with Tandem-X, spatially and temporally? I think Tandem-X is pretty limited by its duty cycle at least and perhaps storage/downlink too. It would be good to hear about longer term capabilities for deriving this product and what might be required to validate Arctic products, for example.*

**R1A4:** We thank the reviewer for recommending Tucker chapter 2 in Carsey sea ice microwave book. We agree it is a great idea to extract ridge/sail characteristics from the 3D output, which is a distinct advantage of SAR out of other measurements. We have worked on these for some months. The statistical features extracted from SAR-retrieved topographic map have been analyzed and related to the Antarctic geophysical environments. We are now preparing another manuscript to present these interesting results. In the revision, based on the 3D output, we have added more discussions on the relative retrieval bias $\epsilon = |h_{\mathrm{mod\_S}} - h_{\mathrm{DMS}}|/h_{\mathrm{DMS}}$ of zoom-in area 1-3. Below texts and Fig. 3 (e)-(g) have been added in Section 6.1:

"The relative retrieval bias $\epsilon$, which can be calculated as $\epsilon = |h_{\mathrm{mod\_S}} - h_{\mathrm{DMS}}|/h_{\mathrm{DMS}}$, is used to quantify the retrieval accuracy. In Fig. 3(e)-(g), $\epsilon$ over area 1-3 are below $25\%$ for most parts, whereas only a few parts, often near to the masked-out regions (transparent pixels), present higher $\epsilon$. Note that the masked-out regions refer to water and thinner ice areas with height less than $0.8\,\mathrm{m}$. The averaged $\epsilon$ over area 1-3 are about $19\%$, $14\%$, and $15\%$, respectively, and is $18\%$ for the whole image, achieving the theoretical $25\%$-error accuracy derived in Section 4.3."

Besides, we have added more discussions regarding the spatial coverage and temporal resolution for the unique InSAR-derived 3D map. The long-term capability of SAR sensor for sea ice monitoring and the requirement to validate Arctic products have been included in the new discussion Section 7.3:

[Figure]

Fig. 3: Sea ice topographic retrieval with the simplified model ($h_{\mathrm{mod\_S}}$). The transect from the DMS DEM is plotted between grey lines. Note that the heights below $0.8\,\mathrm{m}$ are set to be transparent. (a) The whole studied SAR image. (b)-(d) Zoom-in of areas 1-3. (e)-(g) Relative retrieval bias $\epsilon = |h_{\mathrm{mod\_S}} - h_{\mathrm{DMS}}|/h_{\mathrm{DMS}}$ of area 1-3.

"In this study, the InSAR pair in StripMap mode covers $19\,\mathrm{km} \times 50\,\mathrm{km}$ in SAR ground range and azimuth direction, respectively, providing a unique 3-dimensional (3D) topographic map rather than a narrow transect from LIDAR or photogrammetric measurements. TanDEM-X has a regular revisit cycle of 11 days over the Arctic and a larger revisit time due to the particular satellite position configuration required over Antarctica. The current SAR satellites, such as X-band TanDEM-X and COSMO-SkyMed, C-band Sentinel-1 and Radarsat Constellation, as well as the future X-band LOTUSat, L/S-band NISAR, and L-band ROSE-L, will together achieve a long-term sea ice topographic monitoring in both polar regions. Synergistic use of different

SAR satellites offers more extensive spatial coverage and shorter revisit times than a single platform. In the future, the joint use of multi-frequency SAR imagery could develop a better understanding of sea ice properties and processes (Dierking and Davidson, 2021), which would be indispensable for retrieving sea ice topography at a more comprehensive range of ice conditions.

In order to assess the transferability of the proposed model to the Arctic regions, further validations, including co-registered SAR images and topographic reference (e.g., optical/LIDAR measurements), are needed considering the significant difference of ice and snow properties between the Arctic and Antarctic. Ancillary measurements (e.g., snow depth, temperature, ice and snow salinity) at a wide ranger of ice conditions in both polar regions are crucial to understand the properties of various typologies of ice, and therefore are valuable for extending the model to general applicability. Part of the ancillary data (i.e., snow depth and ice freeboard height) would be available in OTASC Level-4 products in the future, offering us an opportunity to interpret sea ice electromagnetic properties."

**R1Q5:** *Lines16-18. Add references that discuss ridge characteristics, etc in addition to Rampal reference for both poles, for example for Antarctic, Lytle et al. Annals Glaciology 1998, two Tin and Jeffries papers in 2003/04, plus Timco and Burden 1997 for Arctic.*

**R1A5:** The references that discuss ridge characteristics in both polar regions have been added as suggested in the Introduction:

"Timco and Burden, (1997) estimated the ratio of the keel-depth (i.e., depth of ice below the seawater) to sail-height (i.e., height of ice above the seawater) for both first-year and multi-year sea ice ridges in the Beaufort Sea and highlighted their differences in ridge height and shape. Haas et al. (1999) presented the pressure ridge frequencies to be $3 - 30$ ridges per kilometre over Bellingshausen, Amundsen, and Weddell Seas in Antarctica. Tin and Jeffries (2003) indicated that first-year ridges in the Antarctic are flatter and less massive than those in the Arctic. Sea ice ridging height is a crucial parameter to evaluate total ice mass in both polar regions (Hibler et al., 1974; Melling and Riedel, 1995; Lytle et al., 1998; Tin et al., 2003). In the Antarctic, the mean height of the ridges in the Weddell Sea was found to be $\sim 1.1\,\mathrm{m}$ which is similar to the ridging statistics from the Ross Sea (Lytle and Ackley, 1991), whereas it is considerably less than the height from the Arctic (Lytle and Ackley, 1991; Dierking, 1995)."

**R1Q6:** *Line 24, Tucker et al reference is for first year ice only. please clarify.*

**R1A6:** It has been clarified in the revision (Introduction):

"Focusing on first-year sea ice in the Alaska region, Tucker and Govoni (1981) observed a square-root relation between the ridge height and thickness, which is further validated by additional in situ observations in (Tucker et al., 1984)."

**R1Q7:** *Line 25. Petty reference discusses both FY and MY and differences. Please mention in text. Also following Toyota paper, there is a really good chapter on Snow by Sturm and Massom in Sea Ice book edited by Thomas and a recent chapter by Webster et al Nature Climate Change 2018.*

**R1A7:** In the Introduction, we have added that Petty's reference discusses both FY and MY and differences:

"Petty et al. (2016) presented a detailed characterization of Arctic sea ice topography across both first-year and multi-year sea ice and analyzed the topographic differences between the two ice regimes."

As suggested, the references have been added:

"The snow layer and ice properties in the Arctic and Antarctic are significantly different due to the diverse growing conditions in the two polar regions (Gloersen, 1992; Walsh, 2009; Sturm and Massom, 2009; Webster et al., 2018)."

**R1Q8:** *Lines 30-31. Add journal papers that utilize DMS data in addition to Dotson and Aversen references.*

**R1A8:** We have add an important conference paper (Nghiem et al.,2018) from OTASC team. Another journal paper using both DMS and SAR data for iceberg topographic retrieval (Dammann et al., 2019) was given in Section 3.1.

- Nghiem, S., Busche, T., Kraus, T., Bachmann, M., Kurtz, N., Sonntag, J., Woods, J., Ackley, S., Xie, H., Maksym, T., et al.: Remote Sensing of Antarctic Sea Ice with Coordinated Aircraft and Satellite Data Acquisitions, in: Proc. IGARSS., pp. 8531–8534, IEEE,https://doi.org/10.1109/IGARSS.2018.8518550, 2018.

- Dammann, D. O., Eriksson, L. E. B., Nghiem, S. V., Pettit, E. C., Kurtz, N. T., Sonntag, J. G., Busche, T. E., Meyer, F. J., and Mahoney, A. R.: Iceberg topography and volume classification using TanDEM-X interferometry, The Cryosphere, 13, 1861–1875,https://doi.org/10.5194/tc-13-1861-2019, 2019.

In the revision, the references has been updated:

"The sea ice topography can be measured by various instruments, such as laser altimeters (Dierking, 1995; Schutz et al.,2005; Abdalati et al., 2010; Farrell et al., 2011, 2020) and stereo cameras using photogrammetric techniques (Dotson and Arvesen., 2012, updated 2014; Divine et al., 2016; Nghiem et al., 2018; Li et al., 2019)."

**R1Q9:** *Line 53. Substitute 'deficient brine' for 'reduced brine'*

**R1A9:** It has been changed as suggested.

**R1Q10:** *Line 61. Substitute 'obtain an' for 'obtain a more' accurate.*

**R1A10:** It has been changed as suggested.

**R1Q11:** *Lines 77-83. This paragraph should be expanded to discuss thin salinity layers at snowice interface as mentioned in the summary with references*

**R1A11:** The paragraph has been expanded by adding more discussions of the high salinity layer at the snow-ice interface in the Introduction:

"In the Arctic first-year thin ice, snow capillary force gives rise to brine wicking, and consequently, a layer of high salinity slush ice appears at the snow-ice interface (Reimnitz and Kempema, 1987; Drinkwater and Crocker, 1988; Nghiem et al., 1995a). In the Antarctic, ice-surface flooding widely occurs resulting from the generally thicker snow layer loading on the thinner ice floes, often followed by freezing of the slush layer at the snow-ice interface (Massom et al., 2001; Jeffrieset al., 2001; Maksym and Jeffries, 2000). Even without flooding, the upward wicking of brine from the ice surface can also form a saline layer at the bottom of the snowpack (Massom et al., 2001; Toyota et al., 2011; Webster et al., 2018). The slush layer at the snow-ice interface would induce significant surface scattering and thus has been included in the sea ice scattering modelling (Nghiem et al., 1995a, b; Maksym and Jeffries, 2000)."

**R1Q12:** *Figure 8. I guess I really don't understand these figures. I looked and looked at how one might determine that these graphs suggest phase centers of 6-7cm and 15-33 cm as described in the text – Lines 312-315. I would appreciate an explanation of what information they are using from these figures and how they are deriving the phase centers. I also hope the editors are getting a review from another person who has a lot more INSAR and radar modeling expertise than me.*

**R1A12:** Sorry for the unclear statement. In the preprint manuscript, Fig. 8 shows the complex coherence, modelled with Eq. (12), in the unit circle. The radius corresponds to the coherence magnitude, the angular rotation to the phase. The phase can be translated to height via

$$h_{\text{volume}} = \frac{\angle \tilde{\gamma}_v}{\kappa_{\text{z\_vol}}} \tag{3}$$

where $\tilde{\gamma}_v$ can be substituted with $\tilde{\gamma}_{\mathrm{mod\_T}}$ derived from Eq. (12) (in the preprint manuscript).

The above equation has been added in Section 2, and the following texts has been added in Section 4.2:

"The sensitivity of $\tilde{\gamma}_{\mathrm{mod\_T}}$ to various parameters is presented in Fig. 9, where the radius and angular rotation corresponds to the coherence magnitude and phase, respectively. The phase can be translated to height via Eq. (3)."

Second, we would like to clarify that the volume-only phase centers are not directly derived from the plots but from Eq. (15) (in the revision). The complex value of $\tilde{\gamma}_{\mathrm{v}}(\sigma_1, z_{01})$ can be obtained according to Eq. (15) (in the revision), and the phase part is denoted as $\angle\tilde{\gamma}_{\mathrm{v}}(\sigma_1, z_{01})$. The derived phase can be converted to height by Eq. (3). As the range of $\sigma_1$ is $1 - 10\,\mathrm{db/m}$, the corresponding phase center height is calculated to be $-6$ to $-7\,\mathrm{cm}$. Similarly, the phase $\angle\tilde{\gamma}_{\mathrm{v}}(\sigma_2, z_{12})$ can be obtained according to Eq. (15) (in the revision) at different values of $\sigma_2$ and converted to height by Eq. (3). Across the range of $\sigma_2$ (i.e., $10 - 200\,\mathrm{db/m}$), the phase center height varies from $-15$ to $-33\,\mathrm{cm}$. These explanations have been added in Section 4.2:

"The complex coherence of the snow volume $\tilde{\gamma}_{\mathrm{v}}(\sigma_1, z_{01})$ can be calculated by Eq. (15) with thickness $z_{01} = 15\,\mathrm{cm}$, and its magnitude and phase can be denoted as $|\tilde{\gamma}_{\mathrm{v}}(\sigma_1, z_{01})|$ and $\angle\tilde{\gamma}_{\mathrm{v}}(\sigma_1, z_{01})$, respectively. Then, the phase center location of the snow volume alone can be calculated by Eq. (3). Across the range of $\sigma_1$ (i.e., $1 - 10\,\mathrm{db/m}$), the snow volume has an individual coherence magnitude (i.e., $|\tilde{\gamma}_{\mathrm{v}}(\sigma_1, z_{01})|$) close to unity and phase center height varying from $-6$ to $-7\,\mathrm{cm}$. Similarly, the ice volume $\tilde{\gamma}_{\mathrm{v}}(\sigma_2, z_{12})$ has an individual coherence magnitude of almost unity and a phase center height between $-15$ to $-33\,\mathrm{cm}$ for the investigated range of ice extinction coefficients."

**R1Q13:** *Figure 14. I think the grey lines for the removed sections are too distracting from both a) and b) and perhaps just not included in the graph. Are the grey removed sections the same segments as described in Figure 4 as being mis-registered and set at 0 height? If so, not sure why they need to be included in Figure 14 at all.*

**R1A13:** As explained in R1A3, in this study, we focus on the snow-covered sea ice thicker than $\sim 2\,\mathrm{m}$, corresponding to $0.8\,\mathrm{m}$ height above seawater. Therefore, the mis-registered segments and pixels with DMS height below $0.8\,\mathrm{m}$ were removed. In the revision, all these removed pixels have been set to be NaN values, and the plot (Fig. 14 in the revision) is updated without grey lines (see Fig. 4 below):

**R1Q14:** *Lines 474. Snow depth is not well correlated with thickness or even FY or MY ice and in both polar regions.*

**R1A14:** In this study, snow depth is assumed as a constant value across the scene due to the limited spatial resolution of available snow measurements. We admit that this single value is likely not representative of the actual spatial snow depth distribution, which is related to ice thickness and ice condition. Therefore, in the revision, we have performed the whole inversion scheme and retrieved heights assuming various $z_1$ values from $-0.05\,\mathrm{m}$ to $-0.75\,\mathrm{m}$, according to the snow depth on Antarctic sea ice reported in Webster et al., (2018). Using varying snow depths, we have demonstrated how the retrieved height would agree with the DMS DEM and discussed the impact of the snow depth parameter on the experimental results. Below texts and figures have been added in the new Section 7.2:

"In Section 4.3, we demonstrated that the influence of snow depth on the simulated coherences is not negligible, and stated that external data of snow measurements should be used in the model. In this study, snow depth is assumed to be invariant across the scene due to the limited spatial resolution of available snow measurements. Therefore, a constant value of $z_1 = -0.18$ is used in the retrieval. Actually, snow on sea ice undergoes temporal- and spatial-variant processes and is strongly coupled with atmospheric, oceanic, and ice conditions. Thus, a single value is not representative of the actual spatial snow depth distribution. In order to assess the impact of the snow depth on the experimental results, we perform the whole inversion scheme with various inputs of $z_1$. During September and November, the snow depth on Antarctic sea ice is reported to be maximum $\sim 1\,\mathrm{m}$

[Figure]

Fig. 4: (a) Sea ice height profiles from DMS measurement (blue) and model (red). Each profile represents the height along a $1 \times 5000$-pixel section at the center of the co-registered segment. (b) Height difference between the DMS measurement and the simplified model-derived height (blue), theoretical model-derived height (red), or original InSAR-derived height (green). The mis-coregistered and $h_{\mathrm{DMS}}$ below $0.8\,\mathrm{m}$ samples are excluded from the plots.

and mainly between $0$ and $0.8\,\mathrm{m}$ (Webster et al., 2018). Therefore, $z_1$ values ranging from $-0.05\,\mathrm{m}$ to $-0.75\,\mathrm{m}$ are selected. For each pixel, we retrieve heights using this range of $z_1$ values, shown as the yellow area in Fig. 5. $\Delta h_{\mathrm{mod\_S}}$ is defined as the difference between the maximum and the minimum retrieved height of every pixel. The distribution of $\Delta h_{\mathrm{mod\_S}}$ along the transect is presented in Fig. 6, where $\Delta h_{\mathrm{mod\_S}}$ has a range of $0.07-1.09\,\mathrm{m}$ with an average of $0.31\,\mathrm{m}$, indicating the fluctuation of model-retrieved height by using different snow depths. This analysis with various snow depth assumptions can help to constrain possible model-retrieved topographies, and $\Delta h_{\mathrm{mod\_S}}$ can be a quantitative indicator for the uncertainty of the retrieved height in the absence of high-resolution snow depth data."

[Figure]

Fig. 5: Yellow area: sea ice height profiles from the simplified model in Pauli-1 polarization with $z_1$ from $-0.05\,\mathrm{m}$ to $-0.75\,\mathrm{m}$. Blue dash line: sea ice height profiles from DMS measurement. Each profile represents the height along a $1 \times 5000$-pixel section at the center of the co-registered segment. The mis-coregistered and $h_{\mathrm{DMS}}$ below $0.8\,\mathrm{m}$ samples are excluded from the plot.

**R1Q15:** *Lines 509-511. This is a good sentence and touches back at some of my points in the summary and the need to clarify some references in the introduction and Basic Concept section, the idea of model that may need to improvement etc as*

[Figure]

Fig. 6: The distribution of $\Delta h_{\mathrm{mod\_S}}$ along the transect.

*well as how these type of products could be expanded in Tandem-X acquisitions and products.*

**R1A15:** The reply to this comment is a repetition from earlier answers (**R1A2 and R1A3**). Specifically, we have clarified more references in the introduction about the different characteristics of snow and ice in the Arctic and Antarctic, see **R1A2**. The idea of the model and how the model can be extended to other ice conditions have been further discussed in the revision, see **R1A3**.

**R1Q16:** *Basic concepts or Model Development or Discussion. I really do think it's important to consider a third thin high salinity layer at the snow-ice interface, whether on thin first year ice or flooded ice. I realize this might be a lot of extra work and at this stage of your study, it may not be of primary importance. This could also be added to the Model section or Discussion section too at minimum, as a topic for further research and what you think the impact might be on the model. Of course, the authors could tell me that they don't think it's a worthy topic at all and won't make any difference. I do think firmly that their two-layer is not universally applicable to all the major ice types and conditions for both polar regions, based on my understanding of their model. New and young ice are often the trickiest anyway to deal with any radar algorithm. Throughout the paper as I was reviewing it, I kept thinking about those two thin salinity layers and differences between first year and multiyear etc in both poles and how this should all be considered in a model of radar penetrating sea ice.*

**R1A16:** The reply to this comment is a repetition from earlier answers (**R1A3**). We agree with the review that a third thin high salinity layer at the snow-ice interface, whether on thin first year ice or flooded ice, is important. The discussions of slush layer have been added in Section 1 (Introduction), Section 2 (Basic conception), and the new Section 7 (Discussion), see **R1A3** for details.

**III. REPLY TO THE COMMENTS OF REVIEWER 2**

**R2Q1:** *The manuscript Antarctic snow-covered sea ice topography derivation from TanDEM-X using polarimetric SAR interferometry by Huang et al. presents the development and validation of a new two-layer plus volume sea ice model with the aim to correct for the height bias associated with InSAR penetration into the snow pack. This model is able to represent the*

*sea ice/snow stratigraphy and associated scattering, and, when simplified and inverted, allows for the estimation of the sea ice plus snow surface topography from TanDEM-X. This retrieval technique shows strong agreement to an Operation IceBridge optical (DMS) DEM that was collected contemporaneously as part of the OIB/TanDEM-X Coordinated Science Campaign. This manuscript is well-written and thoroughly presents novel methods and results that could be useful to the broader sea ice community. I have a few relatively minor comments and suggestions that should be considered, found in the general and specific comments below. The main comments I have on the manuscript deal with (1) the height threshold used (2) X-band scattering/slush layers and (3) the snow depth parameter.*

**R2A1:** We thank the reviewer for the positive comment about our research. We have carefully revised the manuscript based on the following comments.

**R2Q2:** *GC1: To me, it appears there is some mix-up with the height threshold used to keep model-error accuracy to within 25volume (z1-z2) needs to be thicker than 1.5m to achieve this accuracy. However, in later sections only ice+snow heights above the local sea surface (effectively the total freeboard) above 1.5m are used. Doing so filters out ice volumes much thicker than 1.5m, since most of the ice volume is below the waterline. I would suggest the authors confirm that the 1.5m threshold is indeed for the ice volume, and recommend that they filter the InSAR retrieved heights accordingly (which should result in a much lower height-above-sea-surface threshold).*

**R2A2:** We confirm that the $\sim 1.5\,\mathrm{m}$ threshold is for the ice volume to achieve a $\leq 25\%$-error inversion accuracy. In the revision, the applied height threshold was change to $0.8\,\mathrm{m}$ in order to select ice that is deformed and thick without seawater flooding (see Fig. 1(b)). In this case, ice thickness should exceed $2\,\mathrm{m}$ (see details in **R1A3**), corresponding to surface height of $\sim 0.8\,\mathrm{m}$ (Ozsoy-Cicek et al., 2013). Therefore, the samples with height above $0.8\,\mathrm{m}$ are selected for processing. In the revision, experimental parts have been updated as suggested, and the sentences below have been added in the new Section 2.1:

"The relation between ice thickness $H_i$ and surface height $h_{\mathrm{sur}}$ (i.e., ice height above sea surface including snow depth) has been discussed over different regions (Petty et al., 2016; Toyota et al., 2011; Ozsoy-Cicek et al., 2013). Ozsoy-Cicek et al., (2013) showed a linear relation $H_i = c_0 h_{\mathrm{sur}} + c_1$ with $c_0 = 2.24$ and $c_1 = 0.228$ fitted from large-scale, survey-averaged data over the Western Weddell Sea, which is the same region as this study. According to this linear relation, $H_i = 2\mathrm{m}$ corresponds to a surface height of $\sim 0.8\,\mathrm{m}$."

**R2Q3:** *GC2: (This is similar to that from reviewer 1) While the scattering impacts of a slush layer are briefly mentioned, I feel that their impact should either be discussed further or/and incorporated into the model in some way. A slush layer at the snow-ice interface would surely effect the radar return differently than if the snow-ice interface was smooth and dry. Also, some mention of the effects of surface roughness would be beneficial, as snow surface/interface roughness has been found to influence X-band backscatter (Nandan et al. 2016, Remote Sens. Of Envir., https://doi.org/10.1016/j.rse.2016.10.004). Finally, while surface melt may not be present in this particular region or season, a wet snow surface could also influence the X-band backscatter (Dufour-Beauséjour et al. 2020, The Cryosphere, https://doi.org/10.5194/tc-14-1595-2020). This would need to be taken into account if applying this technique to other regions and/or seasons.*

**R2A3:** First, we agree that the slush layer is important for Antarctic sea ice. The reply to this comment is a repetition from earlier answers (**R1A3**). We agree with the review that a third thin high salinity layer at the snow-ice interface, whether on thin first year ice or flooded ice, is important. The discussions of slush layer have been added in Section 1 (Introduction), Section 2 (Basic conception), and the new Section 7 (Discussion), see **R1A3** for details.

Besides, we also agree that roughness plays an important role for the backscattering from surfaces and interfaces. Nandan

et al., (2016) demonstrated the roughness effects through modelling, where they assume the same roughness for the snow-air surface, the snow-snow interfaces, and the snow-ice interface, for a brine-wetted snow cover on smooth FYI ice. While the scope of our Pol-InSAR model is less detailed than their ground-based study, we account for roughness induced surface/interface backscatter effects, in relative terms, by the layer-to-volume ratios and respectively the layer-to-layer ratios. However, in contrast, we neglect an air-snow surface scattering contribution due to the required simplifications for the satellite based Pol-InSAR application. Given the quite different sea ice scenario in our study, we consider this appropriate. The following explanation has been added in Section 4.1:

"An additional parameter, the layer-to-volume scattering ratio, accounts for the (relative) scattering from these interfaces, depending e.g. on roughness and dielectric contrast (Fischer et al., 2018)."

Some mentions and references describing the effects from snow surface, especially a wet snow surface effects during melting season has been added in the new discussion Section 7.3:

"Given the sea ice scenario in this study, we assume that the snow condition over the thick and deformed ice is dry, and the X-band microwaves penetrate both the snow and ice layer. For wet-snow covered sea ice, the penetration capability of X-band is limited (Hallikainen and Winebrenner, 1992), and therefore the proposed approach in this study cannot be applied. In the case of wet snow, as well as medium wet snow, the snow-air interface influenced by the surface roughness needs to be taken into account as it changes the X-band SAR backscatter (Nandan et al., 2016; Dufour-Beauséjour et al., 2020)."

**R2Q4:** *GC3: The paper states that the influence of snow depth on $\gamma_{mod_T}$ is not negligible, and that a priori data from external sources must be used in the simplified model. If I understand correctly, the passive-microwave-derived snow depth data used as the sole model parameter results in a single snow depth value (18 cm) for each pixel across the scene. While I understand that high-resolution snow depth data is generally not available, this single value is likely not representative of the actual spatial snow depth distribution (and perhaps not realistic for heights >1.5m, as a quick hydrostatic calculation of ice thickness with this snow depth yields abnormally thick ice). Therefore, I'm curious as to the impact of the snow depth parameter on the experimental results (beyond what is shown in the simulated results of figure 8), and if/how the retrieved heights would agree with the DMS DEM under e.g. spatially-varying snow depths.*

**R2A4:** In this study, snow depth is assumed as a constant value across the scene due to the limited spatial resolution of available snow measurements. We agree with the reviewer that this single value is likely not representative of the actual spatial snow depth distribution. Therefore, in the revision, we have performed the whole inversion scheme and retrieved heights with various inputs of $z_1$. According to the snow depth on Antarctic sea ice reported in Webster et al., (2018), various $z_1$ are selected from $-0.05\,\mathrm{m}$ to $-0.75\,\mathrm{m}$. Using varying snow depths, we have demonstrated how the retrieved height would agree with the DMS DEM and discussed the impact of the snow depth parameter on the experimental results. Below texts and figures have been added in the new Section 7.2:

"In Section 4.2, we demonstrated that the influence of snow depth on the simulated coherences is not negligible, and have stated that external data of snow measurements should be used in the model. In this study, snow depth is assumed to be invariant across the scene due to the limited spatial resolution of available snow measurements. Therefore, a constant value of $z_1 = -0.18$ is used in the retrieval. Actually, snow on sea ice undergoes temporal- and spatial-variant processes and is strongly coupled with atmospheric, oceanic, and ice conditions. Thus, a single value is not representative of the actual spatial snow depth distribution. In order to assess the impact of the snow depth on the experimental results, we perform the whole inversion scheme with various inputs of $z_1$. During September and November, the snow depth on Antarctic sea ice is reported

to be maximum $\sim 1\,\mathrm{m}$ and mainly between $0$ and $0.8\,\mathrm{m}$ (Webster et al., 2018). Therefore, $z_1$ values ranging from $-0.05\,\mathrm{m}$ to $-0.75\,\mathrm{m}$ are selected. For each pixel, we retrieve heights using this range of $z_1$ values, shown as the yellow area in Fig. 5. $\Delta h_{\mathrm{mod\_S}}$ is defined as the difference between the maximum and the minimum retrieved height of every pixel. The distribution of $\Delta h_{\mathrm{mod\_S}}$ along the transect is presented in Fig. 6, where $\Delta h_{\mathrm{mod\_S}}$ has a range of $0.07 - 1.09\,\mathrm{m}$ with an average of $0.31\,\mathrm{m}$, indicating the fluctuation of model-retrieved height by using different snow depths. This analysis with various snow depth assumptions can help to constrain possible model-retrieved topographies, and $\Delta h_{\mathrm{mod\_S}}$ can be a quantitative indicator for the uncertainty of the retrieved height in the absence of high-resolution snow depth data."

**R2Q5:** *Lines 23-25: I find this sentence slightly confusing as it's written, especially since Petty et al. 2016 also mention the "close correspondence" between the predicted (surface height+square root relation) and OIB-measured thickness. Just noting the +/-2m difference makes it sound like a poor retrieval.*

**R2A5:** $\pm 2\,\mathrm{m}$ is the maximum differences between measured and predicted ice thickness. Considering that the thickness in the study area varies from $0$ to $8\,\mathrm{m}$ and the IceBridge measurements of ice thickness readily include an inherent uncertainty of $0.8\,\mathrm{m}$, this study (Petty et al. 2016) provided a useful method of understanding ice topography and thickness variability. The sentences have been changed in the Introduction:

"Petty et al. (2016) presented a detailed characterization of Arctic sea ice topography across both first-year and multi-year sea ice and analyzed the topographic differences between the two ice regimes. A square-root relation function between sea ice topographic height and thickness was established for ice thickness retrieval (Petty et al., 2016). The results demonstrated a maximum $\pm 2\,\mathrm{m}$ difference between the measured and predicted ice thickness. Note that the measured thickness ranges from $0$ to $8\,\mathrm{m}$ with an initial uncertainty of $0.8\,\mathrm{m}$ (Petty et al., 2016)."

**R2Q6:** *Lines 29-31: Since you mention that characterization of sea ice topography is an active area of research (line 28), I would suggest citing more recent studies using laser altimetry and photogrammetry (e.g. Farrell et al. 2020, https://doi.org/10.1029/2020GL09 Li et al. 2019 https://doi.org/10.3390/rs11070784; and/or others).*

**R2A6:** Thanks for the references. We have added more recent references as suggested.

- Farrell, S., Duncan, K., Buckley, E., Richter-Menge, J., and Li, R.: Mapping sea ice surface topography in high fidelity with ICESat-2,Geophysical Research Letters, 47, e2020GL090 708, https://doi.org/10.1029/2020GL090708, 2020.
- Li, T., Zhang, B., Cheng, X., Westoby, M. J., Li, Z., Ma, C., Hui, F., Shokr, M., Liu, Y., Chen, Z., Zhai, M., and Li, X.: Resolving Fine-Scale Surface Features on Polar Sea Ice: A First Assessment of UAS Photogrammetry Without Ground Control, Remote Sensing, 11,https://doi.org/10.3390/rs11070784, 2019.
- Nghiem, S., Busche, T., Kraus, T., Bachmann, M., Kurtz, N., Sonntag, J., Woods, J., Ackley, S., Xie, H., Maksym, T., et al.: Remote Sensing of Antarctic Sea Ice with Coordinated Aircraft and Satellite Data Acquisitions, in: Proc. IGARSS., pp. 8531–8534, IEEE, https://doi.org/10.1109/IGARSS.2018.8518550, 2018.

In the Introduction, the sentence has been updated as:

"The sea ice topography can be measured by various instruments, such as laser altimeters (Dierking, 1995; Schutz et al.,2005; Abdalati et al., 2010; Farrell et al., 2011, 2020) and stereo cameras using photogrammetric techniques (Dotson and Arvesen., 2012, updated 2014; Divine et al., 2016; Nghiem et al., 2018; Li et al., 2019)."

**R2Q7:** *Line 87: By previous work, do you mean Huang and Hajnsek (2021)? Or previous studies in general?*

**R2A7:** Sorry for the vague statement. The previous work refers to the work in Huang and Hajnsek (2021). This has been clarified in the revision.

**R2Q8:** *Line 91: Same as above comment. If previous work is referring to Huang and Hajnsek (2021), I would suggest writing that explicitly.*

**R2A8:** It has been clarified in the revision.

**R2Q9:** *Line 179-180: How are water-surface points selected? And how many pixels/points are used in this scene? More information would be useful to ensure that these reference surface elevations are not biased due to e.g. newly frozen leads.*

**R2A9:** In this study, 9 points/pixels are labeled from DMS images as the water surface. In addition to the information from the optical photo (i.e., DMS images), we also use the information from InSAR coherence magnitude. Since the interferometric coherence measured over water is very low, a coherence map can be used to detect the water in the sea ice cover. All 9 points are verified with InSAR coherence magnitude below 0.3, a threshold of open-water area mask used in (Huang and Hajnsek, 2021). Then, the heights of 9 points are averaged and subtracted from the DMS DEMs to obtain the sea ice topographic height relative to the local sea level.

These explanations have been added in Section 3.3:

"In total, we label nine points as water-surface reference according to the DMS images. Since the interferometric coherence magnitude over water is very low, it can also be used to classify open water (Dierking et al., 2017). All the nine points have an interferometric coherence magnitude below 0.3, which is the threshold of the open-water area mask in (Huang and Hajnsek, 2021). The average height of the open-water points is subtracted from the DMS DEMs to obtain the sea ice topographic height relative to the local sea level."

**R2Q10:** *Line 199: How many segments are removed vs used due to mis-coregistration? A percentage of rejected or accepted segments would be useful here.*

**R2A10:** The percentage of accepted segments is 76%, which means 12 segments are removed due to mis-coregistration, and the rest 38 segments are accepted for the experiments. The specific numbers have been added in Section 3.4:

"Among the 50 segments, 12 segments which still contain residual mis-coregistration induced by the sea ice non-linear movement or rotation are excluded and will not be used in the following experiments. 76% of the segments from the whole SAR scene are accepted as correctly co-registered segments in this study."

**R2Q11:** *Figure 8: This figure should have subplots labeled (a-f) on the figure, since they are referenced as such in the text. I agree with reviewer 1 that it is not apparent how phase centers are derived from these figures.*

**R2A11:** The labels (a-f) of Fig. 8 and more explanations have been added in Section 4.2. The reply to this comment is a repetition from earlier answers (**R1A12**).

**R2Q12:** *Line 393: Similar to above points, what percentage of pixels are processed (i.e. heights above 1.5m) vs not? With a scene-average height of 1.27m along the DMS DEM (line 494), I suspect that a large portion has been removed.*

**R2A12:** As we explained in R2A2, in the revision, pixels with height above $0.8\,\mathrm{m}$ have been processed into the model, which take up 83% of the total co-registered pixels. This is added in Section 5:

"In order to select the ice that is deformed and thick without seawater flooding, the samples with height above $0.8\,\mathrm{m}$, which are 83% of the co-registered data set, are selected."

**R2Q13:** *Line 398: I assume 18cm is the average snow depth of the whole region, including ice <1.5m? If only samples >1.5m are selected for processing (line 394) I am curious how your results would look if you were able to use snow depth on just the ice with elevation >1.5m. While I know this information may not be available, using some type of spatially varying snow depth assumption may help to constrain possible retrieved topographies.*

**R2A13:** Yes, $18\,\mathrm{cm}$ is the average snow depth of the whole region due to the limited spatial resolution of available snow measurements. The reply to this comment is a repetition from an earlier answer (**R2A4**). In the revision, we have performed the whole inversion scheme and retrieved heights with various snow depths from $-0.05\,\mathrm{m}$ to $-0.75\,\mathrm{m}$. We also have emphasized that this type of varying snow depth assumption will help to constrain possible retrieved topographies, see **R2A4** for details.

**R2Q14:** *Figure 13: It's fairly tough to see the DMS DEM in between grey lines in (b)-(d) and draw any conclusion about its agreement with the SAR data. I would recommend making the lines thinner or reducing the width of the zoomed sections, if possible, so that more of the DMS heights are shown. If inclined, a difference map (InSAR height – DMS height) would be useful to provide a more quantitative 2-D verification.*

**R2A14:** The width of the zoomed sections has been reduced, and Figure 13 has been updated in the revision, see Fig. 3(b)-(d) in this document. Besides, the relative retrieval bias $\epsilon = |h_{\mathrm{mod\_S}} - h_{\mathrm{DMS}}|/h_{\mathrm{DMS}}$ of zoom-in area 1-3 have been included in the revision to provide a more quantitative 2-D verification. Below texts and Fig. 3 (e)-(g) have been added in Section 6.1:

"The relative retrieval bias $\epsilon$, which can be calculated as $\epsilon = |h_{\mathrm{mod\_S}} - h_{\mathrm{DMS}}|/h_{\mathrm{DMS}}$, is used to quantify the retrieval accuracy. In Fig. 3(e)-(g), $\epsilon$ over area 1-3 are below $25\%$ for most parts, whereas only a few parts, often near to the masked-out regions (transparent pixels), present higher $\epsilon$. Note that the masked-out regions refer to water and thinner ice areas with height less than $0.8\,\mathrm{m}$. The averaged $\epsilon$ over area 1-3 are about $19\%$, $14\%$, and $15\%$, respectively, and is $18\%$ for the whole image, achieving the theoretical $25\%$-error accuracy derived in Section 4.3."

**R2Q15:** *Figure 13 also: How are heights less than 1.5m calculated in this SAR image if not selected for processing with this model? Subplot (d) in particular appears to have regions of 0m height that I suspect are not entirely physical.*

**R2A15:** In the revision, the height threshold has been changed to be $0.8\,\mathrm{m}$, and the height below $0.8\,\mathrm{m}$ have been masked out. Therefore, in the updated retrieval map (Fig. 3 in this document and Fig. 13 in the revision), the retrieved height and DMS measurements below $0.8\,\mathrm{m}$ have been set to be transparent and would not be considered in the analyses.

**R2Q16:** *Line 456: I would suggest clarifying that $h_{Model}$ in this case is the simplified model. While I understand it is in the "simplified model" section, to me the third row in Table 1 is the Theoretical model row (as it is the third method). The fact that the RMSE ranges between 0.22 and 0.27 for both models further adds to the confusion.*

**R2A16:** Sorry for the confusion. The statements have been modified as suggested in Section 6.1:

"Table 1 summarizes the performances between the retrieved height from the simplified model and $h_{\mathrm{DMS}}$ for the four polarizations."

**R2Q17:** *Line 487: This line (particularly "larger baselines respectively larger kz values") doesn't quite sound correct as written. Do you perhaps mean the possessive "baselines"?*

**R2A17:** The larger baseline means a larger value of the effective perpendicular baseline $b_\perp$ (Eq.4 in the preprint manuscript) in the interferometric SAR configuration. It has been clarified in the revision in Section 7.1:

"It reveals the potential to establish an inversion scheme by combining observations from a range of different $\kappa_z$ (i.e., the vertical wavenumber in free space), where larger values of the effective perpendicular baseline $b_\perp$, corresponding to larger $\kappa_z$

values, are expected to improve height retrieval accuracy."

**R2Q18:** *Line 499: Should be "25%-error accuracy" to be consistent with previous sections*

**R2A18:** It has been corrected to be "25%-error accuracy".

**R2Q19:** *Technical Corrections:*

- *Line 59: Icebridge -> IceBridge*
- *Line 143: iceberg -> icebergs*
- *Figure 1 caption: rectangular -> rectangle*
- *Line 215: 'flat-earth removed' should be written as 'flat-earth-removed'*
- *Line 254: Provide full names of TDF and TSX at first mention*
- *Line 470: (grammar) well correct -> e.g. adequately/sufficiently/suitably correct*
- *Line 501: comma after "For instance"*

**R2A19:** All above points haven been corrected in the revision.

We sincerely thank the editor and reviewers for helping us improving the manuscript.

---

## Author Response (AR2)

**Reply to the comments**

*"Dear Authors, thank you for the careful revisions of your manuscript and for addressing the reviewer comments. I am happy to accept your manuscript for publication, however, only after considering a flaw in your new Figure 1a:*

*Your figure shows snow ice, as a result of flooding and refreezing of slush, above the water level. However, note that the common assumption is that snow ice forms from slush which by definition (and ignoring brine wicking or other second order processes) must be below the water level. I.e. all snow ice will be located below the water level as well, and the freeboard of slush and/or snow ice is generally considered to be zero. You can easily achieve this by moving you snow and ice column down into the water until the freeboard is zero. Of course, the freeboard of snow ice could become positive if there was subsequent ice growth at the bottom of the ice sheet, without new snow accumulation, however that is a secondary effect as well, as snow thickness will generally tend to increase too.*

*In your discussion, you could also mention that the zero freeboard of snow ice, or the negative freeboard of ice in case of a slush cover, is the reason for the major difficulties of radar or laser altimetry over Antarctic sea ice. The isostatic equations to convert snow or ice freeboard to ice thickness fail in case of zero or negative freeboards. Commonly, at least with laser altimetry it is therefore assumed that the snow freeboard corresponds to snow thickness, which removes the unknown of snow thickness from the equations. Luckily, you do not attempt to calculate thickness from your data, as for surface topography alone the snow thickness to ice thickness ratio does not play a role, except for the backscatter behavior as you correctly discuss.*

*Please could you modify Figure 1a and we will proceed from there.*

*Thank you very much, best regards*

*Christian Haas."*

Dear Dr. Haas,

Thank you very much for the valuable comments and detailed explanations! As suggested, we have modified Fig.1(a) in the revision. Also, in the Discussion (Section 7.3), we have mentioned that the zero freeboard of snow ice, or the negative freeboard of ice in case of a slush cover, is the reason for the major difficulties of radar or laser altimeter over Antarctic sea ice. All the modifications have been marked in the revised manuscript.

Sincerely,

Lanqing Huang,

on behalf of all the authors